# Path-Integrated Loss-Gradient Kernels: Auditing and Similarity for Trained Neural Networks

## Abstract

Despite their success, deep neural networks remain opaque: it is often unclear why a model fails on a particular input, and classical generalization theory offers limited guidance in the overparameterized regime. Gradient-descent training naturally gives rise to path-dependent inner products between data points, but the resulting kernel matrices are asymmetric and can have negative eigenvalues, precluding their use as proper kernels or similarity measures. We show that a simple modification – replacing output gradients with loss gradients in these inner products – restores symmetry and positive semi-definiteness, yielding a Mercer kernel (the path-integrated loss-gradient kernel, PLGK). In particular, the PLGK yields (i) a fine-grained auditing decomposition that attributes how individual predictions arise from training data, and (ii) an intrinsic, behavior-based similarity measure between inputs. We validate both tools in focused experiments, including pruning studies that confirm audit-identified influences predict retraining outcomes, and a capstone analysis demonstrating that adversarial perturbations exploit a cancellation among training influences that prevents the network from learning on adversarial inputs, which can be broken by a simple mode-aware perturbation to largely restore performance.

## 1 Introduction

Despite the widespread success and adoption of DNN-based AI systems, their inner workings are still poorly understood, leading them to be popularized as black-box systems. DNNs are famously able to provide surprisingly good generalization performance despite being massively overparameterized (Zhang et al., 2016), but the inability to know 'how' a model will complete a task, whether it can complete it reliably, and the difficulty in understanding and correcting AI generated mistakes have delayed the widespread deployment of otherwise powerful AI models throughout society.

Various attempts have been made to mitigate these issues through interpretability methods, including feature-importance methods such as LIME (Garreau & Luxburg, 2020) or SHAP (Lundberg & Lee, 2017), as well as more recent work such as integrated gradients (Sundararajan et al., 2016) or Grad-CAM (Selvaraju et al., 2016). However, in many cases these interpretable features are not robust, and despite their intuitive appeal they can be used to support incorrect outcomes or provide low predictive power (Slack et al., 2020; Weber et al., 2024). Another class of interpretability techniques are known as data-based (or instance-based) methods, which attempt to explain decisions in terms of training samples. Recent work within this field includes influence functions (Koh & Liang, 2017) and Prototypes and Criticisms (Kim et al., 2016), but these techniques have not yet gained widespread popularity.

Another recent approach to understanding NNs has focused on mechanistic interpretability based on a deep understanding of how individual circuits functionally carry out relevant computations. Early works included CNN visualization techniques (Zeiler & Fergus, 2014), which were used to understand how complex decisions arose from the simple feature detectors of early layers. More recent work has discovered exciting behaviors in modern LLMs, including polysemanticity (Elhage et al., 2022), SAEs (Bricken et al., 2023), Transformer Circuit models (Elhage et al., 2021), and attribution graphs (Lindsey et al., 2025). However, a major downside of these methods is they are often only possible on simplified models or small sub-circuits and frequently take

an inordinate amount of time and effort to discover. For example, Nanda et al. (2023) recently showed how a simple model mechanically carried out simple addition via a complicated Fourier-transform-based approach that requires summation over many sinusoidal functions.

The Neural Tangent Kernel (NTK) (Jacot et al., 2018) introduced a new approach to understanding DNNs with a scalable kernel method. In the limit of infinite width, the NTK can even be used to replace NNs with a Gaussian process approximation; however, this method is consistently worse than standard DNN-based training (Vyas et al., 2022). Later NTK-based works showed that infinitely wide networks were in the so-called 'lazy' regime and couldn't learn new features, whereas 'rich' regime networks at finite width could (Atanasov et al., 2022; Kumar, 2024), allowing for an explanation of features such as the double-descent loss curve (Adlam & Pennington, 2020).

Despite their capacity to memorize, deep networks often generalize in practice, and phenomena such as double descent complicate classical intuition about overparameterization (Zhang et al., 2016; Belkin et al., 2019). In high-dimensional settings, even the distinction between "interpolation" and "extrapolation" can be ambiguous (Balestriero et al., 2021), and models with similar validation performance may behave very differently across evaluation datasets (D'Amour et al., 2022). These observations suggest that aggregate accuracy alone is an incomplete description of model behavior, motivating methods that can audit and attribute predictions to the specific training updates and examples that produced them.

In this work, we develop an NTK-based theory of how NNs learn to make decisions, showing trained networks can be shown to have feature maps that depend on the trajectory taken through parameter space during training. Our main theory contributions are (i) the loss-gradient symmetrization that converts the prior asymmetric influence-operator into a Mercer kernel, and (ii) showing that this single object yields both a principled similarity measure (IPS) and, when symmetry is relaxed, a high-fidelity audit. In the rest of the paper, we will provide the necessary background and introduce the theory (Section 3), derive a principled similarity measure (Section 4), demonstrate how our theory can be modified to produce a fine-grained 'audit' of NN decision making (Section 5), and finally provide example use cases and a real-world case study on adversarial attacks (Section 6).

## 2 Related Work

The work of Jacot *et al.* (Jacot et al., 2018) on the Neural Tangent Kernel is an important precursor to ours. They show that NNs can be used to generate a Mercer kernel that depends on the network architecture and current parameters. Under certain assumptions (equivalent to the kernel or lazy regime), this kernel does not change during training and the entire learning trajectory can be characterized via a linear differential equation, i.e. kernel gradient descent. Although theoretically insightful, in practice, lazy-regime networks have been shown to perform significantly worse than their adaptive regime counterparts (Vyas et al., 2022) (unsurprisingly, a network being able to adaptively learn and reshape its internal representations during training leads to better performance).

Influence functions (Koh & Liang, 2017) estimate the counterfactual effect of removing or upweighting a single training example on a model's predictions, using an inverse-Hessian-vector product to approximate the change in parameters that would result from retraining without that example. While theoretically grounded, they require access to (or approximation of) the full Hessian, which is expensive and can be unstable for large or non-convex models. Nevertheless, they remain a widely used baseline for data attribution.

The TracIn technique (Pruthi et al., 2020) develops a way to track the 'influence' of training data, assessing whether points are proponents or opponents of a given test point given their effects on the test loss. They show that this can be useful for a variety of downstream tasks, including identifying outliers, unsupervised clustering, and analyzing model failures from top influences. This is in many ways a precursor to our audit technique, but has several downsides. The first is very high computational cost, requiring them to 'throw out' much of the TracIn influence information in order to actually generate results (all results are generated over the last layer only). Additionally, TracIn is developed only for SGD, leading to failures or lower accuracy when using other optimizers or features such as gradient clipping or weight decay. Finally, TracIn's forward-Euler discretization incurs $O(\|\Delta\theta\|^2)$ error per step; our trapezoidal correction (Section 5.1) reduces (since

$\Delta\theta$ is small) this to $O(\|\Delta\theta\|^3)$ at no additional gradient cost, since both endpoint gradients are already available when iterating over checkpoints.

Recently, Wang et al. (2024) developed a new technique (which they call in-run data Shapley) that allows for computing Shapley-style influence scores – based on the cooperative game-theory framework of Shapley et al. (1953), which attributes a fair share of a collective outcome to each contributor – without retraining models on new data subsets. In practice, this is essentially an extension of Google's TracIn, with the addition of a new, highly effective 'ghost dot-product' technique that provides a massive computational speedup, sidestepping one of the major issues with TracIn, and allowing them to scale up to LLM-sized models for experiments. However, without utilizing a proper kernel-based theory, their technique suffers from accuracy issues; These accuracy issues may be significant, as they report only single-iteration errors of $\tilde{3}e^{-4}$, (on one of 3500 training iterations). In addition, they also restrict themselves to standard SGD, leaving extensions to common optimizers such as Adam for future work.

TRAK (Tracing with the Randomly-projected After Kernel) (Park et al., 2023) introduces a data attribution method that achieves strong counterfactual prediction performance while remaining computationally tractable for large-scale models. Their approach approximates the trained model with an instance of kernel regression using the "after kernel" (a linearization around the final trained parameters), then applies random projections to reduce dimensionality and ensembles over a small number of independently trained models. They demonstrate strong results across image classifiers, CLIP, and language models (BERT, mT5), significantly outperforming influence functions and TracIn on their proposed linear datamodeling score (LDS) metric while requiring far fewer model retrains than Shapley-based approaches. However, TRAK is fundamentally aimed at counterfactual data attribution (estimating the effect of removing a training sample), rather than faithfully reconstructing the actual training dynamics of a specific model. As discussed in Remark 5.3, this is a distinct goal from auditing, which seeks to decompose the realized influences within a particular training run. Additionally, TRAK's reliance on the after kernel (a linearization at the final parameters) means it does not track how representations evolve throughout training, in contrast to the path-integrated feature maps central to PLGK. Finally, TRAK requires training multiple independent models (typically 20-50 for best results), whereas our auditing framework operates on a single training run.

Recent work (Domingos, 2020) has shown that every model learned via gradient descent is approximately a kernel machine (e.g. output predictions are kernel-weighted sums of training inputs). In particular, they show that it is a type of kernel they call a path kernel, which is dependent on the path taken through parameter space throughout training. This path kernel can be decomposed into a sum of standard kernels, each of which can simplify to a NTK. However, their path kernel formulation is not a true kernel machine; concretely, Domingos's path kernel for predicting output at test point x takes the form $K_{path}(x,x') = \sum_t \epsilon(x,t)\phi(x,t) \cdot \phi(x',t)$, where the one-sided loss weighting $\epsilon(x,t)$ depends on the test point. This renders the object an influence operator (Definition 2, below) rather than a kernel: the Gram matrix is asymmetric and can have negative eigenvalues, precluding its use as a similarity measure or for geometric analysis. Our key modification–switching from output gradients to loss gradients–symmetrizes both sides of the inner product, yielding a true PSD kernel (Theorem 3.8). As a pure theory paper, they also make onerous assumptions, such as full-batch gradient descent, and provide no empirical tests or experiments. Although this work starts from a similar baseline, we instead develop a true Mercer kernel formulation by using a loss kernel (rather than previous NTK-style output kernels), and extend it beyond theory into usable tools, including empirical validation on more realistic use cases (e.g. Adam trained classifiers).

## 3 Neural Networks as Loss Kernels

### 3.1 Notation

We parameterize a neural network (NN) as a function $f$, which takes in inputs $x$ and targets $y$ and generates responses depending on parameters $\theta$: $\hat{y} = f(x; \theta)$. Our training set is $D_{\text{tr}}$, a dataset of $M$ training samples $D_{\text{tr}} := \{(x_m, y_m)\}, m \in [M]$, while our test set is $D_{\text{te}} := \{(x_n, y_n)\}, n \in [N]$. We abbreviate point-wise outputs as $\hat{y}_m = f(x_m; \theta)$, while $\hat{y} = f(x; \theta)$ represents the overall output function. Our NN is assumed to be trained via a variant of gradient descent, with learning rate $\eta$. Our NN is parameterized by a vector of $P$

parameters, $\theta \in \mathbb{R}^P$; we can write outputs as $f(x; \theta)$. The NN is also trained using a loss function $L(\hat{y}, y)$, which over a fixed training set can also be written as being purely dependent on parameters $\theta$, i.e. $L(\theta)$, or as a function of the data $x$ i.e. $L(x)$. Parameters $\theta$ (and thus $\hat{y}$) change throughout training, and we call the parameters at timestep $\theta_t$, however we usually suppress dependence on $t$. We denote matrices via bold (e.g. a kernel function $K(x, x')$ empirically measured at samples $(x_n, x_m)$ would yield a matrix $\mathbf{K}$ with entries $K_{nm}$).

### 3.2 Preliminary Setup

**Definition 1** (PSD kernel and Gram matrix). *A symmetric function $K \colon \mathcal{X} \times \mathcal{X} \to \mathbb{R}$ is a positive semi-definite (PSD) kernel if, for every $n \geq 1$ and every $\{x_1, \dots, x_n\} \subset \mathcal{X}$, the Gram matrix $\mathbf{K} \in \mathbb{R}^{n \times n}$ with entries $K_{ij} = K(x_i, x_j)$ satisfies*

$$\sum_{i,j} c_i \, c_j \, K_{ij} \; \geq \; 0$$

*for all $c \in \mathbb{R}^n$. By the Moore–Aronszajn theorem, $K$ is PSD if and only if there exists a Hilbert space $\mathcal{H}$ and a map $\phi \colon \mathcal{X} \to \mathcal{H}$ such that*

$$K(x, x') = \langle \phi(x), \, \phi(x') \rangle_{\mathcal{H}}.$$

*Throughout this work, $\mathcal{H} = \mathbb{R}^d$ with the standard inner product, and all kernels are empirical (evaluated on finite datasets). We use 'Mercer kernel' to mean a kernel function whose every finite Gram matrix is PSD.*

**Definition 2** (Influence operator). *Given feature maps $\phi \colon \mathcal{X} \to \mathbb{R}^P$ and a scalar weighting function $\epsilon \colon \mathcal{X} \to \mathbb{R}$, an influence operator is a function $A \colon \mathcal{X} \times \mathcal{X} \to \mathbb{R}$ of the form*

$$A(x, x') \; = \; \big( \epsilon(x) \, \phi(x) \cdot \, \phi(x') \big).$$

*The one-sided weighting generically breaks symmetry $(A(x, x') \neq A(x', x))$, so the associated Gram matrix need be neither symmetric nor PSD. In particular, influence operators can have negative eigenvalues and do not define valid similarity measures.*

**Proposition 3.1** (Completeness identity). *Let $f \colon \mathbb{R}^P \to \mathbb{R}$ be any differentiable function of the parameters $\theta$ (e.g. the loss value, or some other output such as logit or class probability). For any parameter update $\Delta \theta$, the fundamental theorem of calculus along the line segment $\theta(\lambda) = \theta + \lambda \, \Delta \theta$ yields*

$$f(\theta + \Delta \theta) - f(\theta) = \int_0^1 \nabla_\theta f\big(\theta + \lambda \, \Delta \theta\big)^\top \Delta \theta \; \mathrm{d}\lambda. \tag{1}$$

Equation equation 1 is exact and converts a change in an observable into an integral of inner products between gradients and the update direction. In practice, we approximate the integral with a low-order quadrature rule. The trapezoidal approximation gives

$$f(\theta + \Delta \theta) - f(\theta) \approx \tfrac{1}{2} \Big( \nabla_\theta f(\theta) + \nabla_\theta f(\theta + \Delta \theta) \Big)^\top \Delta \theta, \tag{2}$$

which is the basis for our high-fidelity discrete-time audit. We call this a *completeness* identity because it exactly decomposes the total change in $f$ into an integral of local contributions along the update path, with no residual; this property will underpin the completeness criterion for audits (Definition 9).

**Lemma 3.2** (Trapezoidal error bound). *If $f$ is twice differentiable and its Hessian is Lipschitz along the segment $\{\theta + \lambda \, \Delta \theta : \lambda \in [0, 1]\}$ with constant $L$ (i.e., $\|\nabla_\theta^2 f(\theta') - \nabla_\theta^2 f(\theta'')\| \leq L \|\theta' - \theta''\|$ for all $\theta', \theta''$ on the segment), then*

$$\left| \big( f(\theta + \Delta \theta) - f(\theta) \big) - \tfrac{1}{2} \big( \nabla_\theta f(\theta) + \nabla_\theta f(\theta + \Delta \theta) \big)^\top \Delta \theta \right| \leq \frac{L}{12} \, \|\Delta \theta\|^3, \tag{3}$$

*i.e., the error is $\mathcal{O}(\|\Delta \theta\|^3)$.*

*Remark* 3.3 (Piecewise-linear activations). For ReLU networks, the network function is piecewise linear in $\theta$, so within each activation region the Hessian is constant and the trapezoidal rule is exact. At activation boundaries, however, the Hessian is discontinuous and the Lipschitz condition of Lemma 3.2 does not formally hold. In practice, individual gradient-descent steps are small enough that few activation boundaries are crossed per step, limiting the accumulated error. Our empirical reconstruction errors confirm that the trapezoidal approximation remains tight throughout training.

### 3.3   Theory overview and Roadmap

Deep networks are typically treated as black boxes, yet training is fully specified by an optimization trajectory given by $\{\theta_t\}_{t=0}^{T}$. Our goal is to make model behavior auditable by expressing changes in an observable output directly in terms of training updates, with the ability to further break down influences across training samples or model parameters.

1. *NTK View*: Linearizing $f(x;\theta)$ around $\theta_t$ yields the tangent features $\phi(x) := \nabla_\theta f(x;\theta)$ and classical NTK $K(x,x') = \phi(x) \cdot \phi(x')$. Notably, the NTK is a kernel, but does not track network changes over learning time.

2. *Function Update View*: The NTK cannot be directly used to determine

$$\Delta f(x;\theta) = f(x;\theta + \Delta\theta) - f(x;\theta).$$

   The key insight is that

$$\Delta\theta = -\frac{\eta}{M} \sum_{m \in D_{\mathrm{tr}}} \frac{\partial L(\theta_t)}{\partial \hat{y}_m} \nabla_{\theta_t} \hat{y}_m,$$

   and thus

$$\Delta f(x;\theta) = \int_0^1 \nabla_\theta f(x;\theta + \lambda\Delta\theta)\Delta\theta \, \mathrm{d}\lambda \,.$$

   Taking a forward Euler approximation,

$$\Delta f(x;\theta) \approx -\frac{\eta}{M} \sum_{m \in D_{\mathrm{tr}}} \nabla_\theta \hat{y}(x) \frac{\partial L_m(\theta)}{\partial \hat{y}_m} \nabla_\theta \hat{y}_m.$$

   Note the almost-kernel form; it is similar to the NTK, but one side is multiplied by an additional $\epsilon(x) := \nabla_{\hat{y}} L(x)$. We call this the loss-weighted NTK: $lNTK(x,x') = (\epsilon(x)\phi(x)) \cdot \phi(x')$, which is an influence operator. However, the lNTK is not PSD, precluding its use as a similarity measure or for stable geometric analysis.

3. *Path Features*: the lNTK explains single-step dynamics. The path-integrated version (PNTK) can be used to explain the entire trajectory of $\theta_t$. This can be done via a simple summation over time, or alternatively by concatenating the feature maps: for a kernel $K(x,x';t) = \phi(x,t) \cdot \phi(x',t)$, we have $\sum_t K(x,x';t) = \sum_t \phi(x,t) \cdot \phi(x',t) = [\phi(x,0),\cdots,\phi(x,T)] \cdot [\phi(x',0),\cdots,\phi(x',T)]$. Note that this is still an influence operator rather than a kernel.

4. *Recovering Kernel Form*: Working with $\Delta\theta$ necessitates one part of our influence operator consisting of $\nabla_{\hat{y}} L(\theta_t) \nabla_\theta \hat{y}$. Thus, we can recover our kernel form if we analyze $L(x)$ instead of $f(x;\theta)$. We call this the loss gradient kernel: $LGK(x,x') = (\epsilon(x)\phi(x)) \cdot (\epsilon(x')\phi(x'))$. We also introduce the path-integrated PLGK, set up the same way as going from the lNTK to the PNTK. Notably the LGK and PLGK, are true kernels again, but describing network loss rather than network output.

5. *Applications*: We develop two main applications. The first is Intrinsic Perceptual Similarity (IPS), which exploits the kernel form of the PLGK to obtain a principled similarity measure (and its induced distance metrics) for trained neural networks. The other is auditing, which may discard the true kernel nature of the PLGK in order to generate a high fidelity influence operator that can provide a high-accuracy and complete influence breakdown of model outputs. Although IPS and auditing are developed and validated independently in the sections that follow, they share the same core computation: inner products of loss gradients $(\epsilon_m \phi_m) \cdot (\epsilon_n \phi_n)$ over examples $x_m, x_n$ accumulated over training time. The two tools differ in whether the kernel symmetry is preserved (IPS) or relaxed in favor of output-level fidelity (auditing), but their underlying feature representations are identical. In particular, training samples that appear as top audit influences for a given test point will generically have high IPS similarity to that point, and vice versa; the two views are complementary projections of the same path-integrated gradient structure.

### 3.4 The Loss-Weighted NTK (lNTK)

Consider gradient descent on a neural network with scalar learning rate $\eta$ and parameters $\theta \in \mathbb{R}^P$. Assuming the loss decomposes as a mean over a training set $D_{\mathrm{tr}}$ of $M$ samples and depends on $\theta$ only through the network outputs $\hat{y}_m$, the chain rule gives

$$\theta_{t+1} = \theta_t - \frac{\eta}{M} \sum_{m \in D_{\mathrm{tr}}} \epsilon_m \, \phi_m, \tag{4}$$

where we define, for each sample $x_m$:

**Definition 3** (Loss sensitivity and feature vector). *The* loss sensitivity *and* feature vector *at time $t$ are*

$$\epsilon_m(t) := \frac{\partial L(\theta_t)}{\partial \hat{y}_m(\theta_t)} \in \mathbb{R}, \qquad \phi_m(t) := \nabla_\theta \hat{y}_m(\theta_t) \in \mathbb{R}^P.$$

The empirical neural tangent kernel (eNTK) is the Mercer kernel $K_{mn}(t) = \phi_m \cdot \phi_n$ (Jacot et al., 2018). A standard calculation shows that the continuous-time evolution of the network output at any point $x_n$ is $\frac{\mathrm{d}\hat{y}_n}{\mathrm{d}t} = -\frac{\eta}{M} \sum_{m \in D_{\mathrm{tr}}} K_{mn}(t) \, \epsilon_m(t)$ (see, e.g., Jacot et al. 2018). Motivated by this, we define:

**Definition 4** (Loss-weighted NTK (lNTK)). *For arbitrary inputs $x, x'$ the* loss-weighted NTK *is*

$$\mathrm{lNTK}(x, x'; \theta_t) := \epsilon(x) \, K(x, x'; \theta_t) = (\epsilon(x) \, \phi(x)) \cdot \phi(x'). \tag{5}$$

The lNTK gives the instantaneous training influence of $x$ on the network output at $x'$. Note that the lNTK is not itself a kernel, as the one-sided $\epsilon(x)$ weighting breaks symmetry, instead rendering it an influence operator.

**Proposition 3.4** (Single-step approximation). *For a single gradient-descent step,*

$$\hat{y}_n(\theta_t) \approx \hat{y}_n(\theta_{t-1}) - \frac{\eta}{M} \sum_{m \in D_{tr}} \mathrm{lNTK}(x_m, x_n; \theta_{t-1}), \tag{6}$$

*where the approximation error arises from the forward-Euler discretization and is $O\left(\|\Delta\theta\|^2\right)$.*

Equation (6) shows that the network's discrete-time evolution acts as a kernel machine with data-dependent weights $\epsilon_m = \frac{\partial L}{\partial \hat{y}_m}$. See Section 5 for more on the discretization error.

### 3.5 Path-Dependent Kernels

*Remark* 3.5 (Kernel / lazy regime). If the model is approximately linear in $\theta$, also called the *kernel* or *lazy* training regime (Chizat et al., 2019), then the basis functions (i.e. penultimate-layer activations) are effectively fixed, the NTK varies slowly, and the entire learning trajectory is interpretable as a linear kernel machine (Ortiz-Jiménez et al., 2021).

*Remark* 3.6 (Adaptive regime). All modern neural networks of practical interest operate in the *adaptive regime*, where basis functions evolve during training and a significant performance gap separates adaptive from kernel-regime models (Arora et al., 2019). In this regime the NTK becomes time-dependent, $K(x, x'; t)$, and classical NTK theory no longer applies. The lNTK dynamics derived in Section 3.4, however, remain valid at each time step.

Summing the single-step lNTK contributions (Proposition 3.4) over the training trajectory $t = 0, \dots, T-1$ and exchanging the order of summation yields:

**Definition 5** (Path-integrated Neural Tangent Kernel (PNTK)). *For training sample $x_m$ and an arbitrary evaluation point $x_n$, the* PNTK *is the cumulative training influence*

$$P_{mn} := \sum_{t=0}^{T-1} \epsilon_m(t) \, \phi_m(t) \cdot \phi_n(t) = \sum_{t=0}^{T-1} \mathrm{lNTK}(x_m, x_n; \theta_t). \tag{7}$$

*As with the lNTK, the one-sided loss weighting means the PNTK is not a Mercer kernel (cf. the "loss-weighted path kernel" of Domingos 2020).*

**Proposition 3.7** (Trajectory decomposition of network output)**.** *Under the forward-Euler discretization, the network output at any point $x_n$ after $T$ gradient-descent steps satisfies*

$$\hat{y}_n(\theta_T) \approx \hat{y}_n(\theta_0) - \frac{\eta}{M} \sum_{m \in D_{tr}} P_{mn}, \tag{8}$$

*expressing the trained model as its initialization plus a per-sample attribution summed over the entire training path. The error is bounded by $\sum_t O\left(\|\Delta\theta_t\|^2\right)$.*

The trajectory dependence shown by Proposition 3.7 is well supported by prior work: Jacot et al. (2018) show explicit dependence on initialization and architecture; gradient-based optimizers impose an implicit inductive bias that shapes the trajectory (Lyu & Li, 2019; Wilson et al., 2017); and methods such as stochastic weight averaging, which aggregate parameters across trajectory segments, yield improved performance over any single path (Izmailov et al., 2018).

### 3.6 Loss-Gradient Kernel (LGK) and Path-Integrated Loss-Gradient Kernel (PLGK)

The lNTK and PNTK (Sections 3.4 and 3.5) explain changes in network *output* $\hat{y}$. We now show that a simple modification explains changes in network *loss* $L(\hat{y}, y)$ and, crucially, recovers a true Mercer kernel.

The only change is that the rate of change of loss at an evaluation point acquires a second loss-sensitivity factor:

$$\frac{\mathrm{d}L(\hat{y}(\theta_t), y)}{\mathrm{d}t} = -\frac{\eta}{M} \sum_{m \in D_{\mathrm{tr}}} \big(\epsilon_m(t)\,\phi_m(t)\big) \cdot \big(\epsilon(t)\,\phi(t)\big), \tag{9}$$

where $\epsilon(t)\,\phi(t)$ denotes the loss gradient at the evaluation point. That is, the rate of change of loss is the sum over all training samples of the inner product between each sample's loss gradient and the evaluation point's loss gradient.

**Definition 6** (Loss-Gradient Kernel (LGK))**.** *The* single-step loss-gradient kernel *between samples $x_m$ and $x_n$ at time $t$ is*

$$\mathrm{LGK}_{mn}(t) := \big(\epsilon_m(t)\,\phi_m(t)\big) \cdot \big(\epsilon_n(t)\,\phi_n(t)\big). \tag{10}$$

**Definition 7** (Path-Integrated Loss-Gradient Kernel (PLGK))**.** *The* PLGK *accumulates the LGK over the training trajectory:*

$$P'_{mn} := \sum_{t=0}^{T-1} \mathrm{LGK}_{mn}(t) = \sum_{t=0}^{T-1} \big(\epsilon_m(t)\,\phi_m(t)\big) \cdot \big(\epsilon_n(t)\,\phi_n(t)\big). \tag{11}$$

*Under the forward-Euler discretization, the loss at any point $x_n$ after $T$ steps satisfies*

$$L\big(\hat{y}_n(\theta_T),\, y_n\big) \approx L\big(\hat{y}_n(\theta_0),\, y_n\big) - \frac{\eta}{M} \sum_{m \in D_{tr}} P'_{mn}. \tag{12}$$

Unlike the lNTK and PNTK, both sides of the inner product in equation 10 carry a loss-sensitivity factor. This symmetry has an important consequence:

**Theorem 3.8** (The PLGK is a Mercer kernel)**.** *If $f$ and $L$ are differentiable, then the PLGK is a symmetric positive semi-definite kernel with feature map*

$$\Phi^{\mathrm{path}}(x) := \big(\, \epsilon(x,0)\,\phi(x,0),\ \epsilon(x,1)\,\phi(x,1),\ \ldots,\ \epsilon(x,T{-}1)\,\phi(x,T{-}1)\,\big) \in \mathbb{R}^{TP}, \tag{13}$$

*i.e. the concatenation of the loss gradients $\nabla_{\hat{y}(x)}L\nabla_\theta\hat{y}(x)$ across all training steps.*

*Proof.* Expanding the sum in equation 11:

$$\begin{aligned}
P'_{mn} &= \Big[\big(\epsilon_m(0)\,\phi_m(0)\big) \cdot \big(\epsilon_n(0)\,\phi_n(0)\big) + \big(\epsilon_m(1)\,\phi_m(1)\big) \cdot \big(\epsilon_n(1)\,\phi_n(1)\big) + \cdots\Big] \\
&= \underbrace{\big(\epsilon_m(0)\,\phi_m(0),\ \epsilon_m(1)\,\phi_m(1),\ \ldots\big)}_{\Phi_m^{\mathrm{path}}} \cdot \underbrace{\big(\epsilon_n(0)\,\phi_n(0),\ \epsilon_n(1)\,\phi_n(1),\ \ldots\big)}_{\Phi_n^{\mathrm{path}}} \\
&= \Phi_m^{\mathrm{path}} \cdot \Phi_n^{\mathrm{path}}. 
\end{aligned} \tag{14}$$

Any matrix of the form $\Phi^\top \Phi$ (up to a constant) is symmetric PSD, so $\mathbf{P}'$ is a Mercer kernel. $\qquad\square$

*Remark* 3.9 (Design insight vs. proof technique). The proof of Theorem 3.8 is deliberately simple: any inner product of shared feature maps is PSD. The substantive contribution is not the proof itself, but the identification that switching from output gradients $\partial\hat{y}/\partial\theta$ to *loss* gradients $(\partial L/\partial\hat{y})(\partial\hat{y}/\partial\theta)$ symmetrizes the one-sided weighting that renders the lNTK and PNTK mere influence operators (Definition 2). This symmetrization is what separates PLGK from prior path-kernel formulations (Domingos, 2020) and enables the geometric tools (IPS, cosine distance) developed in Section 4, while the influence-operator form is deliberately recovered in Section 5 when fidelity is prioritized over kernel structure.

## 4 Intrinsic Perceptual Similarity

*Remark* 4.1 (The PLGK is not a similarity measure). Although $\mathbf{P}'$ with $P'_{mn} = \Phi_m^{\mathrm{path}} \cdot \Phi_n^{\mathrm{path}}$ is a PSD kernel matrix (Theorem 3.8), the raw Gram matrix $\mathbf{P}'$ is not directly useful as a similarity measure. Informally, a similarity measure should be bounded and should assign maximal similarity to identical inputs; $\mathbf{P}'$ satisfies neither, since scaling the loss sensitivity makes entries arbitrarily large, and $\arg\max_i \mathbf{P}'_{n,i} \neq i$ in general. The raw PLGK captures something akin to *gradient covariance* over training rather than similarity.

A standard remedy is cosine normalization, which preserves the kernel property:

**Definition 8** (Intrinsic Perceptual Similarity (IPS))**.** *For a trained neural network with path feature map* $\Phi^{\mathrm{path}}$ *(Equation (13)), the* intrinsic perceptual similarity *between inputs $x$ and $x'$ is*

$$\mathrm{IPS}(x, x') := \frac{\Phi^{\mathrm{path}}(x) \cdot \Phi^{\mathrm{path}}(x')}{\|\Phi^{\mathrm{path}}(x)\| \, \|\Phi^{\mathrm{path}}(x')\|}. \tag{15}$$

IPS is bounded in $[-1, 1]$ (by Cauchy-Schwarz) with $\mathrm{IPS}(x, x) = 1$, and measures *gradient correlation* across the training trajectory.

**Corollary 4.1.1** (IPS-induced distance metrics)**.** *IPS induces two distance metrics on the feature space* $\{\Phi^{\mathrm{path}}(x) : x \in \mathcal{X}\}$*. Writing them as functions of the corresponding inputs:*

$$d_{\mathrm{chord}}(x, x') = \sqrt{2 - 2\,\mathrm{IPS}(x, x')}, \qquad d_{\mathrm{angle}}(x, x') = \arccos\big(\mathrm{IPS}(x, x')\big). \tag{16}$$

*Remark* 4.2. The distances in Corollary 4.1.1 are true metrics when viewed as operating on the feature representations $\Phi^{\mathrm{path}}(x)$: distinct feature vectors always yield positive distance. As functions of inputs, however, they are pseudo-metrics, since $\Phi^{\mathrm{path}}(x)$ is not guaranteed to be injective – distinct inputs may share the same feature representation and thus have zero distance.

*Remark* 4.3 (Label dependence). Because the loss gradient $\nabla_{\hat{y}}L$ requires a target label, computing IPS for held-out data requires ground-truth labels. IPS is therefore a supervised diagnostic for labeled validation or audit sets, not a label-free out-of-distribution score. Label-free variants (e.g., replacing the cross-entropy loss sensitivity with a self-supervised surrogate such as logit entropy) are a natural extension that we leave to future work.

## 5 Auditing Neural Networks

We now specialize our theory to *auditing*: decomposing the change in a network's output (loss or otherwise) at each evaluation point into per-training-sample contributions. Unlike IPS, which preserves the symmetric kernel structure of the PLGK, auditing prioritizes reconstruction fidelity – closely recovering the true change in network outputs – and in doing so introduces modifications (such as trapezoidal corrections and asymmetric train/audit-side gradients) that break the PSD kernel property. The resulting decomposition is an influence operator rather than a kernel, but achieves near-exact reconstruction of training dynamics. Concretely, an audit produces a matrix $\widetilde{\mathbf{P}} \in \mathbb{R}^{M \times N}$ over training indices $m$ and audit indices $n$, where we call $\widetilde{P}_{mn}$ the influence of $m$ on $n$. This breakdown enables questions such as: which training samples most caused the failure of a mispredicted test example; which samples were most influential for a given class; or how audit points cluster by their training-set influences.

**Definition 9** (Completeness)**.** *An audit $\widetilde{\mathbf{P}}$ is* complete *if summing over the training set exactly recovers the true change in the audited quantity. For the loss auditing case:*

$$L(\hat{y}_n(\theta_T), y_n) = L(\hat{y}_n(\theta_0), y_n) + \sum_{m \in D_{tr}} \widetilde{P}_{mn}, \qquad \forall\, x_n \in D_{\text{audit}}. \tag{17}$$

In practice, numerical integration introduces small errors, so completeness serves as the target criterion against which audit quality is measured rather than an exact guarantee. The PNTK and PLGK decompositions (Proposition 3.7, Definition 7) provide approximate audits via forward-Euler discretization. We now analyze and reduce this approximation error.

### 5.1 Trapezoidal Correction

Consider a single parameter step $\Delta\theta$ and its effect on the loss at audit point $x_n$. The forward-Euler audit assumes

$$L\big(\hat{y}_n(\theta + \Delta\theta)\big) \approx L\big(\hat{y}_n(\theta)\big) + \Delta\theta^\top \nabla_\theta L(\hat{y}_n(\theta)), \tag{18}$$

i.e. that the mean gradient over the step equals the initial gradient. By the fundamental theorem of calculus (cf. Proposition 3.1), the exact (complete) change is

$$L\big(\hat{y}_n(\theta + \Delta\theta)\big) = L\big(\hat{y}_n(\theta)\big) + \int_0^1 \Delta\theta^\top \nabla_\theta L(\hat{y}_n(\theta + \gamma\,\Delta\theta))\ \mathrm{d}\gamma\,. \tag{19}$$

*Remark* 5.1 (Trapezoidal approximation for auditing)*.* The integral in equation 19 can be approximated by the trapezoidal rule at no additional cost: if $\theta_t$ and $\theta_{t+1} = \theta_t + \Delta\theta$ are from successive training steps, both gradients will be available as we iterate over training steps. This yields an $\mathcal{O}(\|\Delta\theta\|^3)$ error per step (Lemma 3.2), compared with $\mathcal{O}(\|\Delta\theta\|^2)$ for forward Euler (an improvement for small $\Delta\theta$).

### 5.2 Audit Update Equations

**Definition 10** (Prediction audit)**.** *Given a GD-trained network with learning rate $\eta$, the* trapezoidal prediction audit *accumulates $\widetilde{P}_{mn}(0) = 0$ with update*

$$\widetilde{P}_{mn}(t+1) = \widetilde{P}_{mn}(t) - \frac{\eta}{M}\,\nabla_\theta L(\hat{y}_m(\theta_t))^\top \left[ \tfrac{1}{2}\nabla_\theta \hat{y}_n(\theta_t) + \tfrac{1}{2}\nabla_\theta \hat{y}_n(\theta_{t+1}) \right]. \tag{20}$$

**Definition 11** (Loss audit)**.** *The* trapezoidal loss audit *replaces the test-side output gradient with the test-side loss gradient:*

$$\widetilde{P}'_{mn}(t+1) = \widetilde{P}'_{mn}(t) - \frac{\eta}{M}\,\nabla_\theta L(\hat{y}_m(\theta_t))^\top \left[ \tfrac{1}{2}\nabla_\theta L(\hat{y}_n(\theta_t)) + \tfrac{1}{2}\nabla_\theta L(\hat{y}_n(\theta_{t+1})) \right]. \tag{21}$$

Both updates require only storing the gradient at $\theta_t$, performing the parameter update, and computing the gradient at $\theta_{t+1}$. We refer to the individual entries $\widetilde{P}_{mn}$ (or $\widetilde{P}'_{mn}$) as the *influence* of training sample $x_m$ on audit point $x_n$: a negative value indicates that $x_m$'s contribution drove the audited quantity (output or loss) downward over training, while a positive value indicates the opposite.

### 5.3 Accuracy Metrics

Since completeness (Definition 9) is verifiable, we can directly measure audit quality. Let $q(X, t)$ denote the quantity of interest evaluated on data set $X$ at time $t$: for output auditing $q(X, t) = \hat{y}_X(\theta_t)$, for loss auditing $q(X, t) = L(\hat{y}_X(\theta_t))$. Write $\widetilde{\mathbf{P}}^*$ for whichever audit matrix is used ($\widetilde{\mathbf{P}}$ or $\widetilde{\mathbf{P}}'$).

**Definition 12** (Mean reconstruction error (MRE))**.**

$$\mathrm{MRE} = \operatorname*{mean}_n \left( \Big| \sum_{m \in D_{tr}} \widetilde{P}^*_{m,n} - \big(q(D_{\text{audit}}, T) - q(D_{\text{audit}}, 0)\big) \Big| \right) \tag{22}$$

MRE measures the mean absolute gap between the audit's reconstructed change and the true change in $q$ over training; a complete audit would achieve MRE $= 0$.

**Definition 13** (Loss (or prediction) correlation (LC))**.**

$$\text{LC} = \underset{n}{\text{corr}}\left( \sum_{m \in D_{tr}} \widetilde{P}_{m,n}^*, \; q(D_{\text{audit}}, T) - q(D_{\text{audit}}, 0) \right) \tag{23}$$

*Remark* 5.2 (Super- and sub-sampling)*.* If these metrics reveal insufficient accuracy, the numerical integration can be refined by evaluating gradients at intermediate points within each step (supersampling; see Appendix A.2.7), arbitrarily improving precision. Conversely, if accuracy is high, computational cost can be reduced by accumulating $\Delta\theta$ over several training steps before updating $\widetilde{\mathbf{P}}^*$ (subsampling).

### 5.4 Relationship to Influence Functions

*Remark* 5.3 (Audit influence vs. counterfactual influence)*.* The audit measures each training sample's *actual contribution* within a particular training run, in contrast to influence functions (Koh & Liang, 2017), which estimate the *counterfactual* effect of removing a sample. To illustrate the distinction: if a data point $x_k$ is duplicated $K$ times, the influence function of any single copy is minimal (removing one copy barely changes the trained model), whereas the audit attributes approximately $1/K$ of $x_k$'s total learning effect to each copy. Both notions are valuable, and the appropriate choice depends on the application.

**Auditing Extensions**    The above kernel methods were developed for a simple GD system, and provides the minimal auditing baseline, designed to demonstrate near perfect fidelity at high cost. Auditing can be further expanded, including partial decompositions, or decomposing auditing across parameters and training time, rather than training data, or decomposing across groups of training data (or parameters, etc) rather than individual data points. The auditing and IPS frameworks can also be expanded to support more advanced features such as more advanced optimizers, learning rate schedulers, normalization, and other features used in realistic tasks. See Appendix A.2 for expansions of the the kernel-derived applications to more realistic use cases, including speed/accuracy tradeoffs.

## 6 Experiments

### 6.1 Auditing Demonstration

*Remark* 6.1 (Single-model analysis)*.* Auditing, by construction, decomposes the behavior of a *specific* trained model: the audit matrix $\widetilde{\mathbf{P}}$ depends on the realized optimization trajectory $\{\theta_t\}_{t=0}^T$, and its entries are deterministic given that trajectory. A single-run analysis is therefore the natural unit of auditing – just as one profiles a specific compiled binary rather than averaging over compiler runs, the goal is to faithfully reconstruct the behavior of the particular model a practitioner has trained and deployed. The question of whether the qualitative findings (e.g. adversarial mode cancellation) are robust across random seeds is distinct from audit fidelity; we provide evidence of robustness by replicating key findings across two different dataset–architecture pairs (MNIST/LeNet-5 and SVHN/CNN, Appendix A.16), which vary in initialization, training trajectory, and data distribution.

We demonstrate auditing on a simple test case, a LeNet5 network trained on the MNIST task. The auditing process captures the training dynamics with very high fidelity, as seen in Figure 1. (See Appendix A.11 for a comparison with the TracIn technique).

Next, we will use auditing to analyze a single, incorrect decision, shown in Figure 2.

What could have caused this error? Since a relatively small amount of data determines the majority of the response (the graph is near zero for most inputs), looking at a small amount of the most helpful (drives example loss down) and harmful (drives example loss up) training data may be insightful (Figure 3)

As expected, the most helpful training data are various 4s (noticeably, frequently those with unusually wide or deep 'top pockets'). However, why are 6s so harmful? To better understand, we can zoom in on the more

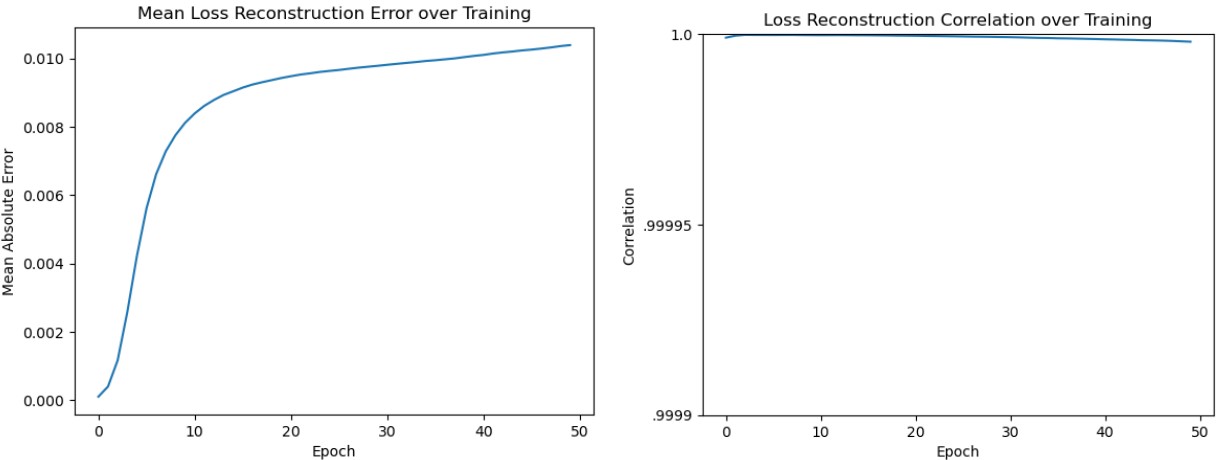

Figure 1: Audit reconstruction results over training time. Left: Mean Reconstruction Error between true dynamics, audit reconstruction. Right: correlation between true and reconstructed dynamics

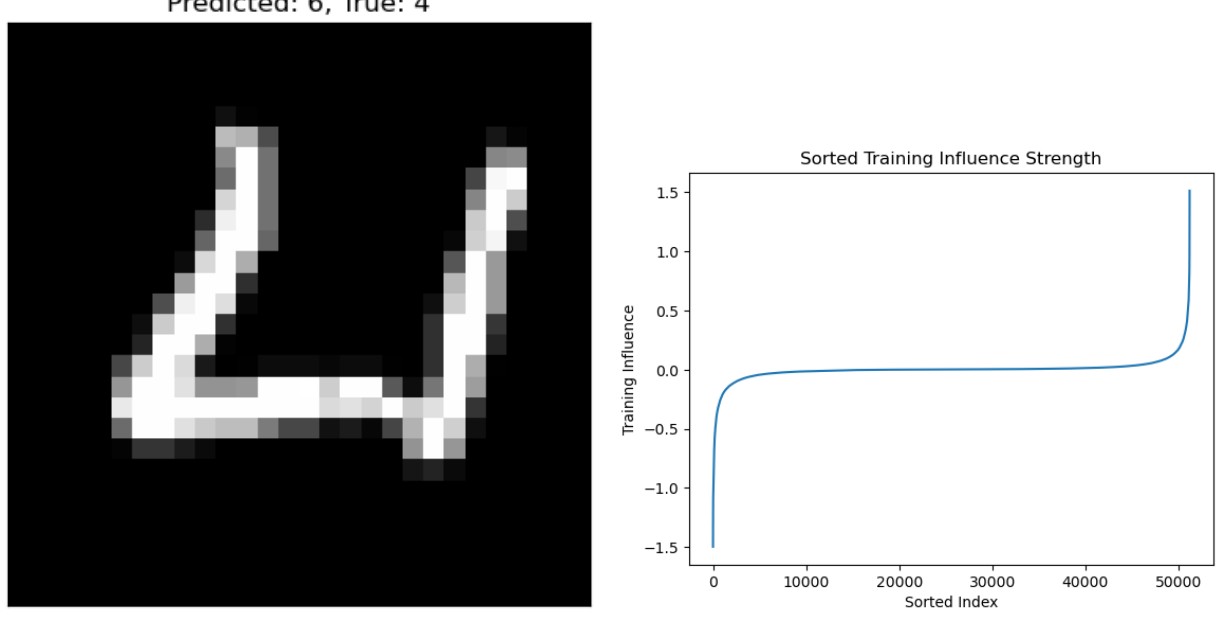

Figure 2: Auditing a single wrong decision Left: Input 4 misidentified as a 6 by the trained network. Right: Sorted per-training-sample influences, showing that a relatively small number of training examples determine the final output

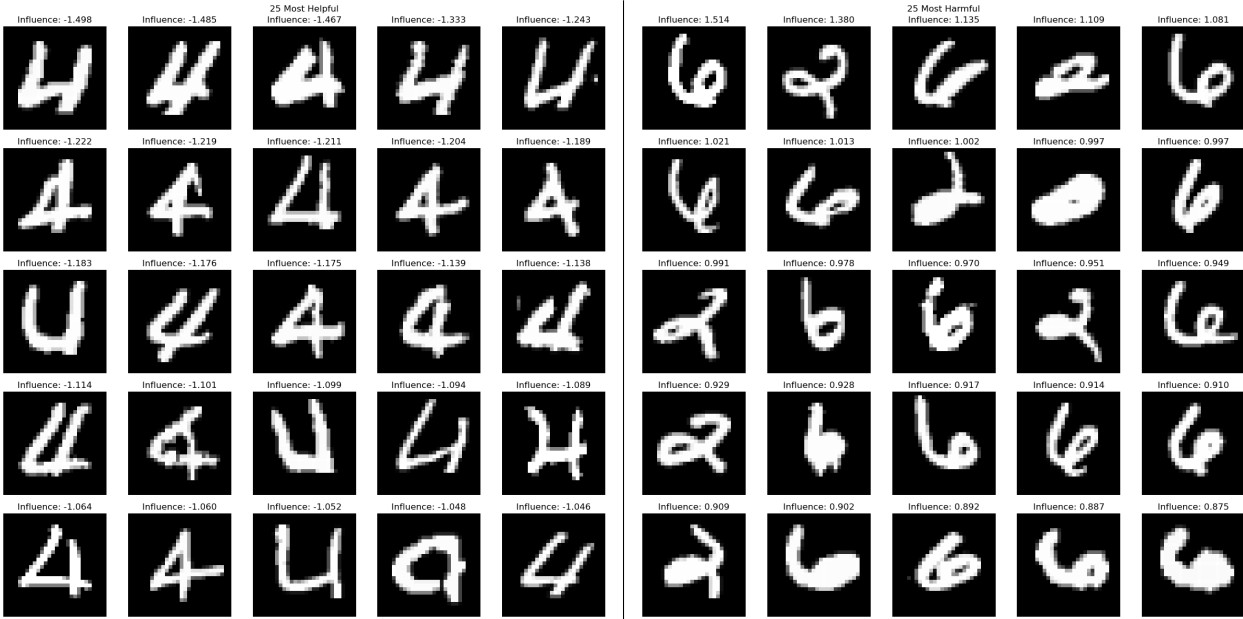

Figure 3: Left: 25 most helpful training data points. Right: 25 most harmful training data points.

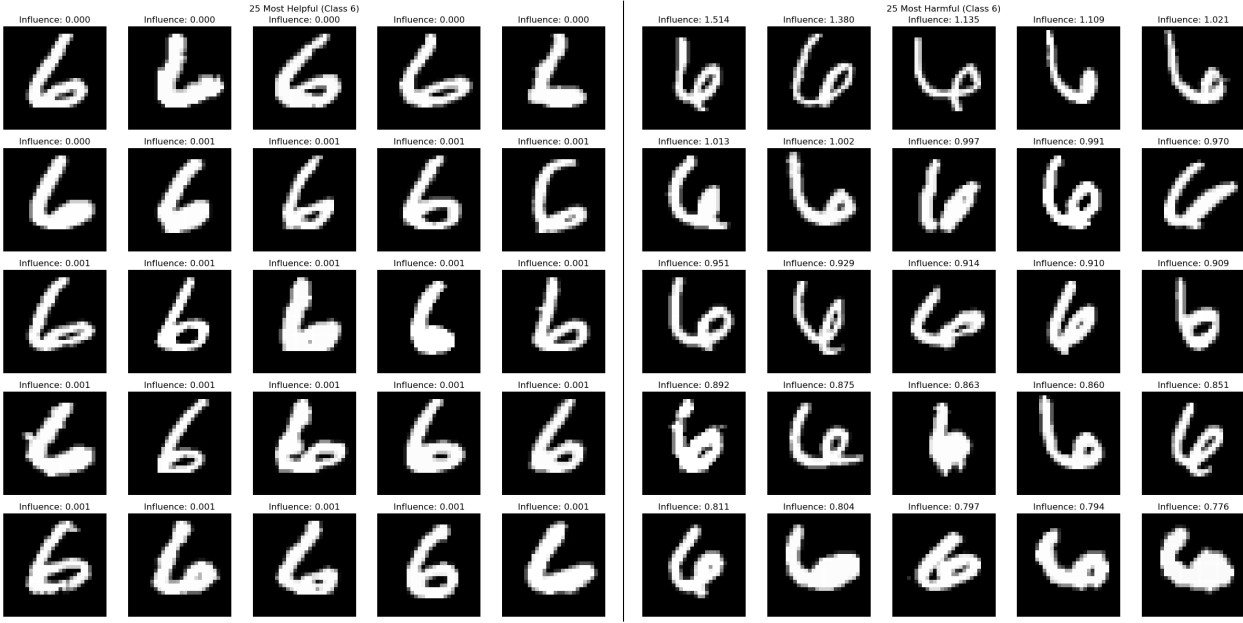

Figure 4: Left: 25 most helpful training data points of class 6. Right: 25 most harmful training data points of class 6.

helpful and harmful training data that are class 6 (Figure 4), revealing that there is a noticeable difference: harmful 6s are tilted horizontally to the left (especially on the initial downstroke), and have a larger or more vertically aligned 'pocket' between the downstroke and the bowl.

## 6.2 IPS Demonstration

We compute IPS using the same task setup as in auditing, with the similarity metric being updated once per epoch. We project the IPS similarity matrix into 2D with UMAP (McInnes et al., 2018) for plotting

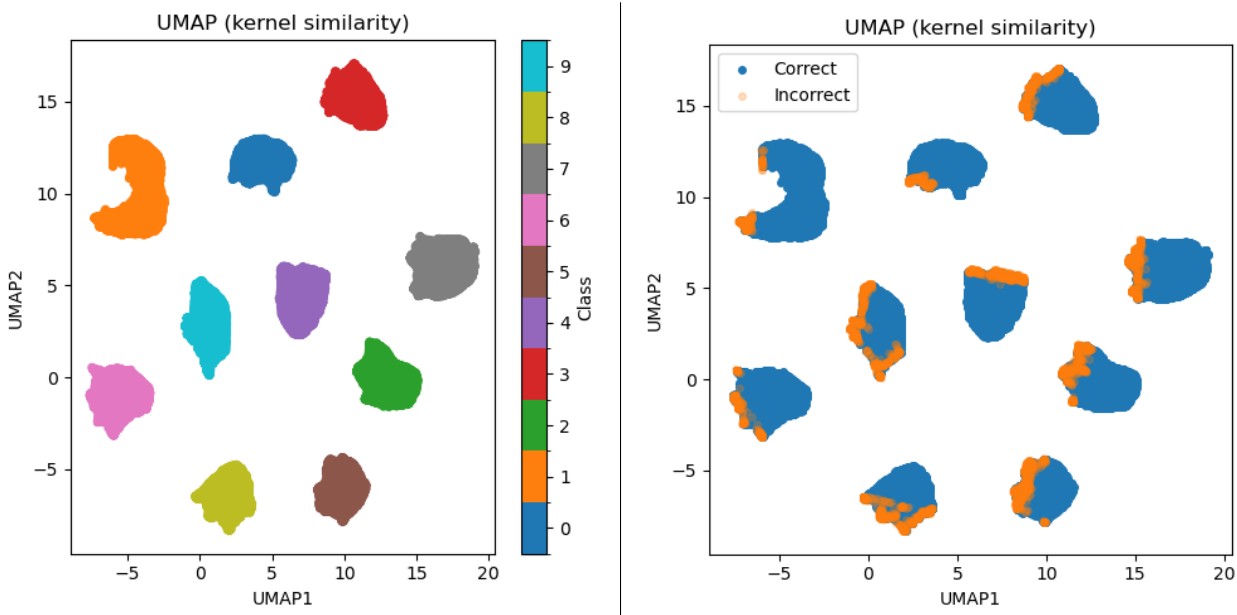

Figure 5: Left: 2D UMAP projection of similarity metric, colored by true class index. Right: 2D UMAP projection of similarity metric, with misclassified points marked. Notice that they are on the outer edge of class clusters

purposes. Our proposed metric captures several key features, including class-based clustering with errors clustering towards the edges of classes (Figure 5)

As a sanity check, we examine the IPS values projected into the 2D UMAP space (Figure 6). The results confirm expected behavior: within-class similarities are positive, between-class similarities are negative, and similarity scales smoothly with UMAP distance in both cases. This validates that the IPS metric and its UMAP projection are mutually consistent.

Next, we demonstrate that the similarity (projected onto UMAP dimensions) captures relevant feature-level information about our data points. In Figure 7, we demonstrate this by showing the highest and lowest points per UMAP dimension that come from class 7. We can see that dimension 1 codes for long top strokes, the presence of a crossbar, and a short height at low values, and for a taller 7 without optional flourishes at high values. Dimension two codes for a short top stroke and an exaggerated sideways downstroke at low values, and for a long top stroke and an initial extra downward serif on the left end of the top stroke at high values.

While IPS is primarily motivated as a tool for analyzing training dynamics, we also ask whether simple scalar summaries of IPS can serve as *supervised* difficulty scores on labeled audit sets. Concretely, on MNIST with LeNet-5 we compute IPS between each test example and the training set, and compute two IPS-based scalars: the $\ell^2$ norm of IPS similarities and a class-conditioned IPS margin (i.e. the difference between mean IPS similarity to training examples of the predicted class and the highest mean similarity to any other class). We then evaluate whether these scalars predict (i) which labeled test examples are misclassified and (ii) their per-example loss. We compare against max-softmax probability, logit margin, and a baseline that computes cosine similarity between test and training examples in the penultimate hidden layer, averaged over the $k$ most similar training examples per class ($k = 10$).[1]

On this MNIST setup, the $\ell^2$ norm is highly predictive of misclassification (AUROC $\approx 0.90$), essentially matching max-softmax and logit margin and outperforming hidden-state similarity. The class-conditioned IPS margin separates correct and incorrect examples almost perfectly (AUROC $\approx 1.0$) and shows strong

---

[1]Note that IPS for test examples is computed using their ground-truth labels, so these are *not* label-free OOD scores but supervised diagnostics. See Remark 4.3

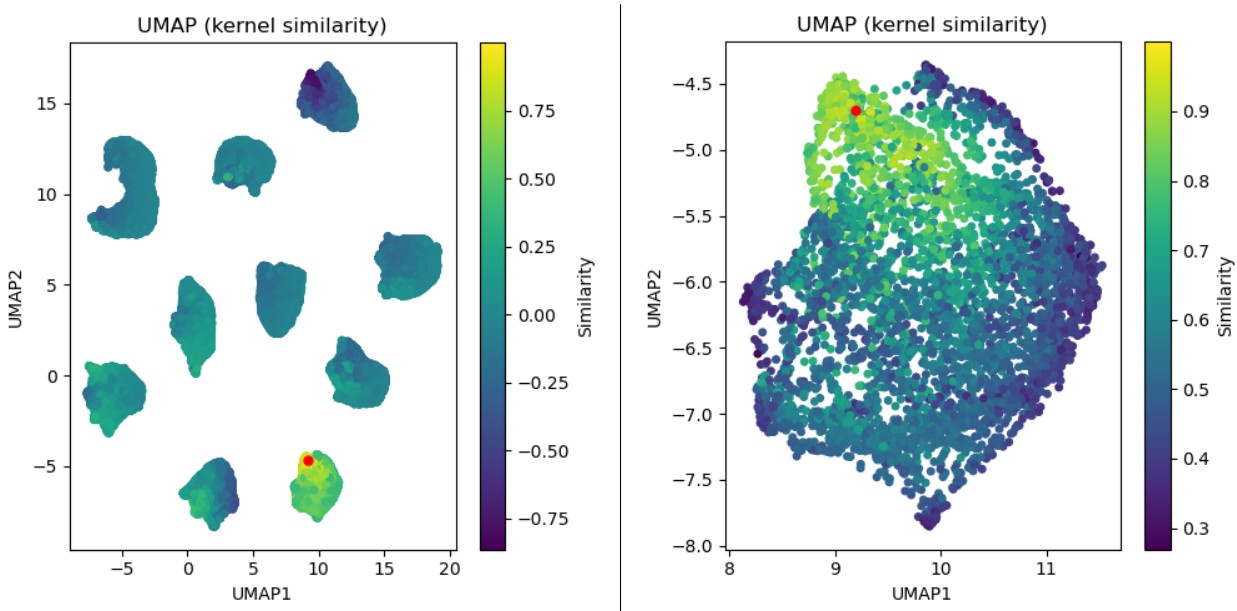

Figure 6: Left: 2D UMAP projection of similarity metric, showing similarity vs a target point. Right: Same, but zoomed into only the same class as the target point.

| Score | AUROC (misclassification) | AUPR (misclassification) | Spearman$(s, -L)$ |
|---|---|---|---|
| IPS $\ell^2$ norm | 0.9009 | 0.5008 | 0.2667 |
| IPS class margin | **1.0000** | **1.0000** | 0.8046 |
| Hidden top-$k$ sim | 0.8633 | 0.3929 | 0.6253 |
| Max-softmax (MSP) | 0.9054 | 0.4964 | **0.9823** |
| Logit margin | 0.9016 | 0.4786 | 0.9729 |

Table 1: **IPS-based scores as supervised difficulty predictors on MNIST.** AUROC/AUPR measure how well each scalar score distinguishes misclassified from correctly classified test examples (higher is better). Spearman correlation is computed between the score $s(x)$ and negative cross-entropy loss $-L(x)$ (higher means better alignment with low loss).

correlation with per-example loss (Spearman $\approx 0.80$), substantially higher than the hidden-state baseline. As expected, logit-based confidence scores remain nearly perfectly rank-correlated with loss, reflecting their direct relationship to the cross-entropy objective. The most informative comparisons are therefore the *label-agnostic* scores (IPS $\ell^2$ norm, hidden top-$k$ similarity) and the Spearman correlation with per-example loss, where IPS remains competitive with standard confidence baselines despite operating in a fundamentally different feature space.

## 6.3 Adversarial Analysis

We next use our auditing techniques to analyze adversarial examples. We first take our (trained) MNIST network, and then use the foolbox package (Rauber et al., 2020) to generate a series of adversarial attacks on a subset of testing data[2]. We then audit the model (both prediction and loss audit), constructing the audit target set from a random selection of training examples together with adversarial and random perturbations of those examples (with the magnitude of the random perturbations matching the magnitude of the successful adversarial perturbations); Figure 8 demonstrates a few training examples with adversarial and random perturbations. Audit reconstruction performance was excellent, with the loss audit reaching a

---

[2]We use 4 attacks for all of our results: Fast Gradient Sign Method (FGSM), PGD-$L_\infty$, Basic Iterative Method (BIM-$L_\infty$), and DeepFool-$L_\infty$. All (non-illustrative) results are shown averaged across all 4 attacks.

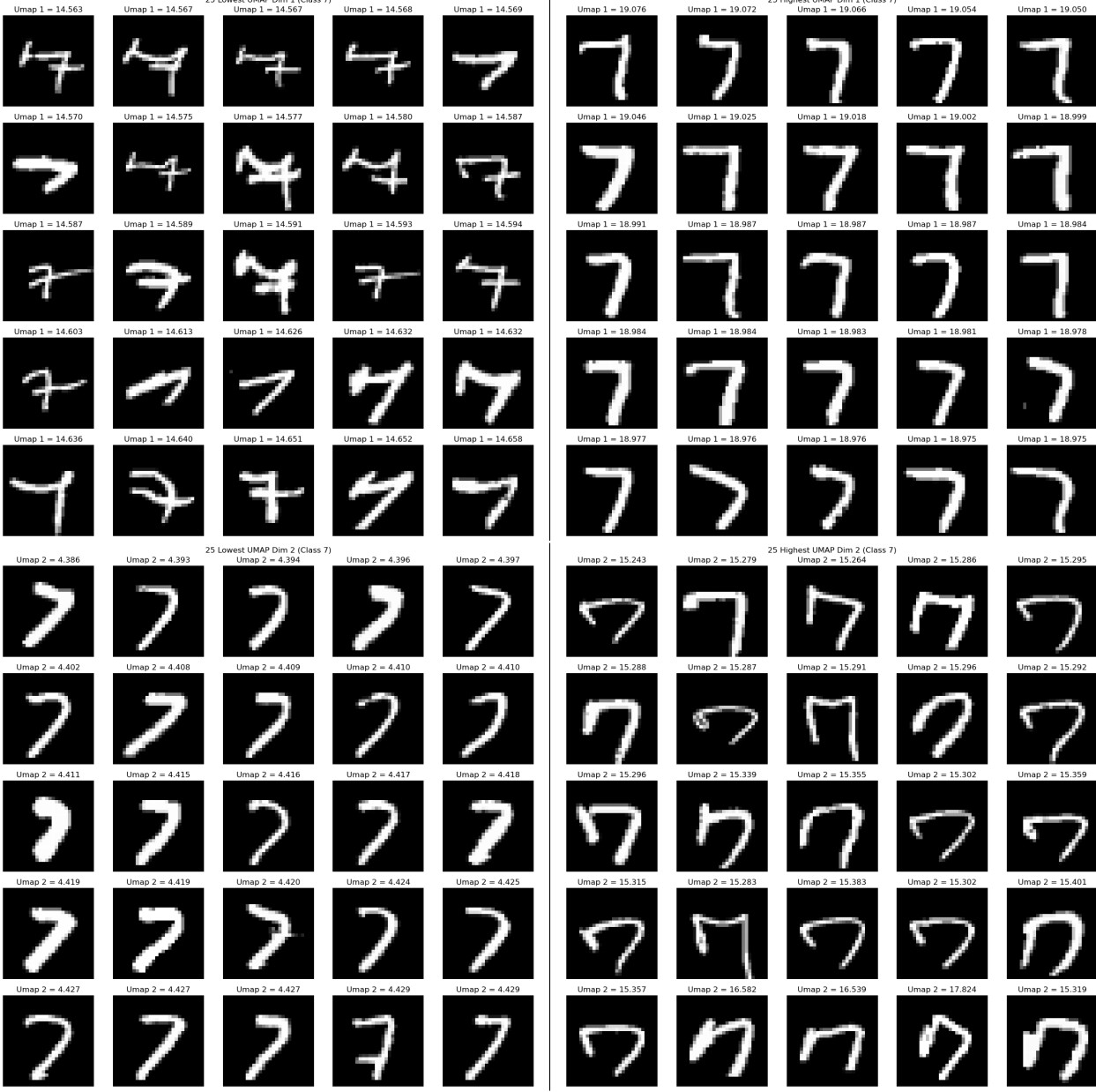

Figure 7: Top Left: 25 lowest UMAP dimension 1 valued class 7 examples. Top Right: 25 highest UMAP dimension 1 valued class 7 examples. Bottom Left: 25 lowest UMAP dimension 2 valued class 7 examples. Bottom Right: 25 highest UMAP dimension 2 valued class 7 examples. We can see clear differences in the 7s across the two dimensions.

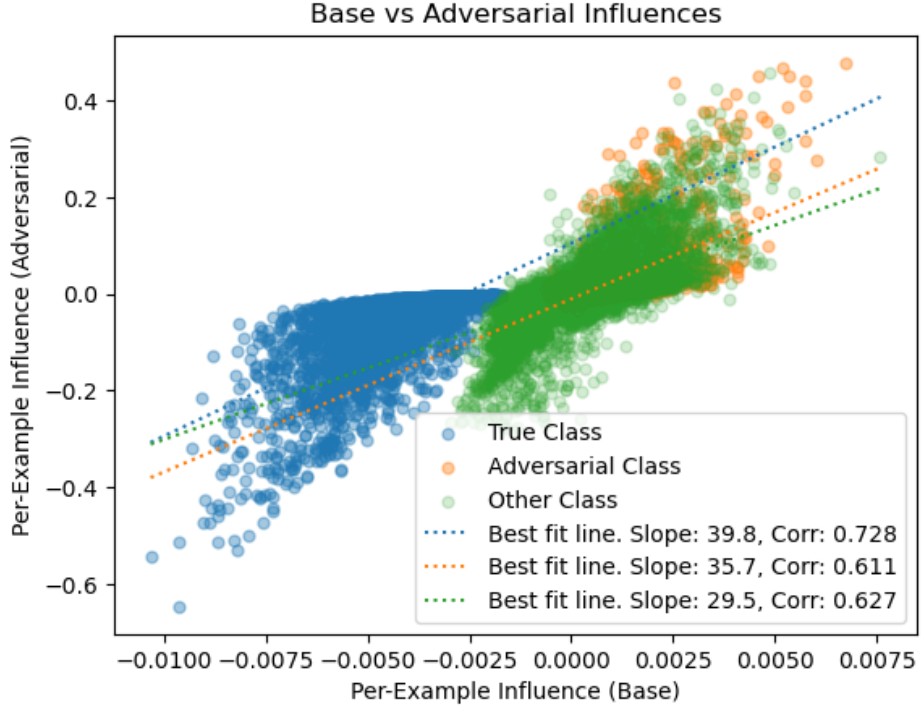

Figure 8: Three data point sets, each consisting of a clean image, an adversarial image, and 2 randomly perturbed images.

Figure 9: Scatter plot of Clean vs Adversarial per-example influences. This reveals that influences are broadly stronger across the entire train set.

final MRE of 0.01, loss correlation of 0.999997, while the prediction audit reached a final MRE of 0.019, and a prediction correlation of 0.9999999.

Adversarial examples, by construction, incur higher loss than their clean counterparts. Our audit framework lets us ask: does this elevated loss arise from a few highly influential training points, or from a broad shift across the training set? Comparing per-sample influences for clean versus adversarial inputs (Figure 9) reveals that influence *magnitudes* increase broadly across the entire training set, rather than being concentrated in a handful of examples. This broad amplification is moderately well approximated by a uniform linear scaling of the clean influences (correlation $> 0.6$).

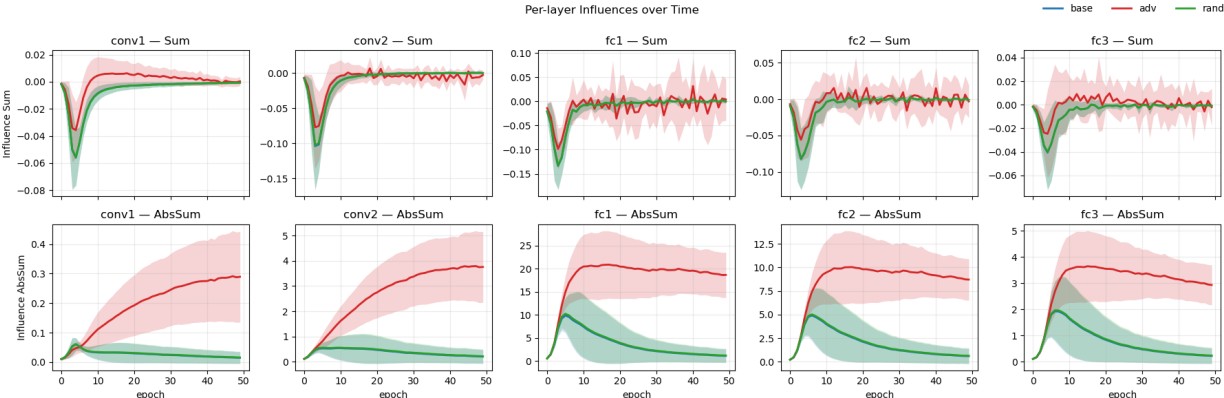

Figure 10: Plots showing the mean and standard deviation over time across example types of the per-layer influence (column), using both the sum of influences (top) and sum of absolute influence (bottom). These reveals that adversarial perturbations appear to primarily be traced to epoch 10 onward in the first 2 fully connected layers.

We can further decompose the audit by restricting the parameter index to individual network layers and retaining the temporal index per epoch (see Appendix A.1 for details), allowing us to determine which layers and training phases contribute most to the difference between clean and adversarial outputs. For each layer and epoch, we report both the signed sum of influences $\sum_m \widetilde{P}'_{mn}$ (which captures net learning direction) and the absolute sum $\sum_m |\widetilde{P}'_{mn}|$ (which captures total influence magnitude regardless of sign), averaged over examples in each group (clean/adversarial/random), broken down across the 5 layers of the Lenet5 architecture and our 50 epochs of training. The results are shown in Figure 10. They show that there is a striking temporal pattern: significant growth in influence within the first 10 epochs, followed by a predictable behavior for the remaining 40 (exponential decay for all cases in the summation setting, exponential decay for base and random perturbations in the absolute summation case, and leveling off for adversarial examples in the absolute summation case). There is also a large difference in scale between the base/random and the adversarial examples in the absolute summation case but *not* the summation case. The two settings also reveal different effects across layers - in the summation case, scales are approximately the same between layers, varying by a factor of no more than 2. However, in the absolute summation case, per-layer scales vary by approximately 2 orders of magnitude, peaking on the first fully-connected layers, suggesting that they may be responsible for most of the adversarial attack's influence.

To better understand the adversarial influence changes, we zoom into a single base/adversarial pair (the first set shown in Figure 8), and analyze data points on the 'perturbation ray' from the baseline image to either the random perturbation or the adversarial perturbation. We also capture audit snapshots throughout training, allowing us to perform a finer scale analysis. We begin by examining how influences develop along the perturbation paths. We consider the temporal profile and the overall scaling of the total influence strength of the examples on the base to adversarial perturbation path. As seen in Figure 11, the temporal patterns change from being initially front loaded (all significant influences occur early in training) to increasingly linear throughout time along the adversarial path, while the total influence magnitude super-linearly increases along the adversarial path. Thus, compared to a random direction (which sees no noticeable change, not shown), the adversarial direction consists of one that receives an enormous influence from the training data throughout the training process, even after the base example stops receiving influence.

For adversarials, final influence magnitudes (measured as $\frac{1}{|G|} \sum_{n \in G} \sum_m |\widetilde{P}'_{m,n}(t)|$, where $G$ is the clean, adv, or rand group) remain elevated, whereas they are small for both clean and randomly perturbed inputs (Figure 12). Notably, loss has largely stopped changing for both clean and adversarial groups by the end of training – but for opposite reasons: clean examples have already been learned (low loss), while adversarial examples remain stuck at high loss despite receiving large-magnitude influences that cancel.

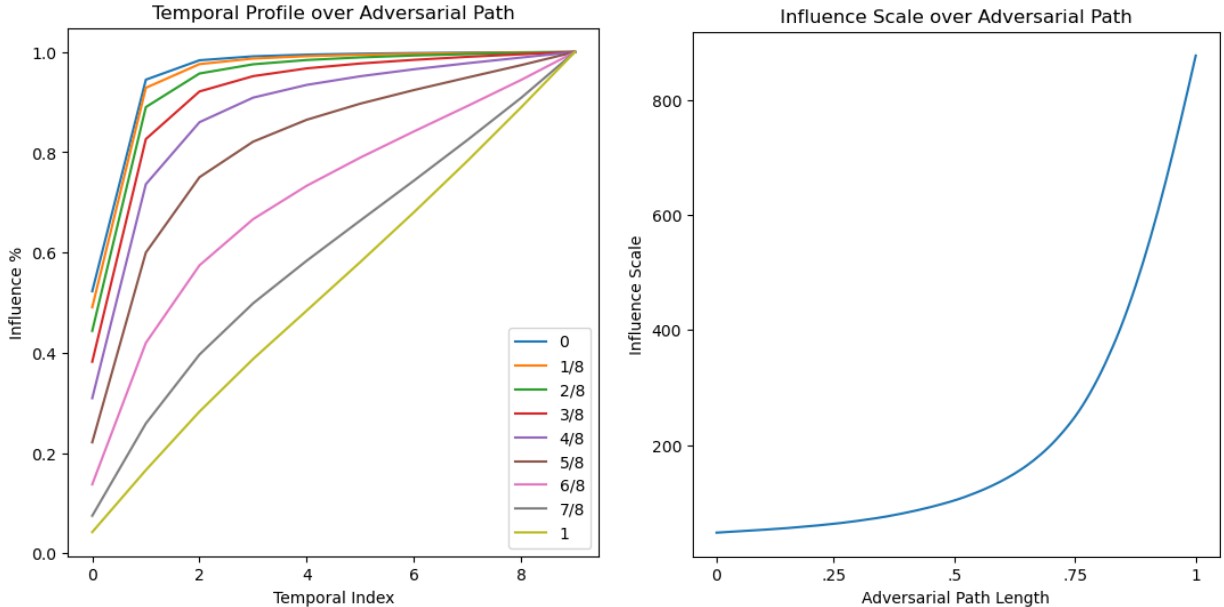

Figure 11: Left: Relative temporal contributions (as a % of final total influence) for points along the linear interpolation from a clean input to its adversarial counterpart. Each line corresponds to a different position along this path (0 = clean, 1 = adversarial); the $x$-axis indexes training epochs. Clean inputs receive influence primarily early in training, while points closer to the adversarial end receive influence throughout. Right: Total influence magnitude as a function of position along the adversarial path, showing a super-linear increase toward the adversarial endpoint.

What might explain this behavior? One hypothesis is a *cancellation* effect, where per-example influences have high magnitudes but opposite signs, so that their sum remains small. To test this, we define two cumulative quantities for each audit point $x_n$ at epoch $t$: the *net influence* $S_n(t) = \sum_m \widetilde{P}'_{m,n}(t)$, which captures the signed total effect of training on the loss, and the *gross influence* $G_n(t) = \sum_m |\widetilde{P}'_{m,n}(t)|$, which captures total influence magnitude regardless of sign. If cancellation is occurring, we expect $G_n$ to grow while $S_n$ levels off. Results from Figure 13 confirm this prediction for adversarial examples: net influence plateaus while gross influence continues to grow. In contrast, random perturbations show minimal values for both quantities.

The temporal patterns suggest a two-phase account of training, as has previously been observed in other works (**?**). During the first $\sim$5 to 10 epochs, the network is in the adaptive regime, reshaping its internal representations to form the directions along which loss reduction is most efficient. After this, the network enters a kernel-like regime where representations are approximately fixed and loss decays exponentially toward zero. This would explain the adversarial dynamics: early in training, both clean and adversarial examples benefit from loss reduction. Once representations stabilize, adversarial examples (which produce high net influence magnitude ($G_n$) while keeping signed influence ($S_n$) small) experience cancellation among training influences and cease to learn, while clean examples have already converged to low loss.

To test this hypothesis, we measure how far the network has progressed toward the kernel regime using three convergence metrics, each comparing training step $t$ to the final step $T$:

(i) Feature function overlap: $\rho_{\text{feat}}(t) = \frac{(\epsilon(x,t)\phi(x,t) \cdot \epsilon(x,T)\phi(x,T))}{\|\epsilon(x,t)\phi(x,t)\| \, \|\epsilon(x,T)\phi(x,T)\|}$, averaged over training examples. This measures how much the direction of the loss-gradient at time $t$ has converged to its final orientation.

(ii) Subspace overlap: the mean cosine similarity between the subspaces spanned by the top-10 right singular vectors of $\Phi_{\text{train}}(t)$ and $\Phi_{\text{train}}(T)$, measuring whether the dominant gradient directions have stabilized, where $\Phi_{\text{train}} \in \mathbb{R}^{M \times P}$ with $m$-th row the loss-gradient feature $\epsilon_m \phi_m$ for training sample $x_m$.

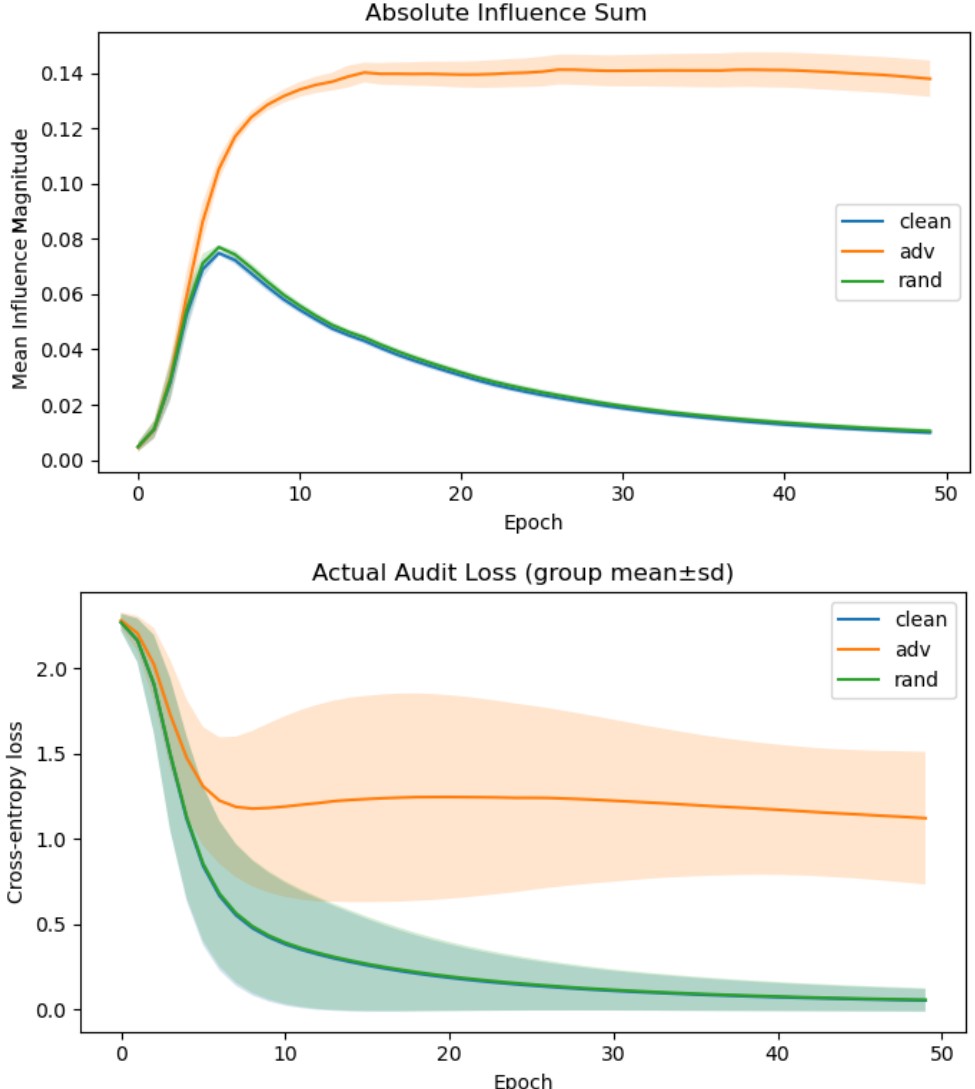

Figure 12: Top: Per-epoch mean influence magnitude, compared across clean, adversarial, and randomly perturbed inputs. Bottom: Per-epoch mean loss across the same three groups.

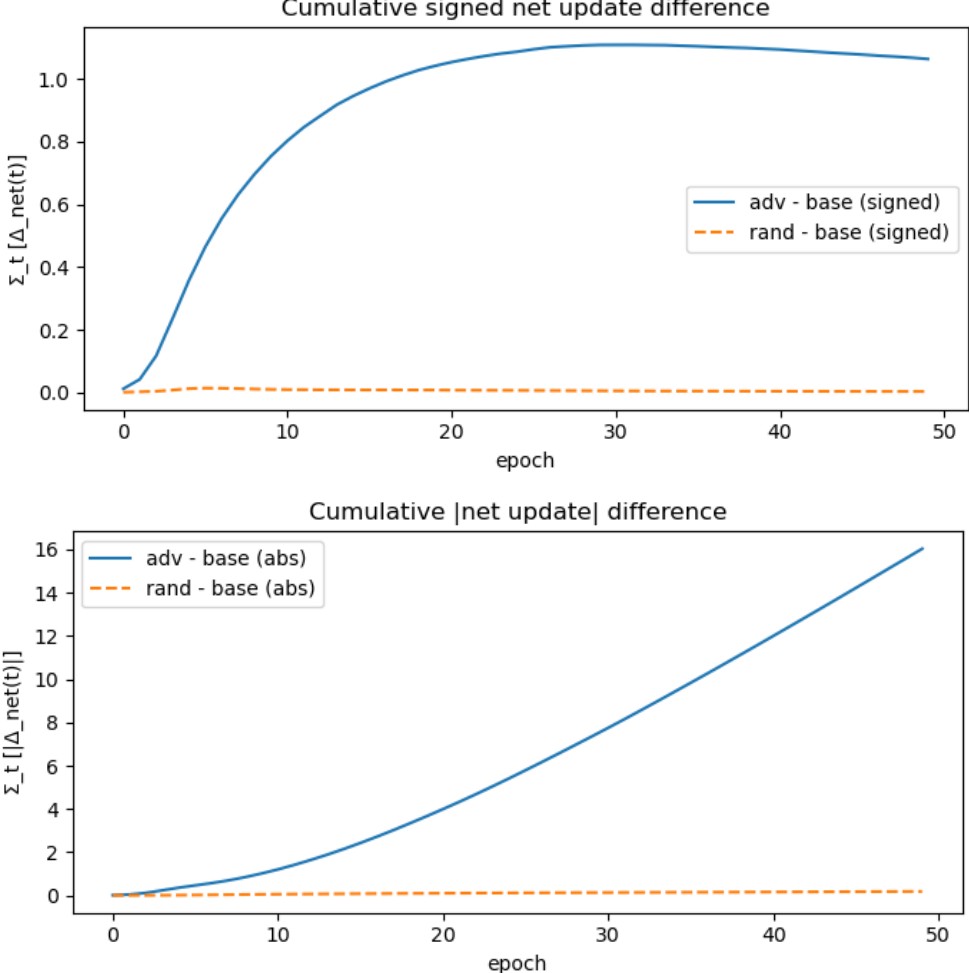

Figure 13: Top: Per-epoch cumulative loss contributions from adversarial or random perturbations. Bottom: As above, but measuring cumulative absolute sum of loss contributions. Adversarial signed contributions level out, while absolute contributions stay large, implying a cancellation effect.

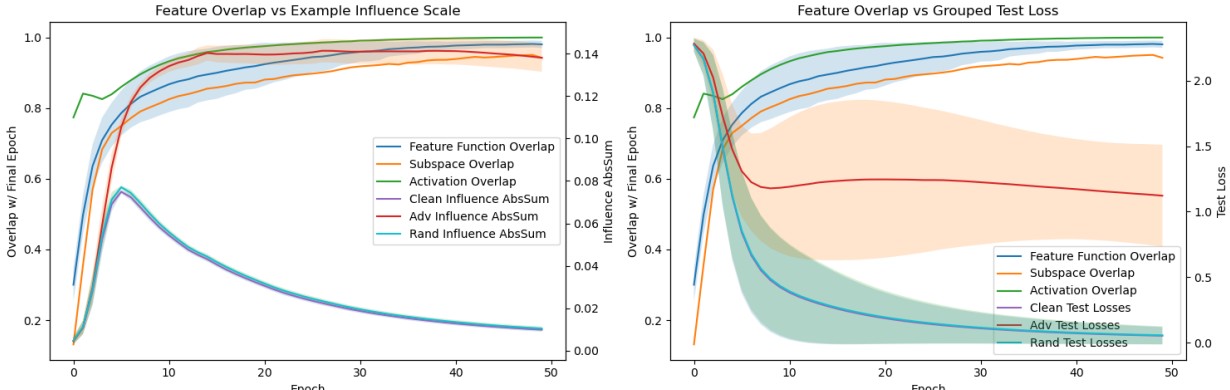

Figure 14: Kernel-Regime metrics (blue/orange/green lines, left y-axes) Left) vs Absolute Influence Sum (purple/red/cyan lines, right y-axis; Right) vs grouped example loss (purple/red/cyan lines, right y-axis). They show an alignment, with all 3 metrics rapidly growing between epochs 5 and 10, with the 'knee' approximately lining up with the peak of the clean and randomly perturbed example influenced absolute sums on the left subplot, and with the knee of the loss curves on the right subplot.

(iii) Activation overlap: $\rho_{\text{act}}(t) = \frac{(\alpha(x,t) \cdot \alpha(x,T))}{\|\alpha(x,t)\| \, \|\alpha(x,T)\|}$, where $\alpha(x,t) \in \mathbb{R}^{H_{final}}$ is the vector of final hidden-layer activations at input $x$ and time $t$, again averaged over training examples. This directly measures whether learned representations have converged.

All three metrics approach 1.0 as the network enters the kernel regime, since fixed representations and loss gradients imply temporal self-consistency.

We plot all three metrics alongside the gross influence and loss curves in Figure 14. All three converge toward 1.0 between epochs 5 and 10. This transition aligns with the peak of clean and random gross influence (after which both decay exponentially) and with the knee of the loss curves (after which clean loss asymptotes while adversarial loss plateaus at an elevated value). The temporal coincidence supports the hypothesis: cancellation becomes effective precisely when representations stabilize.

**Mode decomposition of training influences.** The preceding analysis establishes *when* cancellation occurs (after the kernel-regime transition), but not *how*. Analyzing all $M$ per-sample influences individually is impractical. To identify the low-dimensional structure underlying the cancellation, we decompose the training-set gradient matrix $\Phi_{\text{train}}$. Applying a singular value decomposition $\Phi_{\text{train}} = \mathbf{U}\mathbf{S}\mathbf{V}^{\top}$, we can rewrite the per-step loss change at audit point $x_n$ as

$$\Delta L(\hat{y}_n, t) = -\frac{\eta}{M} \sum_m (\epsilon_m \phi_m) \cdot (\epsilon_n \phi_n)$$
$$= -\frac{\eta}{M} \sum_k (u_k^T \mathbf{1}) \sigma_k (v_k \cdot (\epsilon_n \phi_n))$$

where we sum over modes $k$ of the SVD. To simplify notation, We define $a_k := u_k^{\top} \mathbf{1}$, which measures the net alignment of training examples along the $k$-th left singular direction (positive if more examples push in the same direction, near zero if they cancel). The product $\Lambda_k := a_k \sigma_k$ then captures the overall *drive* of mode $k$: how strongly and coherently the training set collectively pushes learning along direction $v_k$. Finally, $c_{k,n} := v_k \cdot (\epsilon_n \phi_n)$ measures how much audit point $x_n$ projects onto mode $k$. We next examine how the mode coefficients $c_{k,n}$, averaged over adversarial examples, differ from those averaged over clean examples. As shown in Figure 15, only the top modes show significant differences.

Interestingly, the modes appear approximately balanced about $a_k = 0$, suggesting a mechanism: adversarial perturbations may be finely tuned to have significant gradient elements within top modes (e.g. high $c_{k,adv}$),

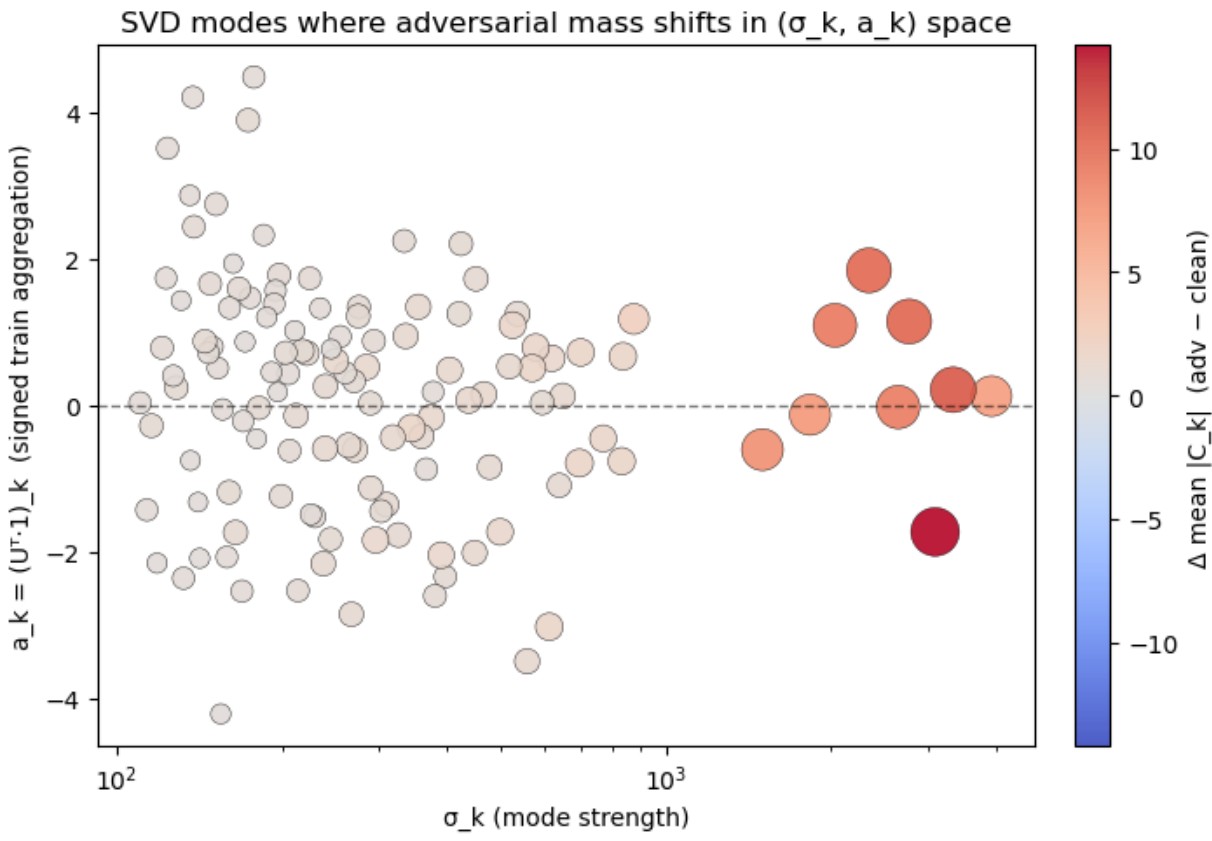

Figure 15: Plot showing the mean change in mode activity between adversarial and clean inputs (size/color), scattered against singular value ($x$-axis) and signed train aggregation ($y$-axis). The difference is concentrated in the top-9 largest singular modes, while using both positive and negative train aggregations.

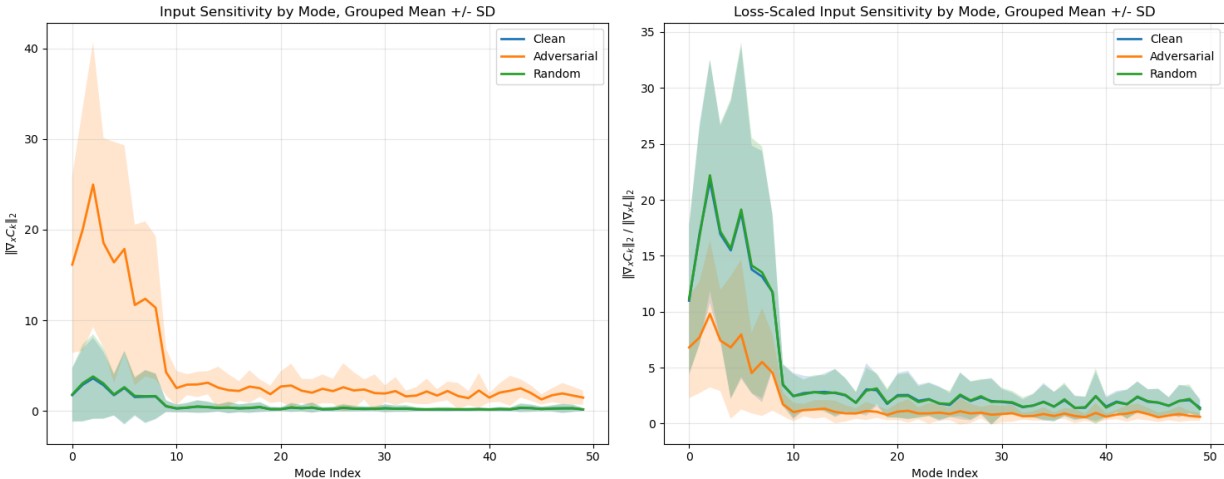

Figure 16: Per-mode input sensitivity (left) and loss-corrected per-mode input sensitivity (right), mean + standard deviation over the three example types. As $c_{k,n}$ directly scales with loss gradient, the loss-gradient corrected version may be more insightful. Both versions clearly show that the top 10 modes have significantly increased input sensitivity compared to the next 40 modes when we analyze the top-50 modes.

resulting in a large loss gain (relative to the clean training point), while carefully tuning the weighted sum $\Lambda_k c_{k,adv}$ to cancel out so that no learning occurs.

One last aspect remains to be explored. Our previous theory of canceling influences explains how adversarial examples can be made to have significant overlap with relevant features, and yet still experience no learning (i.e. loss decreases) over training. However, this does not guarantee that these examples will be 'adversarial' in the sense that they are only a small perturbation (in input space) from a given clean example. Our previous results have shown that adversarial examples primarily differ in the top 10 SVD modes (i.e. $c_{k,n}$ for $k < 10$). This is expected, as changes in these coefficients have an increased effect on loss due to the larger singular values $\sigma_k$. However, these $c_{k,n}$ perturbations are made in *parameter* space, rather than input space, as increasing $c_{k,n}$ requires perturbing the gradient in the $v_k$ direction. However, if $\|\nabla_x c_{k,n}\|$ is too small, a large input perturbation is required to make an adversarial example, potentially conflicting with our proposed method. We empirically measure the final input perturbation sensitivity for the top-50 modes, as shown in Figure 16, finding that the top-10 modes are the most sensitive to input perturbations, further explaining why we see adversarial perturbations primarily in these modes.

This result seems sensible, as we expect that the top modes correspond to useful learned features, with the increased input sensitivity corresponding to useful directions rather than due to random chance. To confirm this, we examine the mean sensitivity of the top-10 directions compared to those of modes 11-50 over training time, shown in Figure 17. As expected, sensitivities start low for both groups, but increase over learning primarily for the top-10 directions. Figure 17 (right) shows the same sensitivities scaled by the inverse of the magnitude of the per-example loss gradient. This normalization accounts for the fact that adversarial examples have systematically higher loss, which inflates their raw input sensitivities via the chain rule. After normalization, the ordering between groups reverses — indicating that the elevated raw sensitivity of adversarial examples is primarily driven by their higher loss rather than by a fundamentally different relationship between inputs and mode coefficients. The top-10 versus remaining mode gap persists under both measures, confirming that the concentration in top modes is a structural property of the learned representations rather than an artifact of loss magnitude.

We have now confirmed that there is significant $|\nabla_x c_{k,n}|$ among the most important modes, allowing for efficient adversarial perturbations. However, it is unclear which factors are most important in practice: the input sensitivity $|\nabla_x c_{k,n}|$, the actual coefficient shift $\Delta c_{k,n}$, or the mode drive $\Lambda_k$. We track each of these factors across the top-$K$ modes for clean, randomly perturbed, and adversarial examples. Individually, each factor

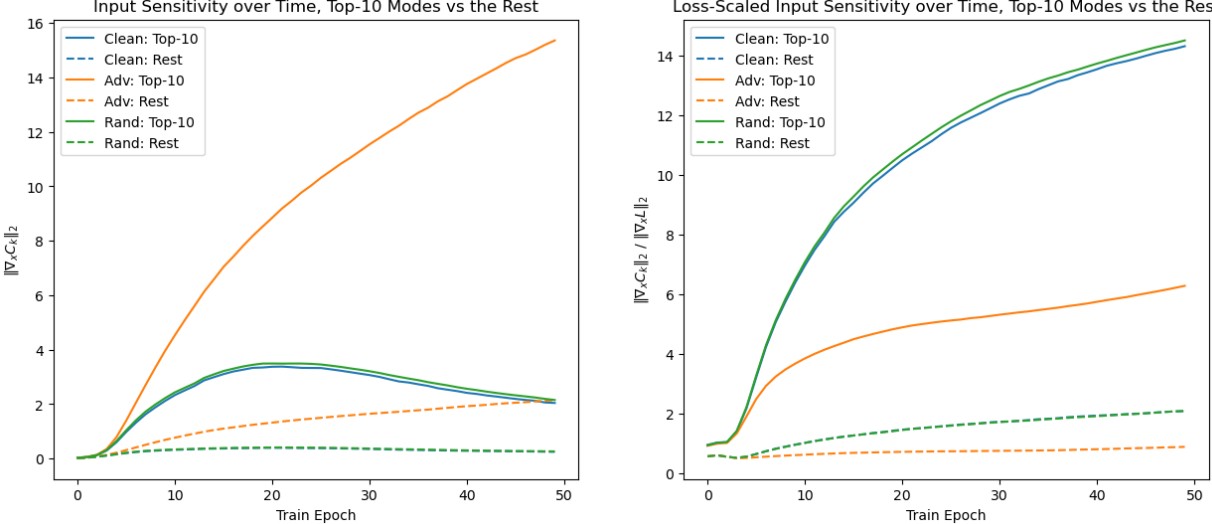

Figure 17: Per-mode input sensitivity (left) and loss-corrected per-mode input sensitivity (right), per-epoch means for top-10 modes vs the next 40. Both versions clearly show that the top 10 modes input sensitivity pulls away from the remaining analyzed modes throughout learning time, e.g. the increased top-10 mode sensitivity is learned throughout training time. Note loss-gradient-scaling reduces the effects of the adversarial (high loss) case.

is only moderately concentrated (participation ratios 6–10; the participation ratio $\text{PR} = (\sum_k w_k)^2 / \sum_k w_k^2$ measures the effective number of components contributing to a distribution of weights $w_k$; it equals 1 if a single component dominates and $K$ if all $K$ components contribute equally), but because they are strongly correlated, adversarial perturbations preferentially shift the highest-energy modes ($|\Delta c_{k,n}|$ vs $\Lambda_k$: correlation $\rho = 0.80$), which are also the most input-sensitive ($\Lambda_k$ vs $|\nabla_x c_{k,n}|$: $\rho = 0.72$), their product yields highly concentrated loss contributions ($\text{PR} = 2.4$, top 5 modes explain 84% of $\Delta L$). Random perturbations shift the same set of modes as adversarial perturbations (consistent with the top modes being the most input-sensitive) but with $\sim 80\times$ lower magnitude distributed uniformly across modes. This indicates that the key property of adversarial perturbations is not *which* modes they affect (both types affect the same ones), but the *magnitude and sign structure* of the shifts: adversarial perturbations produce large, carefully balanced mode coefficient changes whose weighted contributions cancel, whereas random perturbations are simply too weak to meaningfully affect loss dynamics.

The above experiments seem to support our hypothesis that adversarial examples exploit mode cancellation, so our next step is to directly measure this cancellation. However, this requires slightly more care as the above theoretical derivation assumed a full batch GD. For minibatch SGD, we instead have to account for the various gradients induced by different sets of data across minibatches, which may require more modes than a single minibatch. In order to account for this, we use an online SVD update (see Appendix A.8 for details and diagnostics).

Given our SVD-based analysis, we can compute the per-mode contributions across the entire learning trajectory. This allows us to measure the degree of cancellation and how it evolves with the number of retained modes. Concretely, we define a *cancellation ratio* analogous to the net/gross influence distinction introduced earlier, but restricted to the top $K$ SVD modes:

$$\text{CR}(K,t) = \frac{\left| \sum_{k=1}^{K} \Lambda_k \, c_{k,n}(t) \right|}{\sum_{k=1}^{K} |\Lambda_k \, c_{k,n}(t)|},$$

averaged over examples in each group. $\text{CR} = 1$ indicates no cancellation (all mode contributions share the same sign), while $\text{CR} \approx 0$ indicates near-perfect cancellation among modes. As shown in Figure 18, adver-

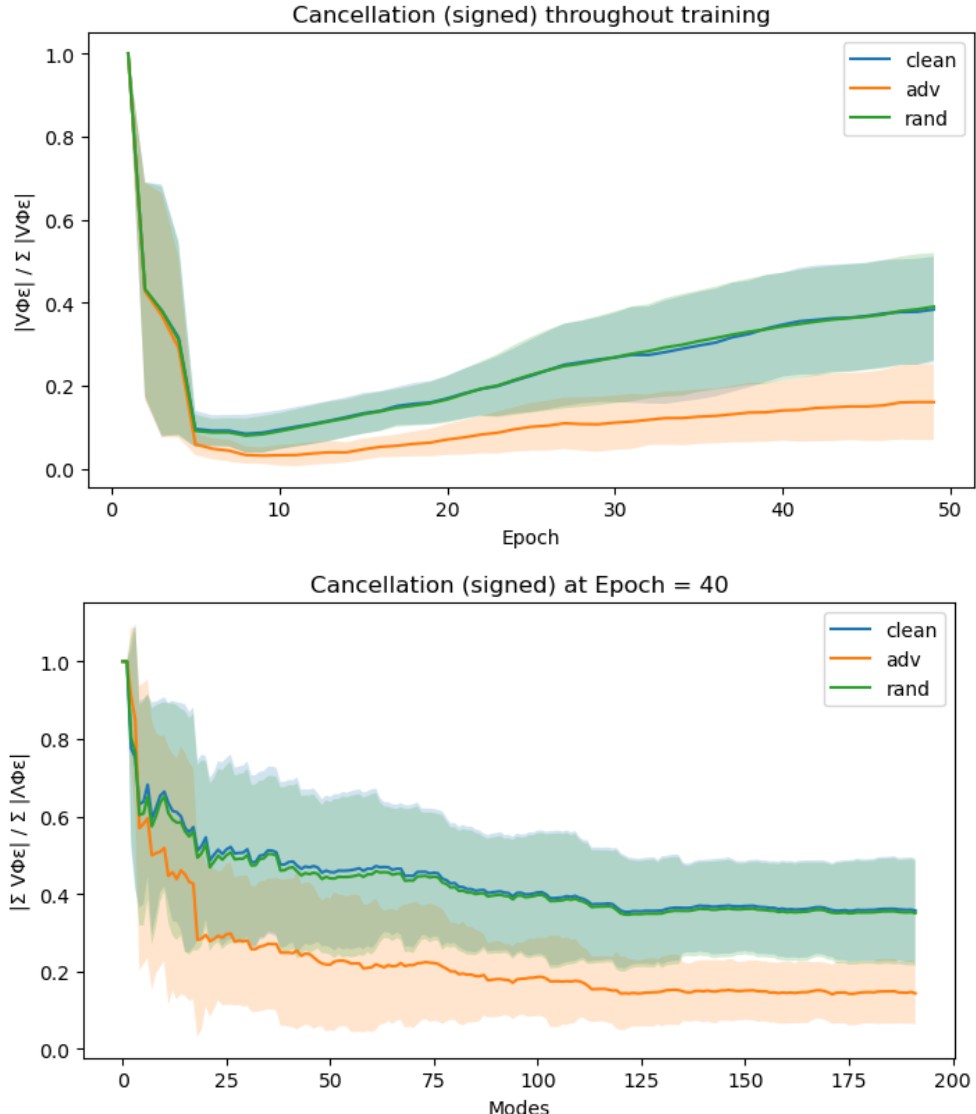

Figure 18: Per-type aggregated (mean ± sd) cancellation metric (lower is more cancellation), shown over learning time for all modes (top), or at fixed time (epoch = 40) across modes (bottom). Adversarial examples demonstrate significantly higher cancellation in the kernel regime portion of training, with all types requiring approximately 125 modes to saturate.

sarial examples show noticeably more cancellation, and need approximately 125 modes to show maximum cancellation.

Our cancellation theory makes a testable prediction: if adversarial examples succeed by balancing mode contributions so that net learning cancels, then a small input perturbation that deliberately *breaks* this balance via shifting mode coefficients in the loss-decreasing direction should restore more typical learning dynamics and reduce loss. This is not proposed as a practical defense against adversarial attacks, but as a direct experimental test of the cancellation mechanism. We call this a *mode-aware correction* and construct it using only the final-time SVD of $\Phi_{\text{train}}$ (details in Algorithm 2). Notably, this procedure does *not* require access to true labels or optimization against the loss: following the label-free branch of Algorithm 2, we use the model's own predicted label $y^* = \arg\max_d f(x_{\text{adv}}; \theta)_d$ as a pseudo-label for computing mode coefficients, and the perturbation objective targets mode-coefficient shifts rather than loss reduction. We obtain parameter-

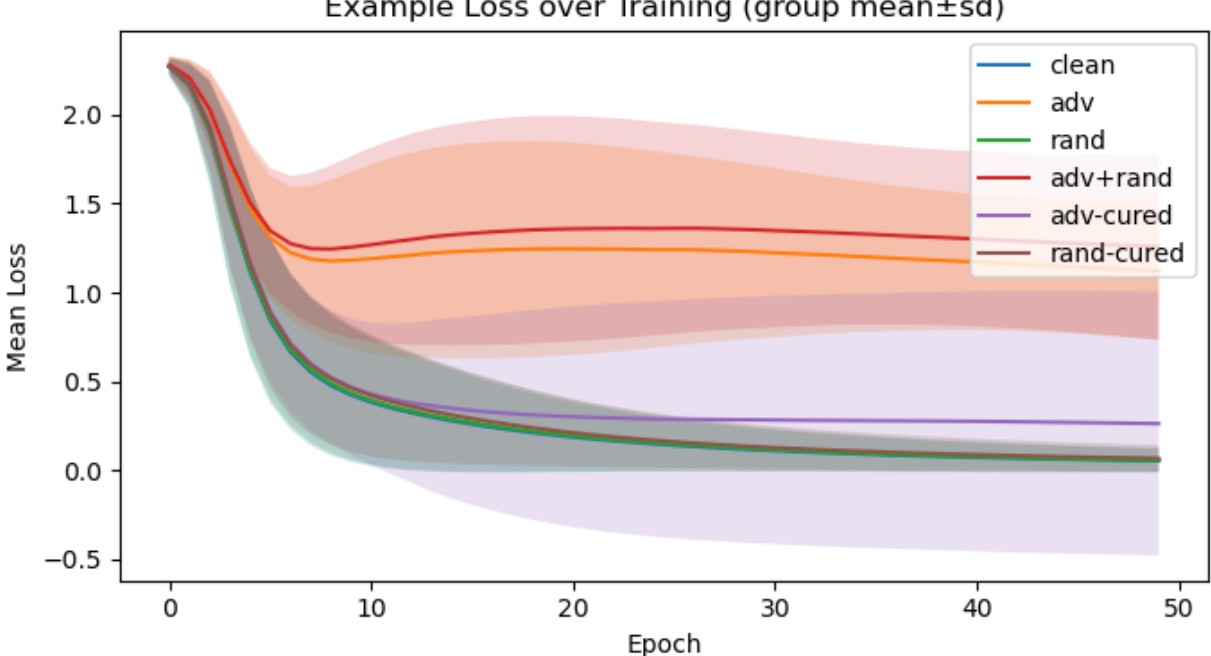

Figure 19: Grouped (mean±sd) example losses throughout the training process. Only adversarial and adversarial + random perturbation groups show elevated final loss, while the mode-aware corrected examples are cured, almost returning to the performance of the clean baseline.

space modes $v_k$ and their audit-point-independent drives $\lambda_k$, and for a given input $x_{\text{adv}}$, we find a bounded perturbation $\delta$ (via projected gradient descent in input space) that pushes the example's mode coefficients $c_k(x) = v_k^\top \nabla_\theta L(\hat{y}(x), y)$ by a fixed amount $-\tau \operatorname{sign}(\lambda_k)$ in the top $K$ modes, while penalizing alignment with the original adversarial perturbation to discourage simply "undoing" it. We apply this procedure to both adversarially perturbed and randomly perturbed training examples, and include a control condition consisting of fresh random perturbations applied to adversarial inputs (scaled to match the balance-breaking magnitude). We then evaluate these three sets of inputs throughout training time. As shown in Figure 19, the mode-aware correction essentially "cures" adversarial example loss while leaving random example loss essentially unchanged, whereas a generic random perturbation does not yield comparable improvements.

Notably, this mode-aware correction is not just undoing the original (adversarial or random) perturbation. The cosine similarity between the original (adversarial) and mode-aware correction directions has a mean of $-0.3$, and looking at examples (such as the one shown in Figure 20) shows that the final combined effect of adversarial and mode aware correction perturbations is not just the original input.

In summary, the new explainability afforded by our PLGK-based auditing allowed us to develop a novel, mode-based theory of adversarial attacks. We then showed that across 4 different ($L_\infty$ based) adversarial attacks on MNIST, our theory's testable predictions are confirmed. Most importantly, a mode-aware (and without access to ground-truth labels) perturbation based on our theory 'cures' adversarial examples, largely removing their adversarial effect.

*Remark* 6.2 (Interpreting the mode-aware correction). The success of the mode-aware correction provides direct evidence for the cancellation mechanism: the correction is constructed entirely from the SVD mode structure of $Phi_{\text{train}}$ (Algorithm 2), targeting mode-coefficient shifts rather than loss reduction, and without access to clean-example locations. That it restores performance (87.5% vs 0.0% for uncorrected adversarials; Section 6.3) while random perturbations of matched magnitude do not (only 15.6%, as even random perturbations can sometimes break adversarial balance) confirms that the *direction* of the correction (breaking the balanced mode structure) matters, not merely the act of perturbing the input. We note that the correction

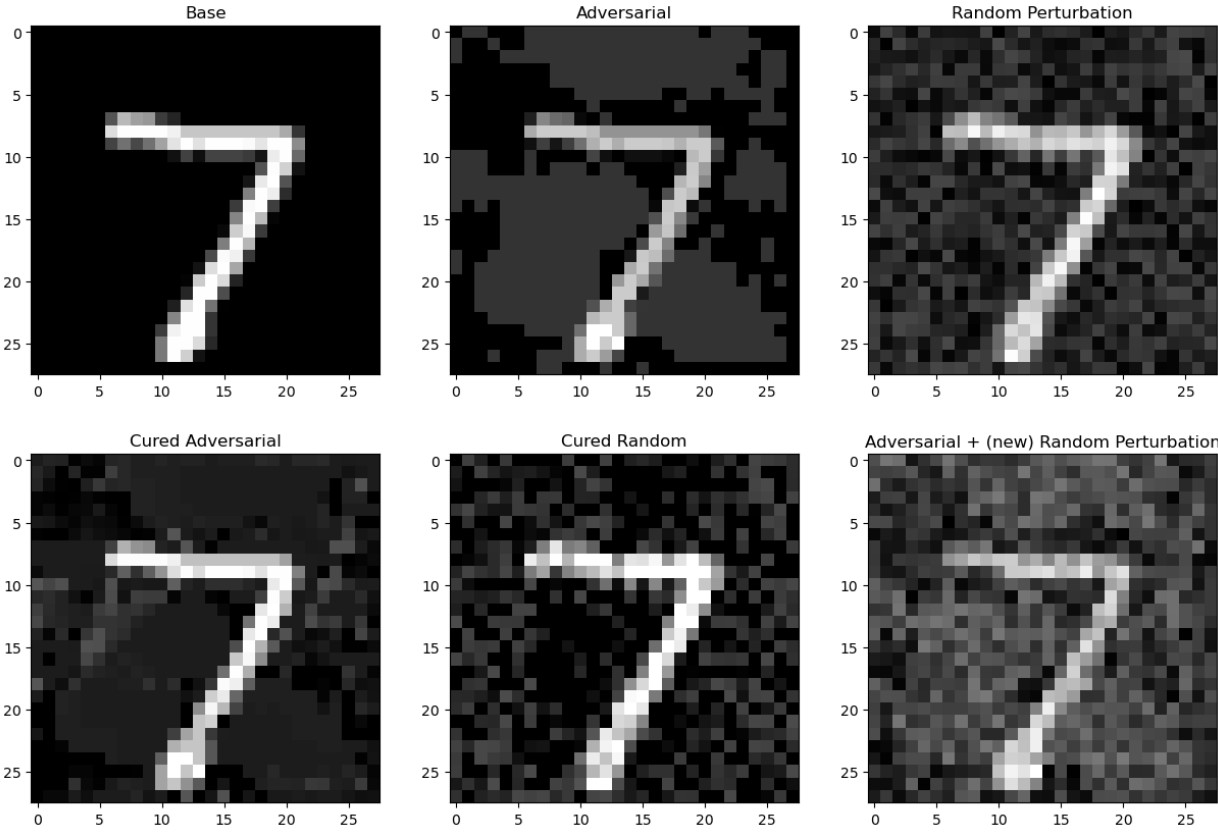

Figure 20: Example of various perturbations from a single clean example. Notably, the cured adversarial perturbation does not merely 'undo' the adversarial perturbation.

is not simply a reversal of the adversarial perturbation (mean cosine similarity between the two directions is only $-0.3$). It could be the case that disrupting mode cancellation also implicitly moves inputs toward some ground truth manifold which true data lies on, but we leave this as an interesting open question that we do not attempt to resolve here; disentangling these potentially related effects would require access to the true data manifold or carefully constructed synthetic settings, which we leave to future work.

| Data Type | Accuracy |
|---|---|
| Clean | 100% |
| Random | 100% |
| Adversarial | 0% |
| Cured Adversarial | 87.5% |
| Cured Random | 100% |
| Adversarial + Random Perturbation | 15.6% |

Table 2: Final test accuracy on various data types. The same model is used in all cases. We start from a baseline of clean examples that are all correctly classified. Random perturbations do not affect performance, while adversarial ones successfully fool the trained classifier. The 'curing' perturbation mostly restores performance while not harming randomly perturbed examples' performance. Random perturbations (rather than curing perturbations) do not show this effect.

### 6.3.1 Connection to Perceptual Manifolds

Recently, Salvatore *et al.* (Salvatore et al., 2026) independently characterized adversarial vulnerability from a purely input-space perspective. They define a network's *perceptual manifold* (PM) for a class $c$ as the set of all inputs $x \in [0,1]^D$ that the network confidently assigns to $c$, and claim that the dimensionality of these PMs is orders of magnitude higher than that of the natural image manifold (e.g.~3060 of 3072 ambient dimensions for CIFAR-10). They argue that this exponential mismatch between machine and human perceptual dimensionality is the fundamental geometric origin of adversarial examples: because a network's PM fills nearly all of input space, any input is close to any class boundary.

Our framework provides a natural explanation for this finding. The total training influence on the loss at $x_n$ at time $t$ is

$$\Delta L(x_n, t) = -\frac{\eta}{M} \, \Phi_{\text{train}}(t) \, \epsilon_n(t) \phi_n(t).$$

Decomposing $\Phi_{\text{train}} = \mathbf{U}\mathbf{S}\mathbf{V}^\top$ into $K$ (truncated) effective modes, this becomes

$$\Delta L(x_n, t) = -\frac{\eta}{M} \sum_{k=1}^{K} \Lambda_k \, c_{k,n},$$

Now consider an audit input perturbation $x \to x + \delta$. Since the mode directions $v_k$ are derived from the training-set SVD and are therefore fixed with respect to the audit input, the mode coefficient shifts (to first order) by

$$\Delta c_{k,n} \approx (\nabla_x c_{k,n})^\top \delta, \qquad \nabla_x c_k = J_x^\top v_k \in \mathbb{R}^D,$$

where $J_x = \nabla_x \phi_x \in \mathbb{R}^{P \times D}$ is the Jacobian mapping input perturbations into the loss-gradient feature space. The columns $\{\nabla_x c_{k,n}\}_{k=1}^K$ thus define an *active subspace* in input space: the set of directions along which input perturbations can affect the network's training-time learning dynamics. Perturbations orthogonal to this subspace–the *nullspace* of $[\nabla_x c_1, \ldots, \nabla_x c_K]$–leave the mode coefficients (and hence the loss dynamics) essentially unchanged.

This provides a reinterpretation of the high-dimensional PMs observed by Salvatore *et al.* If only $K \ll D$ modes carry significant training influence (as our experiments suggest, with $K \le 125$ for MNIST), then the nullspace has dimension of at least $D-K$ (by rank-nullity; it is exactly $D-K$ if the mode gradients are linearly independent, and potentially larger otherwise), and the network is approximately invariant to perturbations along these directions. Any input validly classified as class C along the $K$ active directions will continue to be classified as C regardless of its content in the remaining $D - K$ dimensions. The PM is therefore not high-dimensional because the network has learned an *exponentially rich* representation of a class concept, but rather because it has learned an *exponentially sparse* one: the unused dimensions inflate the PM as a free nullspace. This perspective also explains why Salvatore *et al.* find that the PM dimensionality is statistically indistinguishable between semantically meaningful prompts and gibberish controls in CLIP: the nullspace is a property of the network's learned feature structure, not of the concept being represented, so all classes (including nonsensical ones) are inflated by the same nullspace. Furthermore, this view offers a mechanistic account of why adversarial training reduces PM dimensionality: by requiring correct classification within $\varepsilon$-balls around training points, adversarial training forces the network to become sensitive to more input-space directions, effectively increasing the rank of the active subspace and shrinking the nullspace.

We test this last point, training an adversarially trained MNIST network using PGD-AT (Madry et al., 2017), reaching similar test accuracy and increasing robust accuracy against PGD attacks to 36%. We find that the resulting $\Psi_{train}$ decomposition is substantially less concentrated, requiring more modes to capture similar amounts of energy, as shown in Section 6.3.1. Comparing adversarial attacks carried out on the adversarially-trained network to those on the base network reveals the adversarial-to-clean energy ratio drops from $\approx$120x to $\approx$53x, implying the adversarial training makes the adversarial perturbations less effective at amplifying gradients.

|           | Standard | PGD-AT | Ratio        |
|-----------|----------|--------|--------------|
| $K@90\%$  | 12       | 28     | $2.3\times$  |
| $K@95\%$  | 32       | 60     | $1.9\times$  |
| $K@99\%$  | 109      | 143    | $1.3\times$  |

Table 3: Energy concentration of $\tilde{\Psi}_{train}$ for standard and adversarially trained (PGD-AT) networks. $K@X\%$ denotes the number of modes required to capture $X\%$ of total gradient energy.

## 7 PLGK Scaling

This work demonstrates very high fidelity auditing on small, simple CNN models. We deliberately restricted ourselves to this setting in order to validate our theory and demonstrate near-perfect reconstruction abilities. However, this auditing precision comes at increased computational cost (relative to baseline training), rendering it infeasible on larger, state of the art models and training runs. Although outside the scope of this work, we want to present a preliminary sketch on how PLGK-based tools could be expanded for use in more computationally expensive settings.

First, although our experiments were restricted to simple CNNs, PLGK is theoretically architecture-agnostic (the exception is BatchNorm, which can be replaced with LayerNorm or GroupNorm). We confirm (See Appendix appendix A.18.2) that the PLGK based auditing does function out of the box on transformer-based architectures. Next, there are several methods to either increase accuracy or decrease cost, including using super-sampling or sub-sampling (Appendix appendix A.2.7), auditing data in groups rather than individually (Appendix appendix A.1.2), and partial audits (Appendix appendix A.1.3). Finally, the 'exact reconstruction' requirement can be relaxed; although necessary to show that the PLGK theory is valid, downstream usefulness instead depends on the relative distribution of component influences. We find that a simple (un-normalized) IPS can recover audit-found influence patterns with high (mean .94 correlation across training trajectory) fidelity, and that IPS-based techniques can scale to ImageNet (see Appendix appendix A.18.1 for details).

## 8 Conclusions

In this work, we show that the change in loss over training for GD-trained networks decomposes naturally into evaluations of a path-integrated, loss-weighted Mercer kernel (the PLGK) defined over data points. We extend more classical NTK ideas by applying them to realistic training scenarios (finite width, finite time) rather than infinite-width, linearized limits. Using this framework, we develop two main tools. The first is IPS, which measures an overall similarity between any two data points using the training-path kernel features. We show that IPS (using labels on a held-out audit set) reproduces expected qualitative behaviors and is competitive as a supervised predictor of test loss and accuracy. The second is auditing, which produces a high-fidelity decomposition of the per-component influences that affect a network's final outputs. We validate that auditing achieves near-exact reconstruction on our test cases, with large accuracy improvements over TracIn and TRAK under realistic optimizers[3] (Appendices A.11 and A.12).

Finally, as a case study we use our theory (and the resulting IPS/audit tools) to analyze adversarial examples in small CNNs (SVHN re-creation in Appendix), finding a novel result: adversarial examples show increased influence magnitudes (concentrated in later training and early fully connected layers), are highly concentrated in a small number of high-sensitivity modes, and approximately balance in those modes (resulting in little net learning) once the model has entered a late-training kernel regime. As a capstone, we show that a small "curing" perturbation that breaks this adversarial balance largely restores performance. The mode-cancellation theory is consistent across two architecture–dataset pairs and four $L_\infty$ gradient-based attacks, but our experiments were restricted to small CNNs. We consider our mode-based adversarial findings one adversarial mechanism, with future work required to discover how well it generalizes to e.g. transformers.

---

[3]Note TRAK is optimized for counterfactual estimation rather than reconstruction

This paper uses small-scale experiments on MNIST and SVHN (Appendix A.16) to allow dense logging and complete, maximum-fidelity audits. Our main limitation is scale. Auditing and IPS are $O(T \times M \times N \times P)$ in the dense setting, which is prohibitive for large models. We identify cost-accuracy tradeoffs (Appendix A.2.7) but leave efficient implementations to future work. We also leave several tool extensions for future work, including developing IPS variants that operate without test-set labels and more directly integrating mode-based decompositions into the audit framework. See Appendex Appendix A.18 for further discussions on scaling up the PLGK.

More broadly, our results highlight the value of viewing modern training procedures through a kernel lens. The path-integrated loss-gradient kernel provides a single object from which we can derive high-fidelity influence audits, similarity measures such as IPS, and mode-based analyses that explain phenomena like adversarial vulnerability and late-stage "lazy" behavior. This perspective helps bridge classical kernel/NTK theory and the practice of training finite-width networks with realistic optimizers, and turns the training run itself into something that can be queried and decomposed rather than treated as an opaque process that leaves only final weights behind. We view this as a template for future tools: treating gradient-descent training as a data-dependent kernel machine not only yields new theoretical insight, but also produces practical diagnostics for understanding which examples, layers, and modes matter for a model's behavior in high-stakes settings.

## Broader Impact Statement

This work is primarily concerned with interpretability and auditing: the tools we develop are intended to make trained models more transparent by attributing their behavior to specific training examples, training phases, and parameter modes, which we expect to be broadly useful for debugging, accountability, and understanding model failures. The adversarial analysis carries a dual-use consideration: a clearer mechanistic account of why adversarial examples succeed can inform stronger attacks as readily as stronger defenses, and our "curing" perturbation is presented as a test of the proposed mechanism rather than as a deployable defense. We also note that per-sample influence attribution, by its nature, surfaces information about the role of individual training points, which may warrant care in privacy-sensitive settings. We believe the net effect of improved auditing and interpretability is positive, but we flag these considerations for readers building on the method.

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

# A Appendix

## A.1 Auditing Capabilities

### A.1.1 Auditing by Parameters, Training Time

The base auditing method provides a linear decomposition of network output changes with respect to the training data. However, other decompositions are possible as well. The two most obvious ones are a time-based and parameter-based decomposition. A full decomposition over all three elements would result in a $\widetilde{P}^*_{mntp}$, where $m$ indexes the train data, $t$ the time point, $p$ the parameter, and $n$ the audit. Using the same output function $l$ (representing either loss or prediction outputs) as previously defined,

$$\widetilde{P}^*_{mntp} = -\frac{\eta}{M} \left( \nabla_{\theta_p} L(\hat{y}_m(\theta_t))^\top \left[ \frac{1}{2} \nabla_{\theta_p} l(x_n, t) + \frac{1}{2} \nabla_{\theta_p} l(x_n, t+1) \right] \right)$$

Obviously, the full audit tensor is unwieldy and difficult to compute. One solution is to sum over dimensions that are not of interest (e.g. summing over times points $t$ and parameters $p$ recovers the original $\widetilde{P}^*_{mn}$ matrix from previously).

### A.1.2 Grouped Auditing

Another possibility is to 'group' over parameters of interest. For example, rather than audit every parameter individually, it might make sense to group them into parameter sets that represent a behavior or system of interest (e.g. per attention head, or per layer). Given a set of 'grouped' parameters for the train set, audit set, training time, and parameters (where $\tilde{x}_i$ represents the ith group of type $x$ e.g. $\tilde{p}_2$ is the 2nd parameter group): we can generate a grouped audit:

$$\widetilde{P}^*_{\tilde{m}_i \tilde{t}_j \tilde{p}_k \tilde{n}_l} = -\frac{\eta}{M} \sum_{t \in \tilde{t}_j} \left( \nabla_{\theta_{\tilde{p}_k}} L(\hat{y}_{\tilde{m}_i}(\theta_t))^\top \left[ \frac{1}{2} \nabla_{\theta_{\tilde{p}_k}} l(x_{\tilde{n}_l}, t) + \frac{1}{2} \nabla_{\theta_{\tilde{p}_k}} l(x_{\tilde{n}_l}, t+1) \right] \right)$$

where $l(x_{\tilde{m}_i}, t)$ is a vector of $l(x_m, t)$ for all $m \in \tilde{m}_i$ and $\theta_{\tilde{p}_k}$ is a vector of $\theta_p$ for all $p \in \tilde{p}_k$. Grouping (especially across multiple dimensions) can reduce the complexity and cost of computing $\widetilde{\mathbf{P}}^*$. However, the downside is that grouping hides within-group information - for example, if grouping all parameters within a layer, it is impossible to determine which sub-parameters of the layer were most important (without doing a new analysis), and if grouping over training data, reveals that one group is more impactful than another, it is impossible to tell which particular training data within the dataset is the cause.

### A.1.3 Partial Auditing

Rather than the full set of training data, training time, or parameters, a partial set (or partial grouped set) can instead be used. This eliminates the 'completeness' principle, as we can no longer sum over entire dimensions to recover the change in network output across the audit set, rendering the MLE metric unusable. However, relative influences between audit elements (or groups of elements) may still be relevant, and LC can still be used as a metric to determine audit quality and whether supersampling or subsampling is appropriate. Partial auditing can dramatically reduce costs, and may be a natural choice if only a subset of data is of interest.

### A.1.4 Auditing Targets

Our usage of a generic model output $l(x, t)$ allows for changing the audit target. Although loss and outputs are the most obvious things to audit, in principle ANY function of the network can be audited. For example, the sum of outputs could be audited (at reduced dimensionality compared to auditing each output individually), the activations of an intermediate layer, or any other quantity of interest. The only requirement is that $\nabla_\theta l(x, t)$ is computable!

### A.2 Extending Auditing to Training Variants

The auditing and IPS derivations in the main text assume full-batch gradient descent with a fixed scalar learning rate. In practice, modern training pipelines employ minibatch sampling, learning rate schedules, adaptive optimizers, momentum, gradient clipping, weight decay, and/or normalization, which modify the per-step parameter update $\Delta\theta$ relative to the vanilla GD case. In this section, we show how to extend both auditing and IPS to accommodate these techniques.

The two tools require different treatment of these extensions, owing to their different goals. Auditing seeks to faithfully reconstruct the realized per-sample influences of a specific training run. This requires that each parameter update $\Delta\theta_t$ be exactly decomposed into individual training-sample contributions, which means every modification to the gradient must be explicitly tracked and attributed. Extensions to auditing therefore involve modifying the update equations (Definitions 10 and 11) to reflect the true optimizer dynamics.

IPS, by contrast, requires only that the loss-gradient features $\epsilon_m(t)\phi_m(t)$ be evaluated at the parameters $\theta_t$ that the optimizer actually produces. Since IPS operates on the realized path through parameter space rather than on how that path was generated, optimizer-specific details need not be modeled: their effects are already encoded in the evaluated features. The only exceptions arise when an optimizer introduces a per-parameter weighting (e.g., adaptive preconditioning) that should be reflected in the inner product defining the kernel; we address this case explicitly in Appendix A.2.3.

#### A.2.1 Minibatches

In practice, the training set $D_{\mathrm{tr}}$ is partitioned into minibatches. Let $B_t \subset D_{\mathrm{tr}}$ denote the minibatch used at training step $t$, with $|B_t|$ samples. We assume non-overlapping partitions, so that one epoch cycles through all of $D_{\mathrm{tr}}$ exactly once; the extension to sampling with replacement is straightforward.

For auditing, the only modification is that each parameter update is driven solely by the samples in the current minibatch. The update equations (Definitions 10 and 11) are therefore restricted to $m \in B_t$, with the normalizing constant adjusted accordingly. For example, the trapezoidal loss audit (Definition 11) becomes

$$\widetilde{P}'_{mn}(t+1) = \widetilde{P}'_{mn}(t) - \frac{\eta}{|B_t|}\nabla_\theta L(\hat{y}_m(\theta_t))^\top \left[\frac{1}{2}\nabla_\theta L(\hat{y}_n(\theta_t)) + \frac{1}{2}\nabla_\theta L(\hat{y}_n(\theta_{t+1}))\right], \quad m \in B_t, \qquad (24)$$

with $\widetilde{P}'_{mn}(t+1) = \widetilde{P}'_{mn}(t)$ for $m \notin B_t$.

For IPS, minibatch training requires no change to the kernel computation itself: the loss-gradient features $\epsilon_m(t)\phi_m(t)$ are still evaluated at the realized parameters $\theta_t$, and the path feature map (Equation (13)) is extended by concatenating over all minibatch steps rather than full-batch steps.

#### A.2.2 LR Scheduling

Learning rate scheduling replaces the fixed scalar $\eta$ with a time-dependent $\eta(t)$, encompassing common strategies such as linear warmup, step decay, and cosine annealing. For auditing, the only change is to substitute $\eta(t)$ into the update equations. The trapezoidal loss audit (Definition 11) becomes

$$\widetilde{P}'_{mn}(t+1) = \widetilde{P}'_{mn}(t) - \frac{\eta(t)}{|B_t|}\nabla_\theta L(\hat{y}_m(\theta_t))^\top \left[\frac{1}{2}\nabla_\theta L(\hat{y}_n(\theta_t)) + \frac{1}{2}\nabla_\theta L(\hat{y}_n(\theta_{t+1}))\right] \qquad (25)$$

with the prediction audit modified analogously.

For IPS, the learning rate schedule affects the magnitude of each parameter step $\Delta\theta_t$, and thus the magnitude of the per-step loss-gradient features. However, since IPS is cosine-normalized, it is invariant to this scaling and requires no modification.

#### A.2.3 Adaptivity

Adaptive optimizers such as Adam, RMSprop, and AdaGrad apply a time-dependent, per-parameter scaling to the gradient before updating parameters. We model this as a diagonal preconditioner $A_t \in \mathbb{R}^{P \times P}$, so that

the parameter update at step $t$ takes the form

$$\Delta\theta_t = -\frac{\eta(t)}{|B_t|} A_t \sum_{m \in B_t} \nabla_\theta L(\hat{y}_m(\theta_t)). \tag{26}$$

All widely used adaptive optimizers fit this template; they differ only in how $A_t$ is computed from the gradient history. Since $A_t$ depends on the full batch gradient (or its running statistics) rather than on individual samples, it is shared across all $m \in B_t$ and can be stored as a single vector $\in \mathbb{R}^P$

**Auditing**   The preconditioner modifies $\Delta\theta_t$ but does not affect how the audit point's output or loss responds to that change. The audit-side gradient ($\mathrm{d}\hat{y}_n/\mathrm{d}\theta$ or $\mathrm{d}L(\hat{y}_n)/\mathrm{d}\theta$) measures the sensitivity of the audited quantity to parameter perturbations via the chain rule (Proposition 3.1), and is therefore evaluated at the realized $\theta_t$ without preconditioning. Concretely, the trapezoidal loss audit becomes

$$\widetilde{P}'_{mn}(t+1) = \widetilde{P}'_{mn}(t) - \frac{\eta(t)}{|B_t|} \left(A_t \nabla_\theta L(\hat{y}_m(\theta_t))\right)^\top \left[\frac{1}{2}\nabla_\theta L(\hat{y}_n(\theta_t)) + \frac{1}{2}\nabla_\theta L(\hat{y}_n(\theta_{t+1}))\right], \tag{27}$$

for $m \in B_t$, with the prediction audit modified analogously. Note that $A_t$ appears only on the training-side (left) factor of the inner product.

**Worked example: RMSprop**   RMSprop maintains an exponential moving average of squared gradients:

$$v_t = \alpha\, v_{t-1} + (1 - \alpha)\, g_t^2, \qquad g_t = \sum_{m \in B_t} \nabla_\theta L(\hat{y}_m(\theta_t)), \tag{28}$$

where $g_t^2$ denotes the element-wise square. The preconditioner is then $A_t = \mathrm{diag}\big(1/(\sqrt{v_t} + \epsilon)\big)$. Adam extends this by additionally incorporating momentum (Section 9.2.5); the two extensions compose straightforwardly.

**IPS**   Unlike auditing, IPS uses symmetric inner products between loss-gradient features, and the preconditioner should be reflected on both sides. The natural modification is to define a preconditioner-weighted inner product at each step:

$$\mathrm{LGK}^A_{mn}(t) := (\epsilon_m(t)\,\phi_m(t))^\top A_t\, (\epsilon_n(t)\,\phi_n(t)). \tag{29}$$

Since $A_t$ is diagonal with strictly positive entries, this remains a valid inner product and the resulting path-integrated kernel is still PSD. Equivalently, one can absorb the preconditioner into the feature map by defining $\tilde{\Phi}_m(t) = A_t^{1/2}\,\epsilon_m(t)\,\phi_m(t)$, reducing the computation to a standard (unweighted) inner product of the rescaled features. Cosine-normalized IPS then measures gradient correlation in the preconditioner-weighted geometry.

### A.2.4   Gradient Rescaling

Several common training techniques rescale the aggregated gradient before it is applied to the parameters. Gradient clipping, for example, enforces a maximum norm by computing

$$g'_t = c_t\, g_t, \qquad c_t = \min\left(1, \frac{C}{\|g_t\|}\right), \tag{30}$$

where $g_t = \sum_{m \in B_t} \nabla_\theta L(\hat{y}_m(\theta_t))$ is the raw minibatch gradient and $C$ is the clipping threshold. Gradient normalization is the special case $c_t = 1/\|g_t\|$. Since $c_t$ is a scalar that does not depend on any individual sample, it distributes uniformly across per-sample contributions. The modified trapezoidal loss audit is therefore

$$\widetilde{P}'_{mn}(t+1) = \widetilde{P}'_{mn}(t) - \frac{\eta(t)}{|B_t|} \left(c_t\, A_t \nabla_\theta L(\hat{y}_m(\theta_t))\right)^\top \left[\frac{1}{2}\nabla_\theta L(\hat{y}_n(\theta_t)) + \frac{1}{2}\nabla_\theta L(\hat{y}_n(\theta_{t+1}))\right], \tag{31}$$

for $m \in B_t$, where $A_t$ is the adaptive preconditioner from Appendix A.2.3 (or the identity for non-adaptive optimizers). In the case of per-element clipping, the scalar $c_t$ is replaced by a diagonal matrix $C_t$ with entries $c_{t,p} = \mathrm{clip}(g_{t,p})/g_{t,p}$, which composes with $A_t$ in the same way.

For IPS, gradient rescaling only affects how $\theta_t$ is reached and requires no modification, by the same reasoning as in previous sections.

### A.2.5   Weight Decay and Regularization

Decoupled weight decay (as used in AdamW) adds a sample-independent term to the parameter update:

$$\Delta\theta_t = -\frac{\eta(t)}{|B_t|}A_t \sum_{m\in B_t} \nabla_\theta L(\hat{y}_m(\theta_t)) \;-\; \eta(t)\,\lambda\,\theta_t, \tag{32}$$

where $\lambda$ is the decay coefficient. The second term depends only on the current parameters and cannot be decomposed into per-sample contributions without additional machinery. We therefore track it as a separate, sample-independent residual. The audit decomposition becomes

$$l(x_n, T) - l(x_n, 0) = \sum_{m\in D_{\mathrm{tr}}} \widetilde{P}'_{mn} \;+\; R_n, \tag{33}$$

where $\widetilde{P}'_{mn}$ is computed from the per-sample gradient terms exactly as before, and $R_n$ accumulates the weight decay contribution at each step:

$$R_n(t+1) = R_n(t) - \eta(t)\,\lambda\,\theta_t^\top \left[\frac{1}{2}\nabla_\theta l(x_n, t) + \frac{1}{2}\nabla_\theta l(x_n, t+1)\right]. \tag{34}$$

Completeness (Definition 9) is preserved: the sum of per-sample attributions plus $R_n$ exactly recovers the true change in the audited quantity. In practice, $R_n$ tends to be a small fraction of the total change for typical values of $\lambda$. The same treatment applies to $\ell^2$ regularization implemented as part of the loss function, since the $\lambda\theta$ term in the gradient is equally sample-independent regardless of implementation.

(In principle, the weight decay term can be recursively attributed to past training samples by unrolling the parameter trajectory through its gradient history, yielding exact per-sample attribution at $O(M\times P)$ storage cost. We leave efficient implementations of this approach to future work.)

For IPS, weight decay modifies the parameter trajectory but not the form of the loss-gradient features evaluated at the realized $\theta_t$, and therefore requires no modification.

### A.2.6   Momentum

Momentum-based optimizers accumulate an exponentially weighted moving average of past gradients. Using the standard formulation (as in Adam), the momentum buffer satisfies the recurrence

$$m_t = \beta\,m_{t-1} + (1-\beta)\,g_t, \qquad m_0 = 0, \tag{35}$$

where $\beta \in [0, 1)$ is the decay factor and $g_t = \sum_{m\in B_t}\nabla_\theta L(\hat{y}_m(\theta_t))$ is the minibatch gradient at step $t$. The parameter update is then $\Delta\theta_t = -\frac{\eta(t)}{|B_t|}A_t\,m_t$ (with $A_t$ the adaptive preconditioner from Appendix A.2.3, or the identity for non-adaptive optimizers). Classical heavy-ball momentum is a special case of this formulation.

**Per-sample decomposition**   Because the momentum recurrence is linear in past gradients, unrolling it yields

$$m_t = (1-\beta)\sum_{s=0}^{t}\beta^{t-s}\,g_s = (1-\beta)\sum_{s=0}^{t}\beta^{t-s}\sum_{m\in B_s}\nabla_\theta L(\hat{y}_m(\theta_s)), \tag{36}$$

so each training sample's contribution to $m_t$ is an exponentially decayed sum of its gradient at every step where it appeared in a minibatch. Unlike weight decay (Appendix A.2.5), no sample-independent residual term arises: the momentum buffer decomposes exactly into per-sample contributions.

**Auditing**   Substituting the unrolled momentum into the audit update, the trapezoidal loss audit becomes

$$\widetilde{P}'_{mn}(t+1) = \widetilde{P}'_{mn}(t) - \frac{\eta(t)}{|B_t|}\left(A_t\,(1-\beta)\sum_{s=0}^{t}\beta^{t-s}\,\nabla_\theta L(\hat{y}_m(\theta_s))\,\mathbf{1}[m\in B_s]\right)^\top \left[\frac{1}{2}\nabla_\theta L(\hat{y}_n(\theta_t)) + \frac{1}{2}\nabla_\theta L(\hat{y}_n(\theta_{t+1}))\right],$$
$$\tag{37}$$

where $\mathbf{1}[m \in B_s]$ is the indicator for sample $m$ appearing in minibatch $B_s$. In practice, the sum over past steps can be truncated at a window of size $k$: the contribution of a gradient from $k$ steps in the past is attenuated by a factor $\beta^k$, which for typical values (e.g., $\beta = 0.9$, $k = 50$) is below $5 \times 10^{-3}$. A circular buffer storing the last $k$ per-sample minibatch gradients suffices for implementation.

**IPS** Momentum modifies the parameter trajectory but not the loss-gradient features evaluated at the realized $\theta_t$, and therefore requires no modification to the IPS computation.

### A.2.7 Sub- and Super-sampling

Both auditing and IPS can be made more compute efficient at the cost of lowering accuracy, while auditing can also do the opposite, trading off additional compute for every higher fidelity.

In order to reduce costs, IPS and auditing can sub-sample along the training trajectory, updating every $k$ steps rather than every parameter update. As IPS is a dimensionless quantity, this can be done directly. However, auditing requires more care: auditing critically depends on capturing the entire training trajectory. Thus, in order to audit over $k$ individual updates, we do the following:

$$\widetilde{P}'_{mn}(t + k) = \widetilde{P}'_{mn}(t) - \frac{\eta}{M} \left( \sum_{i=1}^{k} \nabla_{\theta_{t+i}} L(\hat{y}_m(\theta))^\top \left[ \frac{1}{2} \nabla_\theta L(\hat{y}_n(\theta)) + \frac{1}{2} \nabla_\theta L(\hat{y}_n(\theta_{t+k})) \right] \right)$$

where we accumulate $k$ steps of training gradients, but only use the endpoints $t$ and $t + k$ to estimate the effects on the audit reconstruction.

We can improve auditing fidelity by using a more accurate estimate of the effects of the training gradients on the audit reconstruction. We do this by using a more accurate multi-point stencil, using intermediate parameters (virtually constructed by linearly interpolation, e.g. $\theta_{t+0.5} = \theta_t + \frac{1}{2}(\theta_{t+1} - \theta_t)$. Thus, a mildly super-sampled audit reconstruction step would consist of

$$\widetilde{P}'_{mn}(t + 1) = \widetilde{P}'_{mn}(t) - \frac{\eta}{M} \left( \nabla_{\theta_t} L(\hat{y}_m(\theta))^\top \left[ \frac{1}{4} \nabla_\theta L(\hat{y}_n(\theta)) + \frac{1}{2} \nabla_\theta L(\hat{y}_n(\theta_{t+.5})) + \frac{1}{4} \nabla_\theta L(\hat{y}_n(\theta_{t+1})) \right] \right)$$

### A.2.8 Ghost Dot Product

The In-Run Data Shapely paper (Wang et al., 2024) utilized a technique called the 'ghost dot product' in order to reduce computational cost. The same technique can be used here, albeit with some caveats.

The ghost dot product technique makes use of the following:

$$\nabla_\theta L(\hat{y}_i(\theta)) = \nabla_{A_i(\theta)} L(\hat{y}_i(\theta)) \nabla_\theta A_i(\theta))$$

for any intermediate activation $A_i(\theta)$. But this means that

$$\nabla_\theta L(\hat{y}_i(\theta)) \cdot \nabla_\theta L(\hat{y}_j(\theta)) = \left( \nabla_{A_i(\theta)} L(\hat{y}_i(\theta)) \nabla_\theta A_i(\theta) \right) \cdot \left( \nabla_{A_j(\theta)} L(\hat{y}_j(\theta)) \nabla_\theta A_j(\theta) \right)$$

$$= \left( \nabla_{A_i(\theta)} L(\hat{y}_i(\theta)) \cdot \nabla_{A_j(\theta)} L(\hat{y}_j(\theta)) \right) * \left( \nabla_\theta A_i(\theta) \cdot \nabla_\theta A_j(\theta) \right)$$

Consider a simple intermediate linear layer $A_k = W x_k + b$ for intermediate input $x_k$ and output $A_k$. Then

$$\nabla_W L(\hat{y}_i(\theta)) \cdot \nabla_W L(\hat{y}_j(\theta)) = \left( \nabla_{A_i(\theta)} L(\hat{y}_i(\theta)) \cdot \nabla_{A_j(\theta)} L(\hat{y}_j(\theta)) \right) * \left( \nabla_W A_i(\theta) \cdot \nabla_W A_j(\theta) \right)$$

$$= \left( \nabla_{A_i(\theta)} L(\hat{y}_i(\theta)) \cdot \nabla_{A_j(\theta)} L(\hat{y}_j(\theta)) \right) * (x_i \cdot x_j)$$

where terms of the form $\nabla_{A_k} L(\hat{y}_k(\theta))$ are already available via the backpropogation of the loss during training.

Although the ghost-dot product method is quite efficient on GD training, it interacts poorly with the auditing method's adaptivity and momentum tracking techniques. In both cases, the reversed order of summation used by the ghost-dot product method means that the tracking techniques introduced above must be repeated per layer, often making the non ghost dot product method superior if adaptivity or momentum are used (particularly if both are used!).

Figure 21: Final test accuracy on a re-trained MNIST task where a percentage of the training data is pruned via various techniques. Most Audit/IPS based pruning strategies do better than the baseline random, while cutting the least influential (via auditing) performs the best.

### A.3 MNIST Task Details

MNIST experiments are carried out on a LeNet5 Network architecture. We use 200 training batches, with a batch size 256. The audit set is one batch of test data. We train for 50 epochs, using the Adam optimizer with lr = 1.5e-5. Final test accuracy is 93% (this is lower than most MNIST cases, because of our relatively low amount of training. Extending to 150 epochs increases this to 97% - we only do 50 epochs as beyond this point training slows done significantly, and we focus on showing numerical accuracy during the fastest changing parts of the training setup; accuracy is secondary in this diagnostic setting).

### A.4 MNIST Data Pruning

Since our kernel-based tools (IPS and Auditing) expose per-sample influences and similarities, a natural use case is to find which data is unimportant and can be safely removed via data pruning. We test a variety of data pruning techniques: Cutting the least influential data (Audit influence, either by sum or abs sum), the most redundant data (IPS similarity clusters, either removal or up-weighting), the most harmful data (Audit high loss influence), and a class-balanced version of the least influential data. We compare against a baseline of cutting random data or class-balanced random data (averaged across 5 seeds). Figure 21 shows the results (except for cutting the most harmful, which was removed due to very low performance hiding the detail of other methods) - essentially all methods considered do better than the random removal baseline, with cutting the least impactful being the best overall, with the class balanced variant showing increased benefits at higher cut percentages.

This capability also lets us identify the most important data. We can compare how impactful our data importance ranking is (relative to random) by pruning the most important (rather than least data), e.g. with the goal of reducing final train performance. We test a few techniques: cutting the most influential data (Audit influence, either by sum or abs sum), the most redundant data, the highest impacts on loss (harmful or helpful), and cutting entire clusters (via IPS). Figure 22 shows the results - all methods considered have

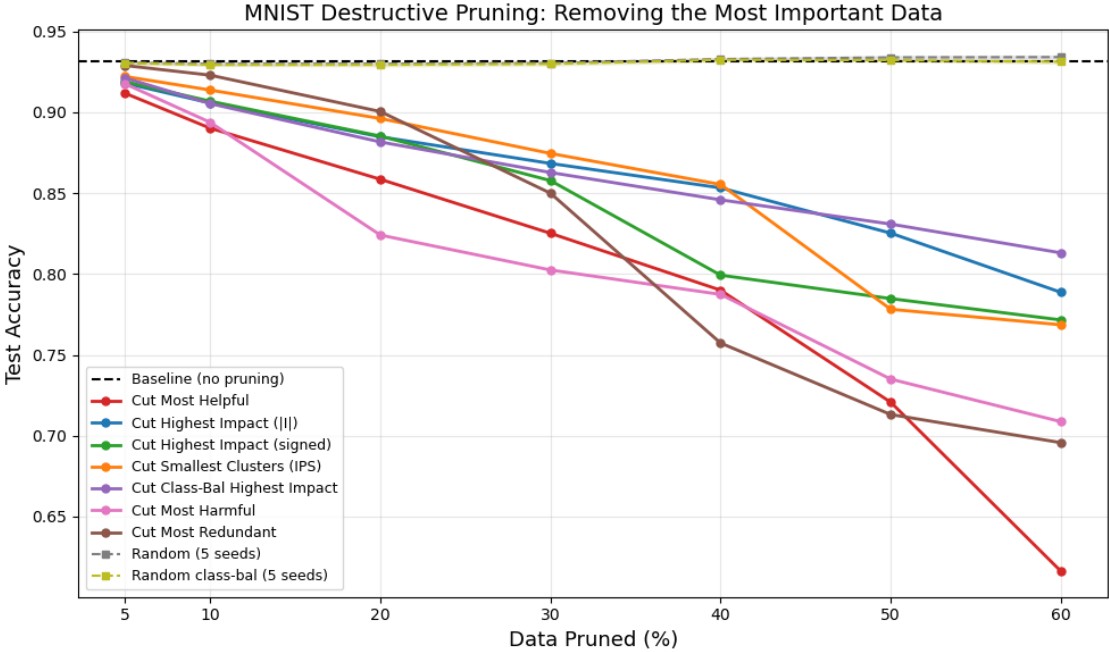

Figure 22: Final test accuracy on a re-trained MNIST task where a percentage of the training data is pruned via various techniques, with the goal of decreasing accuracy via correctly identifying the most helpful data to remove. Most Audit/IPS based pruning strategies do better than the baseline random, while cutting the most harmful or helpful (via auditing) performs the best.

significantly more impact than cutting data at random, with cutting by impact being the least powerful and cutting by helpful/harmful generally being the most.

### A.5 MNIST Auditing Profiling

We profiled running our auditing framework with various options on the MNIST task, compared to a baseline (training only) run. Note that we designed our MNIST task for extremely high audit accuracy and full fidelity capture, so significant time and memory savings are possible when trading off accuracy.

| Method | Time (s) | Memory (GiB) | Final MLE | Final Corr |
|---|---|---|---|---|
| Baseline Train | 226.5 | 0.435 | — | — |
| Audit | 4040.0 | 2.915 | 0.010386 | 0.999998 |
| Audit (Non-trapezoidal) | 3953.5 | 2.859 | 0.090122 | 0.990681 |
| Supersample (2x) | 4162.7 | 3.039 | 0.010401 | 0.999998 |
| Subsample (2x) | 3159.2 | 4.991 | 0.013338 | 0.999976 |
| Subsample (5x) | 2035.2 | 5.084 | 0.021712 | 0.999786 |

Note that supersampling has minimal effect here, as the baseline audit is already approaching the numerical fidelity ceiling.

### A.6 The Gradient Feature Tensor

The mode decomposition introduced in the main text operates on a matrix $\Phi_{\text{train}}(t^*) \in \mathbb{R}^{M \times P}$ formed at a single reference time $t^*$. Here we note that this matrix is one slice of a more fundamental three-index object, and that different contractions and unfoldings of this object recover the audit, IPS, and mode decomposition as special cases.

### A.6.1   Definition

Define the *gradient feature tensor*

$$\mathcal{G} \in \mathbb{R}^{T \times M \times P}, \qquad \mathcal{G}[t, m, p] \;=\; \eta(t)\, \nabla_{\theta_p} L(\hat{y}_m(\theta_t)), \tag{38}$$

where $t \in \{0, \dots, T-1\}$ indexes training steps, $m \in [M]$ indexes training examples, and $p \in [P]$ indexes parameters. Each entry is the $\eta$-weighted per-example loss gradient with respect to a single parameter at a single training step. This is simply the collection of all per-example gradient vectors $\eta(t)\, \epsilon_m(t)\, \phi_m(t)$ computed during training, arranged into a three-dimensional array.

### A.6.2   Existing tools as tensor contractions

Every kernel, influence operator, and mode decomposition introduced in this work corresponds to a specific contraction or unfolding of $\mathcal{G}$.

**Auditing (contraction over $P$).** Contracting the parameter index between a training example $m$ and an audit point $n$ recovers the per-step influence:

$$\mathcal{A}[t, m, n] \;=\; \sum_p \mathcal{G}[t, m, p]\, \nabla_{\theta_p} \ell(x_n, \theta_t), \tag{39}$$

where $\ell$ is the audited quantity (loss or prediction). Summing over $t$ yields the audit matrix $\widetilde{P}_{mn}$ (Definitions 10 and 11), up to trapezoidal corrections on the audit-side gradient.

**PLGK and IPS (contraction over $T$, then $P$).** Concatenating along the time axis produces the path feature map $\Phi_{\mathrm{path}}(x_m) \in \mathbb{R}^{TP}$ (Eq. 13). The PLGK Gram matrix is then the inner product $P'_{mn} = \Phi_{\mathrm{path}}(x_m) \cdot \Phi_{\mathrm{path}}(x_n)$, which contracts both $T$ and $P$. IPS applies cosine normalization to this Gram matrix.

**Mode decomposition (unfolding along $M$).** Fixing $t = t^*$ and forming the matrix $\Phi_{\mathrm{train}}(t^*) = \mathcal{G}[t^*, :, :] \in \mathbb{R}^{M \times P}$ yields the object whose truncated SVD defines the mode directions $v_k$, singular values $s_k$, and derived quantities $(a_k, \lambda_k, c_k)$ used throughout the experiments.

### A.6.3   Alternate unfoldings

Different matricizations of $\mathcal{G}$ yield objects with distinct interpretations.

**Mode-1 unfolding: $T \times (MP)$.** Each row is a flattened snapshot of all training gradients at time $t$. The induced Gram matrix

$$K_{\mathrm{time}}[t, t'] \;=\; \sum_{m, p} \mathcal{G}[t, m, p]\, \mathcal{G}[t', m, p] \tag{40}$$

is a *temporal kernel* measuring the overall similarity of the gradient landscape between training steps $t$ and $t'$. Its eigendecomposition would yield temporal modes–directions in "training-time space" along which the gradient structure varies most. See Appendix A.7 for results of this analysis.

**Mode-3 unfolding: $P \times (TM)$.** Each row is a single parameter's gradient history across all examples and training steps. The Gram matrix $K_{\mathrm{param}} \in \mathbb{R}^{P \times P}$ measures parameter covariance over the training trajectory. At a single time slice, $K_{\mathrm{param}}(t^*) = \Phi_{\mathrm{train}}(t^*)^\top \Phi_{\mathrm{train}}(t^*)$, whose eigenvectors are exactly the mode directions $v_k$ from the main-text SVD. The path-integrated version $K_{\mathrm{param}} = \sum_t \Phi_{\mathrm{train}}(t)^\top \Phi_{\mathrm{train}}(t)$ would capture parameter directions that are important across the entire training trajectory rather than at a single reference time.

**Joint (example, time) unfolding: $(TM) \times P$.** Treating each (time step, example) pair as a separate observation yields a matrix $\mathcal{G}_{(1,2)} \in \mathbb{R}^{TM \times P}$ whose SVD produces mode directions in parameter space that explain variance across *both* training time and training examples simultaneously. Unlike the single-snapshot

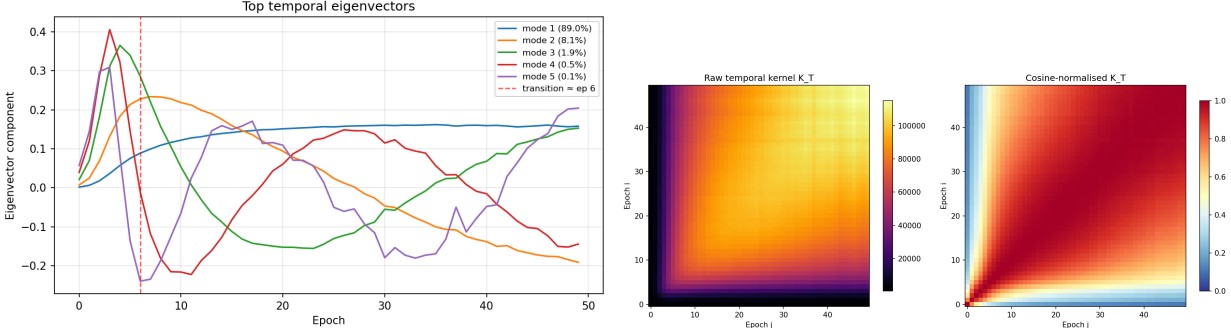

Figure 23: Eigenmode loading over time. Mode 1 tracks the general gradient magnitude. Modes 2-5 track the adaptive to kernel regime transition, which peaks around epoch 6. Mode 1 is distinct from other modes, which are highly correlated, as can be seen in the heatmaps.

modes, these path-integrated modes would naturally weight training phases by their gradient magnitude (early high-loss epochs contribute more) and would distinguish parameter directions that matter throughout training from those relevant only during specific phases.

*Remark* A.1 (Paper View). The tensor perspective reveals that the choice of which index to contract versus preserve determines which "view" of training dynamics one obtains. The experiments in this work primarily use the single-time-slice mode-2 unfolding ($M \times P$ at fixed $t^*$), which is the most computationally tractable and sufficient for the applications demonstrated. The alternate unfoldings–particularly the temporal kernel (Equation (40)) and the path-integrated joint modes–offer natural directions for future investigation into the temporal structure of learning and the interaction between training phases and mode formation.

## A.7 $T \times (MP)$ **Kernel Results**

We build a variant of the kernel discussed above, using a once per epoch snapshot to capture the temporal dimension. We find a very low dimensional temporal structure, essentially organized into two discrete blocks. The very first temporal mode appears to measure 'overall gradient' activity, and monotonically increases over time, nearly saturating later in training. Modes 2-5 appear to quantify the adaptive to kernel regime transition, peaking between modes 5-15. Modes 6+ ($<.4\%$ of total energy) are essentially noise.

We also examine the Participation Ratio (PR), which indicates that the effective dimensionality of the system falls over time (most quickly during the adaptive to kernel transition), as indicated by the eignespectrum steepening.

## A.8 Online Feature Modes (SVD) Estimator

We use Oja's update rule (Oja, 1989) to build an online SVD estimate for the entire batch of features, streaming over minibatches features. This estimate is then frozen, and used as a projector for the next epoch's features in order to extract per-mode contributions. We first confirm that this method is valid, measuring the fraction of energy captured by the projection relative to the baseline (e.g. $\frac{||V\Phi\epsilon||_F}{||\Phi\epsilon||_F}$ for projector $V$) over training time. As this is fixed per epoch, we expect a decline in performance within-epoch, and must ensure that this is small enough to be viable. As shown in Figure 25, by epoch 10 our method has generated a projector set that captures above 99% of the energy throughout all remaining epochs, validating our approach.

## A.9 Adversarial Cure Algorithms

See Algorithms 1 and 2.

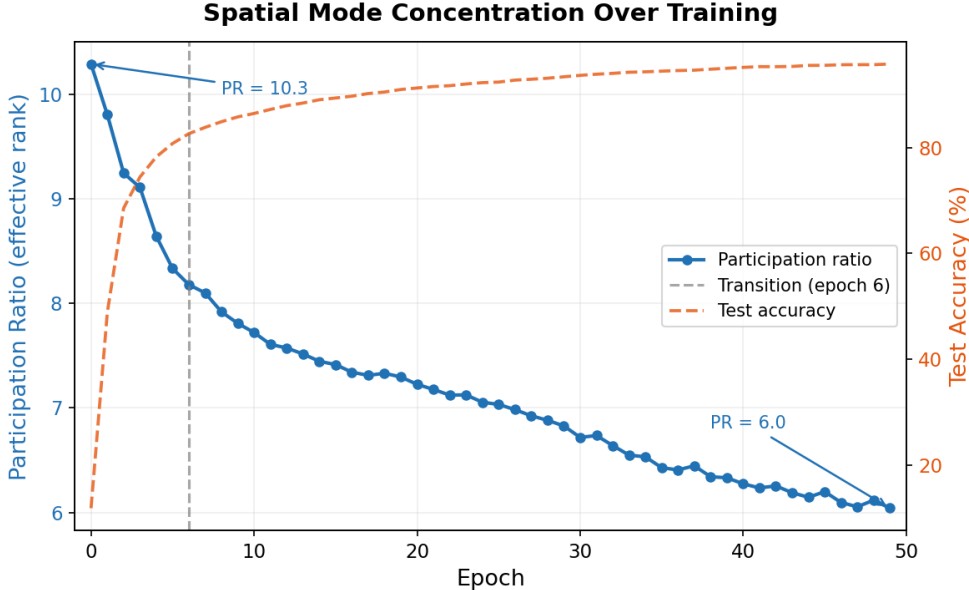

Figure 24: Participation Ratio (PR) over time. This measure of eigenspectrum steepness indicates the effective dimensionality reduces over time, particularly during the adaptive to kernel transition.

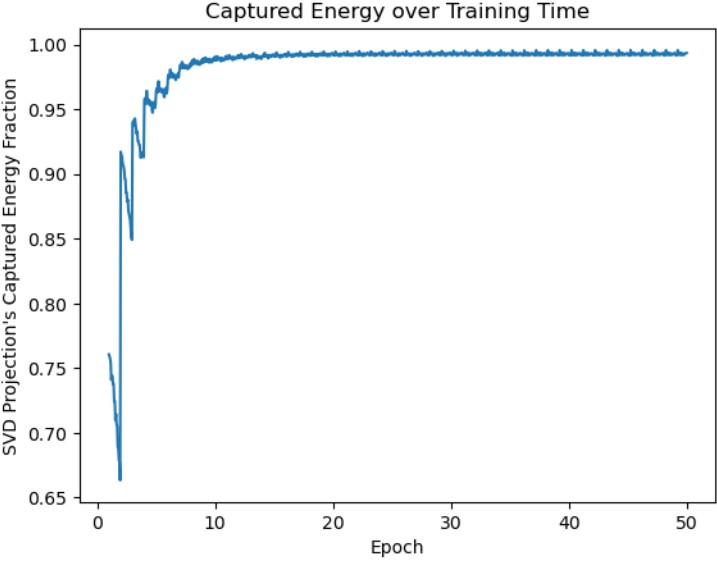

Figure 25: Energy captured by our Oja-rule generated projector over training time. The projector is fixed once per epoch, so we expect a decline in performance over the epoch, scaling with the adaptivity of the network during that epoch. Although initially quite variable, by epoch 10 the energy captured remains above 99%.

---

**Algorithm 1** Compute top-$K$ modes and drives from final-time train features

---

**Require:** Train-feature matrix $\Phi_{\text{train}} \in \mathbb{R}^{M \times P}$ with rows $g_m^\top$, where $g_m := \epsilon_m \phi_m = \nabla_\theta \ell(f(x_m; \theta), y_m)$
**Require:** Number of modes $K$
**Ensure:** Parameter-space modes $\{v_k\}_{k=1}^K$, singular values $\{s_k\}_{k=1}^K$, mode drives $\{\lambda_k\}_{k=1}^K$
 1: Compute truncated SVD: $\Phi_{\text{train}} = USV^\top$ with $S = \text{diag}(s_1, \ldots, s_K)$
 2: **for** $k \leftarrow 1$ **to** $K$ **do**
 3: $\quad a_k \leftarrow u_k^\top \mathbf{1}$ $\qquad\qquad\qquad\qquad\qquad\qquad\qquad\qquad\qquad$ ▷ $u_k$ is the $k$-th column of $U$
 4: $\quad \lambda_k \leftarrow a_k \, s_k$
 5: $\quad v_k \leftarrow k$-th column of $V$
 6: **end for**
 7: **return** $\{v_k\}_{k=1}^K$, $\{s_k\}_{k=1}^K$, $\{\lambda_k\}_{k=1}^K$

---

## A.10 Theoretical Analysis of Adversarial Influence Cancellation

$$\Delta L(\hat{y_n}, t) \approx -\frac{\eta}{M} \sum_{m \in M_{train}} \frac{\mathrm{d}L(y_m)}{\mathrm{d}\theta(t)} \cdot \frac{\mathrm{d}L(y_n)}{\mathrm{d}\theta(t)}$$

$$= -\frac{\eta}{M} \sum_{m \in M_{train}} \epsilon(t)_m \phi(t)_m \cdot \epsilon(t)_n \phi(t)_n$$

Let $\Phi(t)_M$ be the concatenation of $\epsilon(t)_m \phi(t)_m$ s.t. $\Phi(t)_M \in \mathbb{R}^{M \times p}$. Let $U, S, V = SVD(\Phi(t)_M)$, with rank $R$. Then

$$\sum_{m \in M_{train}} \epsilon(t)_m \phi(t)_m \cdot \epsilon(t)_n \phi(t)_n = \sum_{r \in R} S_r (U_r \otimes V_r) \cdot \phi(t)_n$$

For $r' << R$, $r < r'$ implies that $V_r \cdot \phi(t)_n$ large means that a a perturbation along $\phi(t)_n$ will likely cause a large change in loss, since this is within the current top modes of the loss sensitivity. However, by itself this would also imply that any loss change (particularly an increase) would be transient, as this perturbation is also aligned with the directions of maximum learning. However, if $\sum_r Sr(Ur \otimes Vr) \cdot \phi(t)_n = 0$, then overall learning is zero, with cancellation among $r$ components. Note that $U, S, V$ are fixed (as we don't modify the training setup), so the only free parameter is $\phi(t)_n$. Per $r$, we can qualify overlap as a single scalar: $S_r V_r \cdot \phi(t)_n$, which is then broadcast against the (data-dimensional) $U_r$. Thus, an adversarial perturbation could broadly be an input that maximizes loss perturbation, subject to minimizing the (data-dimensional weighted) learning sum over all modes.

## A.11 Comparing Auditing to TracIn

We repeat our auditing MNIST experiment, but using the TracIn algorithm. We test in two different modes - only evaluating the last layer (as is done in the TracIn paper to save on computational cost) and over all layers (for a closer comparison to our auditing method). Note that the in-run data shapely (IRDS) method's first-order tracking method is computationally equivelant to the full-layer TracIn case. In both cases, reconstruction accuracy is significantly degraded compared to auditing.

## A.12 Comparing Auditing to TRAK

We repeat our auditing MNIST experiment, but using the TRAK algorithm. Notably, TRAK attempts to measure the counterfactual value-add of a data point, rather than its current influence (see Remark 5.3). However, we can still compare the two methods' abilities at overall reconstruction (even though TRAK is designed for counterfactual estimation rather than reconstruction), by looking at how the sum of TRAK's per-sample importance compares to the actual output, allowing us to use the same ML and correlation based metrics. The results from Table 4 show that TRAK has dramatically worse reconstruction accuracy, and that this accuracy does not include with additional checkpoints. However, TRAK does have an advantage in its increased speed, running approximately 10x faster than a high fidelity audit. Note we have not tested auditing on TRAK's intended use case (counterfactual value), where we expect it would be superior as designed.

---

**Algorithm 2** Mode-aware cure via projected gradient descent (PGD)

---

**Require:** Model $f(\cdot, \theta)$, loss $\ell(\cdot, \cdot)$
**Require:** Adversarial (or random) input $x_{\text{adv}}$, clean reference $x_{\text{clean}}$ (optional)
**Require:** Modes $\{v_k\}_{k=1}^K$, drives $\{\lambda_k\}_{k=1}^K$, shift magnitude $\tau$
**Require:** Constraint radius $\varepsilon$, norm $p \in \{\ell^\infty, \ell^2\}$, steps $T$, step size $\alpha$
**Require:** Weights $\lambda_{\text{match}}, \lambda_{\text{ortho}}, \lambda_{\text{loss}}$
**Require:** (Optional) true label $y_{\text{true}}$ (used iff $\lambda_{\text{loss}} > 0$)
**Ensure:** Cured input $x_{\text{cure}}$

1: $x_0 \leftarrow x_{\text{adv}}, \quad x \leftarrow x_{\text{adv}}$
2: **if** $x_{\text{clean}}$ is provided **then**
3:     $\delta_{\text{ref}} \leftarrow x_{\text{adv}} - x_{\text{clean}}$
4: **else**
5:     $\delta_{\text{ref}} \leftarrow 0$
6: **end if**
7: **if** $\lambda_{\text{loss}} > 0$ **and** $y_{\text{true}}$ is provided **then**
8:     $y^\star \leftarrow y_{\text{true}}$
9: **else**
10:     $y^\star \leftarrow \arg\max f(x_{\text{adv}}; \theta)$                  ▷ pseudo-label for label-free cure
11: **end if**
12: $g_{\text{adv}} \leftarrow \nabla_\theta \ell(f(x_{\text{adv}}; \theta), y^\star)$
13: **for** $k \leftarrow 1$ **to** $K$ **do**
14:     $c_{\text{adv}}[k] \leftarrow v_k^\top g_{\text{adv}}$
15:     $c_{\text{tgt}}[k] \leftarrow c_{\text{adv}}[k] - \tau \cdot \text{sign}(\lambda_k)$
16: **end for**
17: **for** $t \leftarrow 1$ **to** $T$ **do**
18:     $g \leftarrow \nabla_\theta \ell(f(x; \theta), y^\star)$
19:     **for** $k \leftarrow 1$ **to** $K$ **do**
20:        $c[k] \leftarrow v_k^\top g$
21:     **end for**
22:     $L_{\text{match}} \leftarrow \frac{1}{2} \|c - c_{\text{tgt}}\|_2^2$
23:     $\delta \leftarrow x - x_0$
24:     $L_{\text{ortho}} \leftarrow \cos^2(\delta, \delta_{\text{ref}})$             ▷ penalize alignment; discourages "undoing"
25:     **if** $\lambda_{\text{loss}} > 0$ **then**
26:        $L_{\text{loss}} \leftarrow \ell(f(x; \theta), y_{\text{true}})$
27:     **else**
28:        $L_{\text{loss}} \leftarrow 0$
29:     **end if**
30:     $L \leftarrow \lambda_{\text{match}} L_{\text{match}} + \lambda_{\text{ortho}} L_{\text{ortho}} + \lambda_{\text{loss}} L_{\text{loss}}$
31:     $x \leftarrow x - \alpha \cdot \text{Normalize}_p(\nabla_x L)$
32:     $x \leftarrow \text{Proj}_{\mathcal{B}_p(x_0, \varepsilon)}(x)$             ▷ project onto $\ell_p$-ball around $x_0$
33:     $x \leftarrow \text{clip}(x, 0, 1)$
34: **end for**
35: **return** $x$                                          ▷ $x_{\text{cure}}$

---

## A.13 Unique Value of PLGK-Auditing compared to Tracin, TRAK

In addition to having higher reconstruction accuracy, the PLGK-based audit also allows for a (potentially simultaneous) decomposition of of a trained network's outputs along the parameter, training-time, and data axes simultaneously. This temporal and structural decomposition enables the per-layer, per-epoch, and per-mode adversarial analysis presented in the main text. This analysis cannot be generated by other methods that discard the training trajectory, and this capability directly lead to our novel adversarial findings.

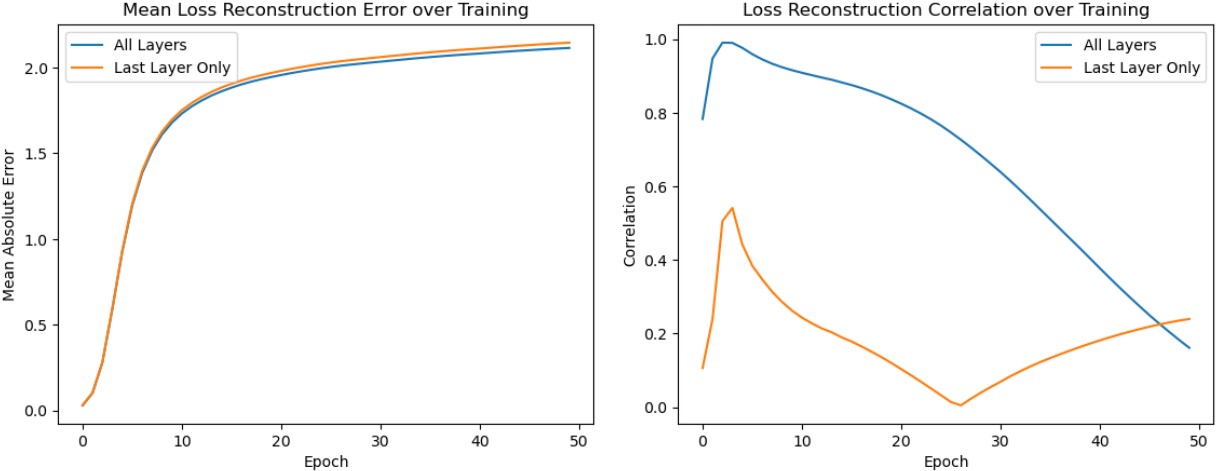

Figure 26: TracIn reconstruction results over training time. Left: Mean Absolute Reconstruction Error between true dynamics, TracIn reconstruction. Right: correlation between true and reconstructed dynamics. Notice the scale differences in the y-axis compared to the audit reconstruction results

Table 4: TRAK vs. Audit Attribution Comparison (MNIST, LeNet-5)

| Method | MLE $\downarrow$ | Correlation $\uparrow$ | Total Time (s) |
|---|---|---|---|
| Audit (trapezoidal) | **0.0104** | **0.999998** | 4040.0 |
| TRAK (1 checkpoint) | 2.3902 | 0.6099 | **334.1** |
| TRAK (5 checkpoints) | 2.9349 | 0.5152 | 447.1 |

### A.14 Loss-Auditing (Kernel View), Using Eigendecomposition

The SVD-mode analysis in Section 6.3 decomposes the *training-side* feature matrix $\Phi_{\text{train}}$ to identify which gradient directions drive adversarial cancellation. A complementary view is to decompose the PLGK Gram matrix $\mathbf{P}'$ itself via eigendecomposition, yielding a spectral representation of the loss audit that separates per-mode contributions on both the training and evaluation sides simultaneously. This view is particularly natural when the audit set coincides with the training set (e.g., for self-auditing or leave-one-out style analyses), and also provides a closed-form expression for extending the audit to new, non-training evaluation points. We include this derivation for completeness; empirical exploration of eigendecomposition-based auditing is left to future work.

From our baseline,

$$K'(t)[i,j] = (\epsilon(t)_i \phi(t)_i) \cdot (\epsilon(t)_j \phi(t)_j)$$

$$P'[i,j] = \sum_t K'(t)[i,j]$$

$$L(\hat{y}_n, T) - L(\hat{y}_n, 0) \approx -\frac{\eta}{M} \sum_{m \in D_{\text{tr}}} P'[m,n]$$

Define $P'$ to be over the train set (e.g. both $i, j \in D_{\text{tr}}$). Then, we can perform an eigendecomposition on $P'$,

$$P' = Q\Lambda Q^{-1}$$

Then, we can instead write

$$L(\hat{y}_n, T) - L(\hat{y}_n, 0) \approx -\frac{\eta}{M} \sum_k \lambda_k (\sum_m Q_{m,k})(Q^{-1})_{k,n}$$

Simplify by defining $s_k = (\sum_m Q_{m,k})$

$$L(\hat{y}_n, T) - L(\hat{y}_n, 0) \approx -\frac{\eta}{M} \sum_k \lambda_k s_k (Q^{-1})_{k,n}$$

Note, $s_k$ is the kth right-eigenvector sum, while $(Q^{-1})_{k,n}$ is the kth left-eigenvector nth element.

What about the loss of a non-training example? If we augment $P'$ one new column $n^*$, then

$$L(\hat{y}_{n^*}, T) - L(\hat{y}_{n^*}, 0) \approx -\frac{\eta}{M} \sum_k s_k (Q_{:,k} \cdot P[:, n^*])$$

$$= -\frac{\eta}{M} \sum_k s_k \sqrt{\lambda_k} (u_k \cdot \phi_{n^*})$$

where $u_k$ is the kth element of the U matrix from $\text{USV} = \text{SVD}(\Phi_{train})$.

### A.15  SVD Modes with Random Perturbation

We repeat our examination technique used to test how adversarial mode activations $c_{k,adv}$ differ from $c_{k,clean}$, repeating it for a test case of how randomly perturbed examples differ from clean examples. As shown in Figure 27, once again only the top modes show significant differences, albeit with a much lower scale. This implies that adversarial modes are chosen in such a way that there is a much larger than expected (via random perturbation) change in $C_k$.

### A.16  SVHN

We repeat some of our experiments, but on a new setting. We use the SVHN task (Netzer et al., 2011), and a small CNN architecture (with roughly 2.5 times more parameters compared to LeNet5 MNIST.

### A.17  SVHN Auditing Profiling

We repeated our profiling experiment, running our auditing framework with various options on the SVHN task, compared to a baseline (training only) run. Compared to the MNIST profiling, we show only the main auditing options. Once again we prioritized extremely high auditing accuracy, so significant time savings are possible at lower fidelity.

| Method | Time (s) | Memory (GiB) | Final MLE | Final Corr |
|---|---|---|---|---|
| Baseline Train | 1805.1 | 1.562 | — | — |
| Audit | 15059.4 | 10.697 | 0.040377 | 0.997890 |
| Supersample (2x) | 16691.4 | 10.821 | 0.032734 | 0.999074 |
| Subsample (2x) | 13311.7 | 15.999 | 0.125541 | 0.983261 |

Note this setup has slightly lower baseline auditing accuracy than MNIST, allowing the supersample (x2) to demonstrate improved fidelity.

Comparing runtime across MNIST and SVHN allows us to get a rough estimate of the audit time complexity, which is approximately linear in the models' parameter counts. Thus, we can expect the approximate 5x-10x time cost for auditing compared to baseline training to continue to hold as models scale up in size and complexity.

### A.17.1  SVHN - Audit

We complete a high-accuracy loss audit, with final audit metrics of a mean loss reconstruction error of 0.04, and a loss reconstruction correlation of 0.9979, as shown in Figure 28.

We again use auditing to analyze a single, incorrect decision, shown in Figure 29.

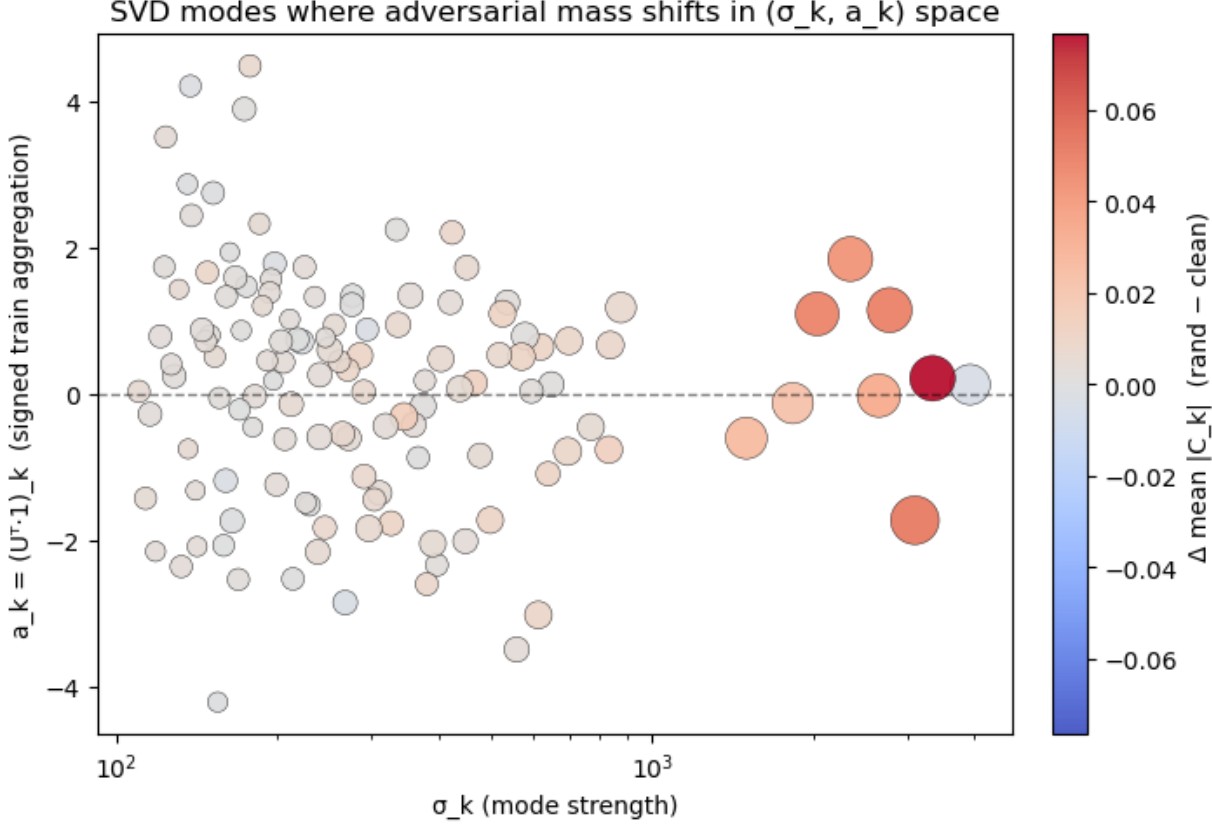

Figure 27: Plot showing the mean change in mode activity between randomly perturbed and clean inputs (size/color), scattered against mode strength (x-axis) and signed train aggregation (y-axis). The difference is again concentrated in the top-9 largest singular modes, while using both positive and negative train aggregations. However, notably the first mode has near zero difference, while overall mode strength variation is much lower compared ot the adversarially perturbed case in the main text

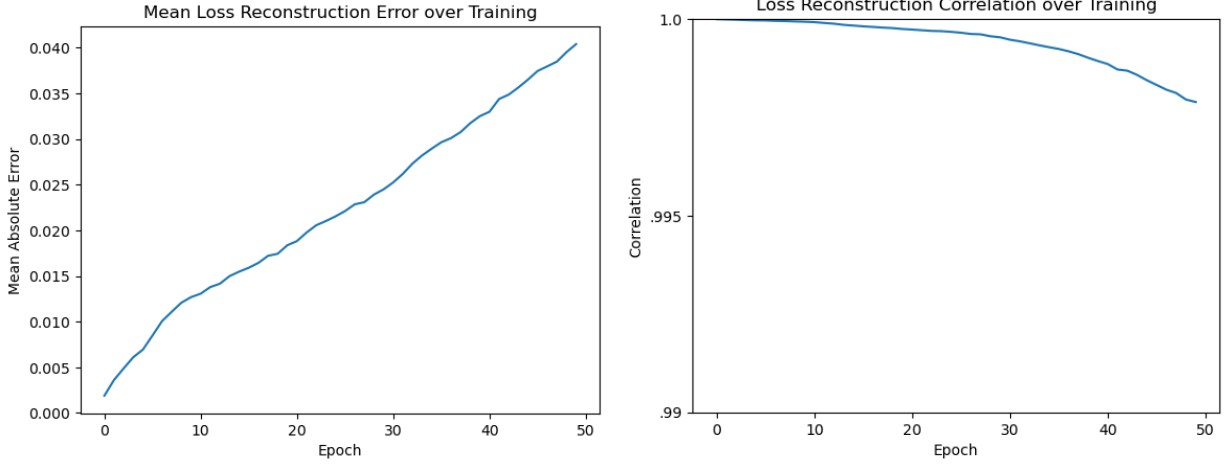

Figure 28: Audit reconstruction results over training time. Left: Mean Absolute Reconstruction Error between true dynamics, audit reconstruction. Right: correlation between true and reconstructed dynamics

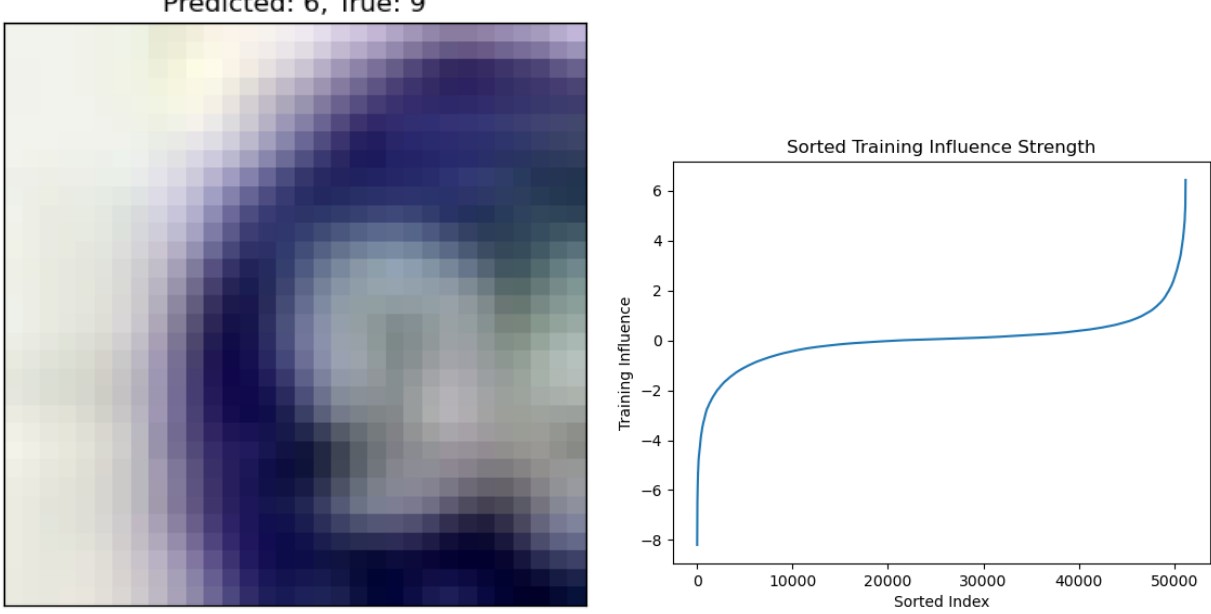

Figure 29: Auditing a single wrong decision Left: Input 9 misidentified as a 6 by the trained network. Right: Influence strength distribution, showing that a relatively small number of training examples determine the final output

This 9 is misinterpreted as a 6. Why might this have happened? We can again look at the top and bottom instances of the true and predicted class. From the true class, we can see that the most harmful (or least useful, as their influence is still negative) 9s are stereotypical 9s (centered, highly legible, upward curving final bottom stroke), whereas the most helpful 9s are unusual(e.g. rotated, un-centered, smaller, with a bottom stroke that does not curve back up, etc). (Figure 30)

The 6s are even more insightful, as they can provide both positive and negative influence. As seen in Figure 31, helpful sixes tend to have a top stroke that does not curve back down, while harmful sixes are much more likely to either be rotated or have a top stroke that does curve back down, suggesting that this feature (shared between 6s and 9s) has been inappropriately used to determine class 6 in this case.

### A.17.2  SVHN - Similarity Metric

We compute our similarity metric over the SVHN training task as well, again projecting the similarity matrix into 2D via UMAP (Figure 32).

We again examine the actual similarity metric (projected into our 2D UMAP) in Figure 33. This shows much higher similarities within the class, with overall similarity scaling with the UMAP distance (both within and between classes)

Finally, we repeat our demonstration that the similarity (projected onto UMAP dimensions) captures relevant feature-level information about our data points. In Figure 34, we demonstrate this by showing the highest and lowest points per UMAP dimension that come from class 7. We can see that dimension 1 codes stereotypical blue 7s on white backgrounds (for high values), while lower values often encode for blocker 7s, and are often white. By contrast, dimension 2 codes for stereotypical white 7s on blue backgrounds (for high values), while lower values code for blurry or difficult 7s. Although these dimensions are less clear than MNIST (due to the increased dimensionality of 'seven-ness' in SVHN), they are clearly still meaningful differences across each dimension.

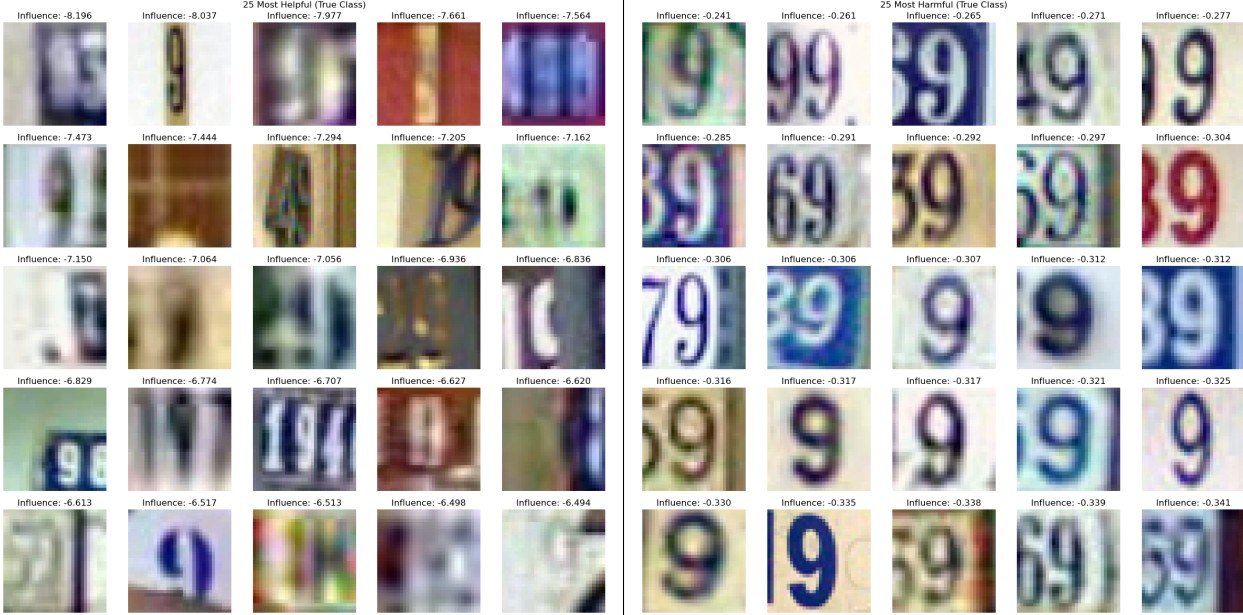

Figure 30: Left: 25 most helpful true class training data points. Right: 25 most harmful true class training data points.

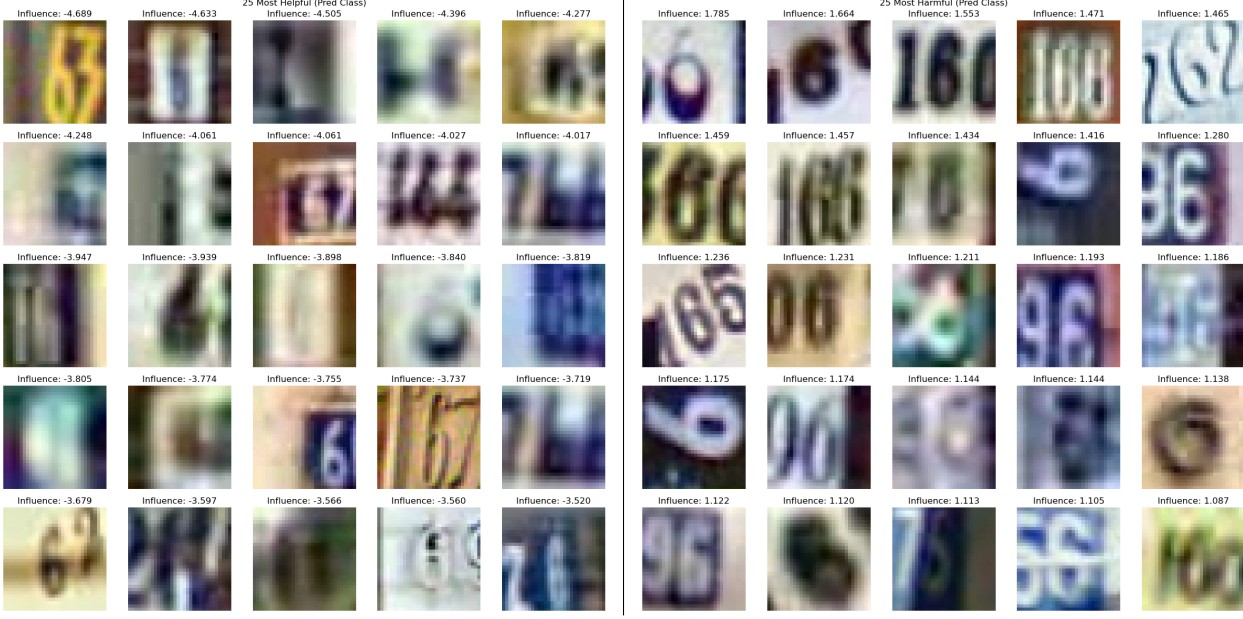

Figure 31: Left: 25 most helpful predicted class training data points. Right: 25 most harmful predicted class training data points.

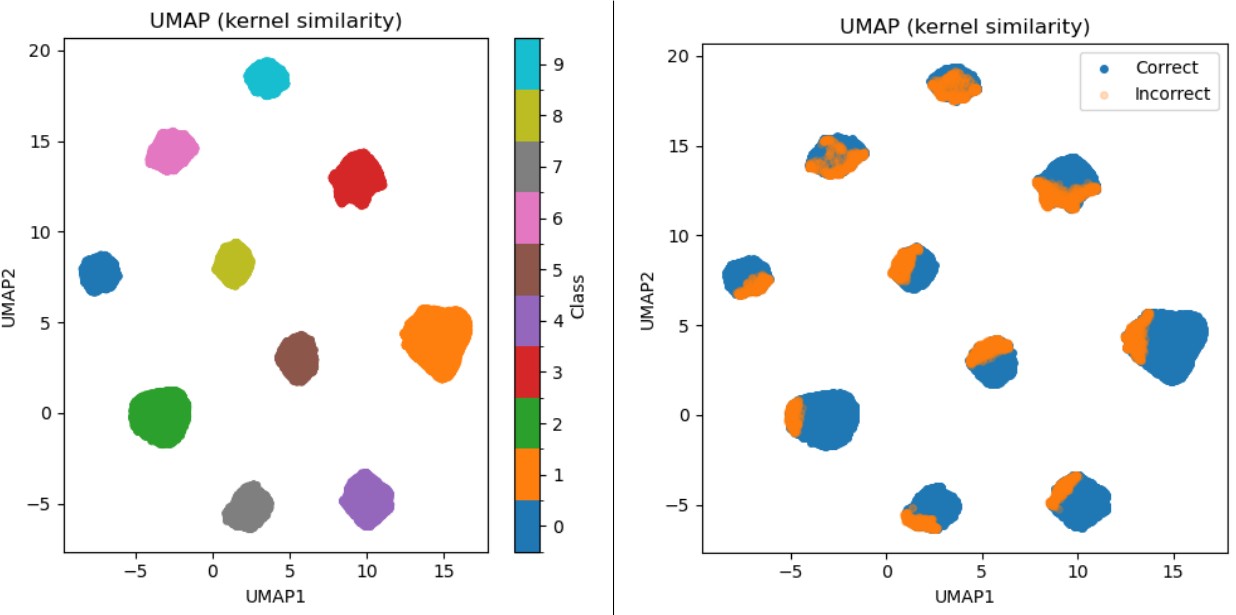

Figure 32: Left: 2D UMAP projection of similarity metric, colored by true class index. Right: 2D UMAP projection of similarity metric, with misclassified points marked. Notice that they are on the outer edge of class clusters

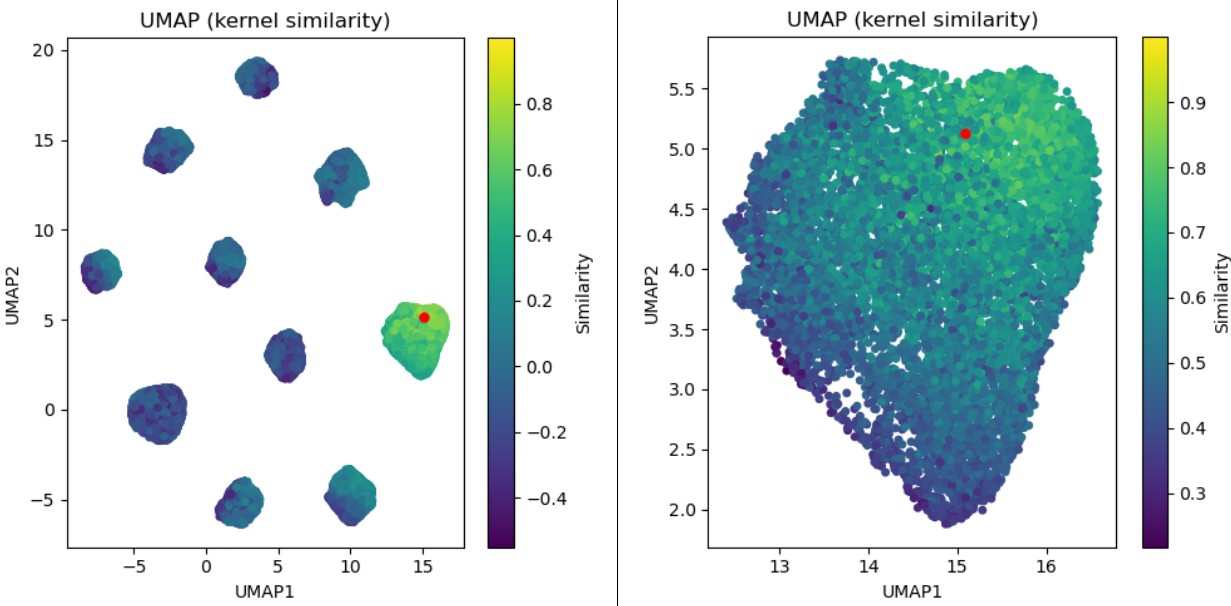

Figure 33: Left: 2D UMAP projection of similarity metric, showing similarity vs a target point. Right: Same, but zoomed into only the same class as the target point.

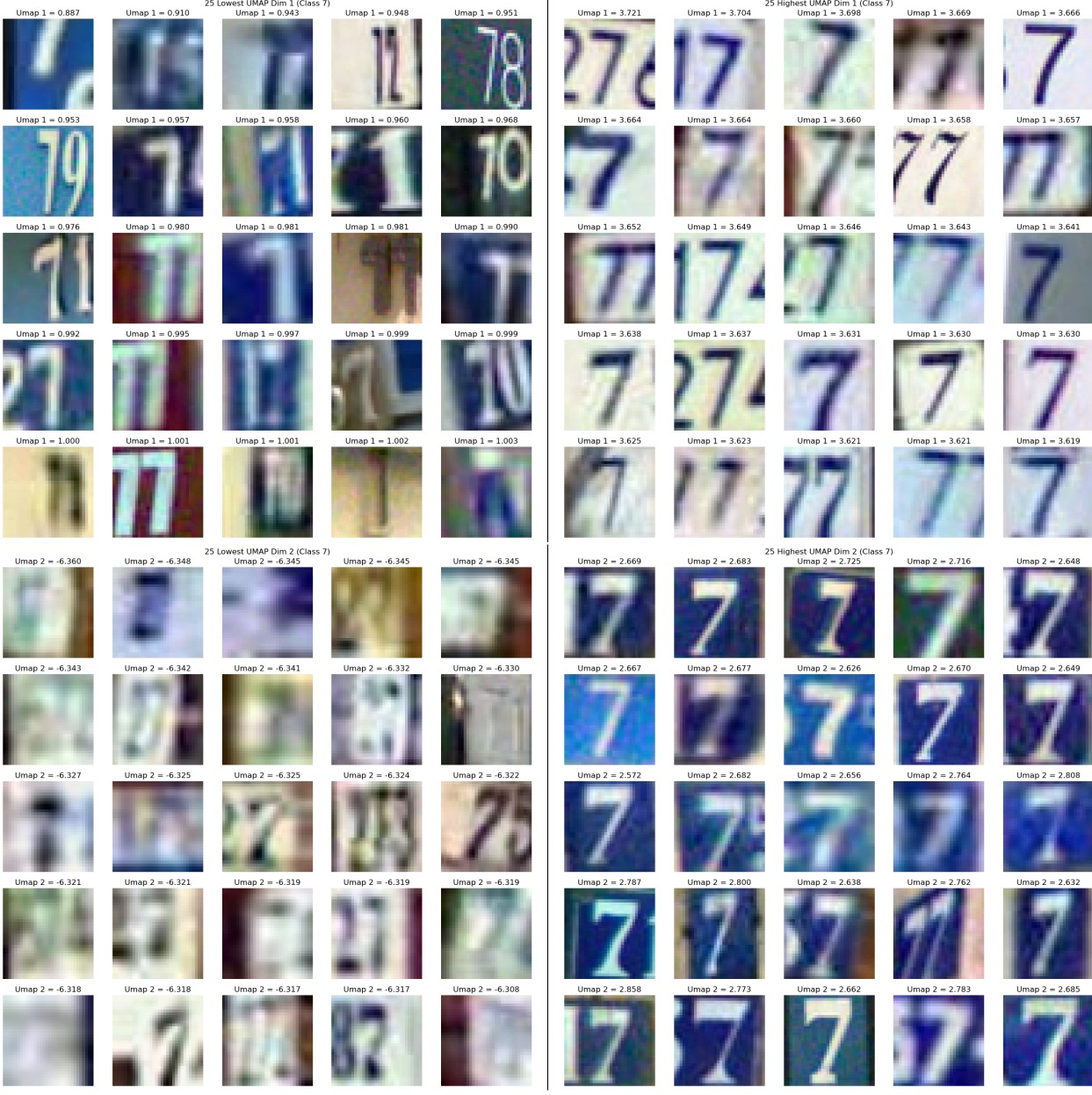

Figure 34: Top Left: 25 lowest UMAP dimension 1 valued class 7 examples. Top Right: 25 highest UMAP dimension 1 valued class 7 examples. Bottom Left: 25 lowest UMAP dimension 2 valued class 7 examples. Bottom Right: 25 highest UMAP dimension 2 valued class 7 examples. We can see clear differences in the 7s across the two dimensions.

Figure 35: Three data point sets, each consisting of a clean image, an adversarial image, and 2 randomly perturbed images. Compared to MNIST, adversarial example require lower attack strength (and less visible corruption) to succeed.

### A.17.3 SVHN - Adversarial

We repeat some of the adversarial experiments in the SVHN setting. We perform standard training, then use the foolbox package to generate a set of adversarial examples (of test data), which we again link to unperturbed and randomly perturbed test examples (see Figure 35). We then loss audit learning performance on these sets of perturbed points, reaching a final MLE of 0.046, and a final prediction correlation of 0.993.

We again compare the clean vs adversarial influences (Figure 36, revealing that the influence strength increase is broadly distributed among the training set, and is very well approximated by a simple linear scaling which increases overall influence magnitude (corr $>$ .99).

We skip the in depth adversarial-path and single-adversarial experiments, as we already have our hypothesis (cancellation of training influences leading to 0 learning, clustering in top modes of an SVD breakdown), and instead seek to confirm that we still find the relevant findings in the SVHN task.

We first confirm that adversarial examples have higher influence magnitudes, and loss that stops decaying(Figure 37). Compared to the MNIST case, the transition period takes longer, with final balancing occurring only towards the end of training. Nevertheless, there are clear gaps between both influence magnitude and loss across groups by the end of training.

We next confirm that this behavior aligns with a rich/adaptive to lazy/kernel regime transition, using our previous set of 3 metrics, and comparing them to the loss and influence curves. We plot our results in Figure 38. As expected, all 3 metrics begin to converge towards 1.0, however the knee occurs between epochs 10 and 20 (depending on metric), lining up with when the influence scales begin to separate and adversarial loss begins to stop decreasing.

Finally, we confirm that the differences between adversarial and clean examles primarily occur in the top-SVD modes e.g. how adversarial mode activations $c_{k,adv}$ differ from $c_{k,clean}$. As shown in Figure 39, only the top modes show significant differences in the SVHN task as well.

When compared to the difference between random and baseline perturbations (Figure 40), we see that the differences are about 2 OOMs stronger.

Overall, we find that a similar story holds in the SVHN case. As the network becomes increasingly kernel regime, a gap opens up between adversarial and randomly perturbed/unperturbed examples in terms of both loss and influence scale. These differences result in decreased / halted learning, leading to an elevated final loss, which must occur by influence cancellation. A SVD-mode lens reveals that these differences are strongly clustered within the top-10 modes.

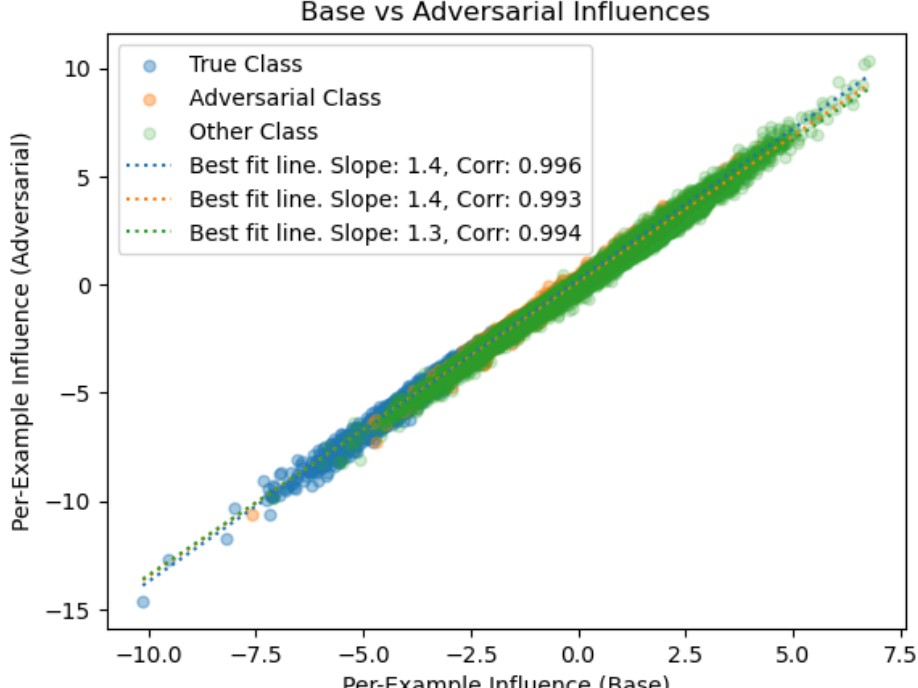

Figure 36: Scatter plot of Clean vs Adversarial per-example influences. This reveals that influences are slightly stronger across the entire train set for adversarial examples.

Table 5: Scaling experiments summary. Per-epoch and total wall times measured on a single AWS `g5.xlarge` (NVIDIA A10G, 24 GiB). "Raw" is the cost of training alone (no audit), measured by running the same training loop with the audit machinery disabled. "Mode" is either full audit or the IPS-only accumulator. All runs use audit-aligned projection with $k = 64$ matching the audit set size. *Pearson* is the correlation between predicted per-audit-sample loss change and the actual measured change at end of training. The IPS–audit per-sample influence correlation on CIFAR ResNet-18-GN was measured directly (Table 6) at per-audit mean Pearson 0.94–0.95, stable from epoch 20 through epoch 120; no independent audit reference was computed at ImageNet scale, since establishing that reference is precisely the cost the IPS variant is designed to avoid.

| Setting | $|\theta|$ | $|D_{\text{tr}}|$ | Mode | Raw s/ep | Mode s/ep | Overhead× | Peak GPU | Final Acc |
|---|---|---|---|---|---|---|---|---|
| CIFAR / ResNet-18-GN | 11.2M | 49,920 | Audit | 14.9 | 126 | 8.5× | ∼6.0 GiB | 56.6% |
| CIFAR / ViT-Tiny/4 | 2.70M | 49,920 | Audit | 15.8 | 178 | 11.3× | 4.62 GiB | 46.8% |
| ImageNet / ResNet-18-GN | 11.7M | 1.28M | IPS-only | 2,489 | 6,772 | 2.7× | 22.4 GiB | 41.3% top-1 |

## A.18 Scaling Experiments

We validate the practical scalability of the PLGK framework with three runs spanning two architectures, two datasets, and both the full audit and the lighter-weight IPS-only variant. The full results are summarized in Table 5; the subsections below describe each setting and the overall lessons.

### A.18.1 Scaling to Realistic Datasets

The full audit accumulates a momentum buffer of recent training-sample gradients in order to attribute every parameter update to the training examples that contributed to it. With buffer size $B$, batch size $b$, and effective projected dimension $k$, the dominant memory cost is the $\mathcal{O}(B \cdot b \cdot k \cdot 4)$ momentum buffer plus the per-step audit gradient compute. On CIFAR-10 with ResNet-18-GN at our representative production

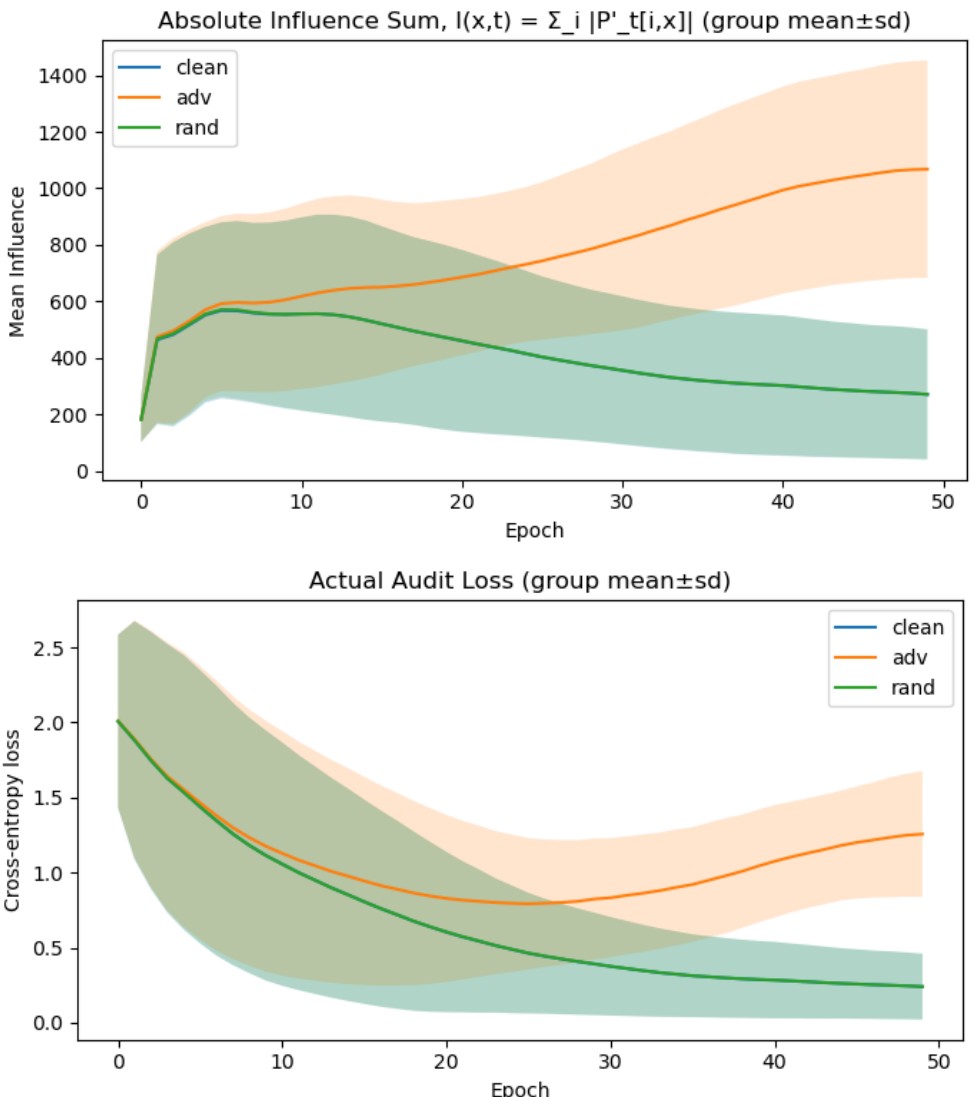

Figure 37: Top: Per-epoch mean influence magnitude, compared across clean, adversarial, and randomly perturbed inputs. Bottom: Per-epoch mean loss, across the same 3 groups.

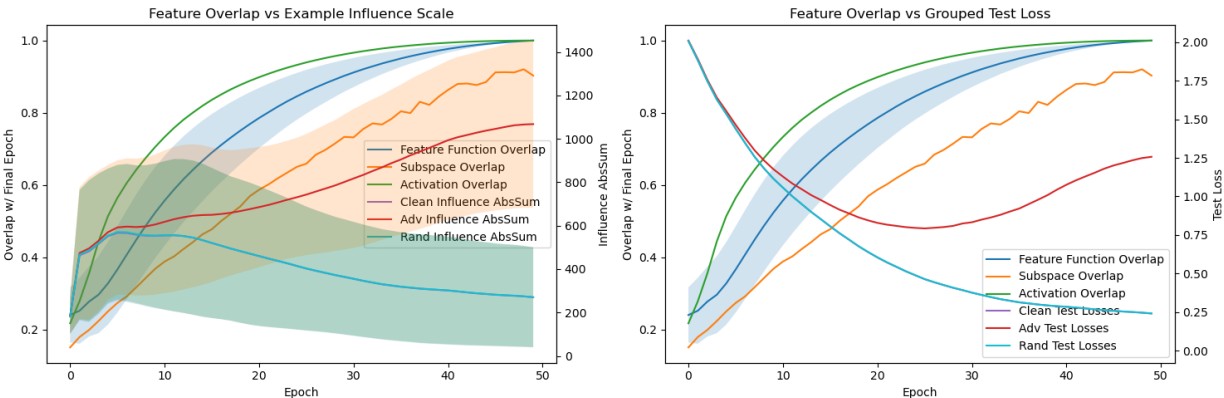

Figure 38: Kernel-Regime metrics (blue/orange/green lines, left y-axes) Left) vs Absolute Influence Sum (purple/red/cyan lines, right y-axis; Right) vs grouped example loss (purple/red/cyan lines, right y-axis). They show an alignment, with all 3 metrics rapidly growing between epochs 10 and 20, with the 'knee' approximately divergence in influence scale (left plot) and where adversarial loss begins to stop decreasing (right plot). Compared to MNIST, this transition occurs later and slower.

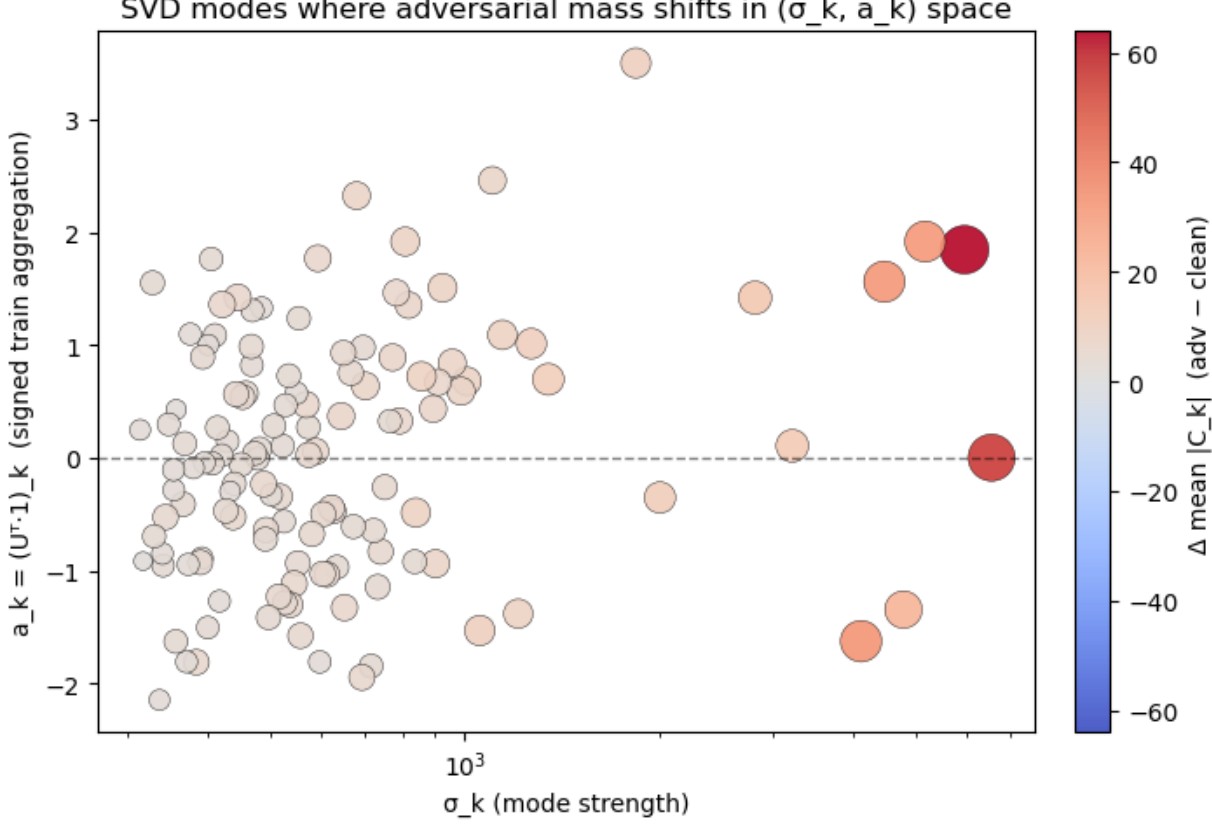

Figure 39: Plot showing the mean change in mode activity between adversarial and clean inputs (size/color), scattered against mode strength (x-axis) and signed train aggregation (y-axis). The difference is concentrated in the top-9 largest singular modes, while using both positive and negative train aggregations.

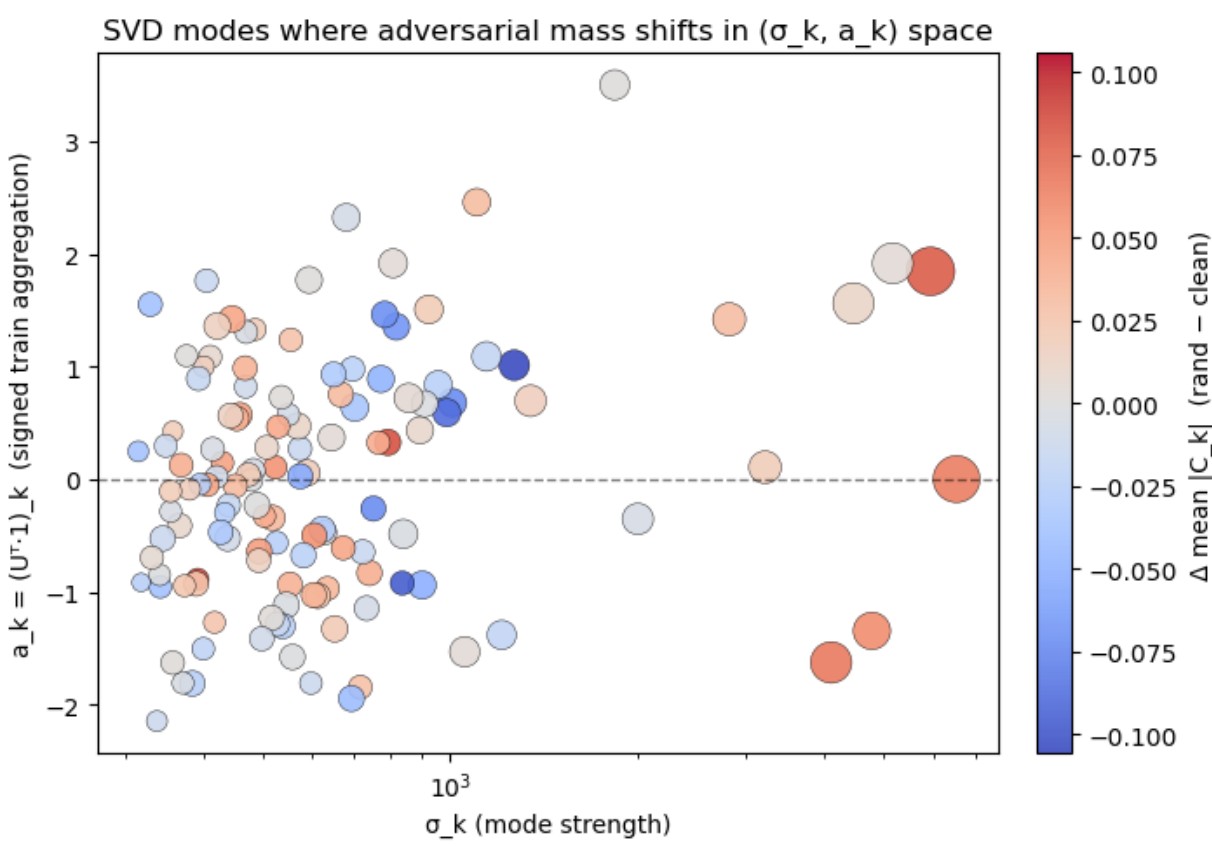

Figure 40: Plot showing the mean change in mode activity between random and clean inputs (size/color), scattered against mode strength (x-axis) and signed train aggregation (y-axis). The differences are small, and spread among all considered modes.

Table 6: IPS–Audit per-sample influence correspondence on CIFAR-10 / ResNet-18-GN, 120-epoch run, $k$=64 audit-aligned projection, $B$=20 momentum buffer, lr=$10^{-6}$. The lr (lr-weighted) IPS variant is reported; the raw and lr-eta variants give qualitatively similar numbers. Per-audit Pearson is the column-wise correlation between the audit's per-sample influence and the IPS-predicted per-sample influence, aggregated across the 64 audit samples. Stable correlation through ep120 establishes IPS as a faithful low-cost proxy for the full audit at this architecture and dataset scale.

| Epoch | All-cells Pearson | Per-audit mean | Per-audit median | Per-audit min |
|---|---|---|---|---|
| 20 | 0.932 | 0.941 | 0.944 | 0.870 |
| 50 | 0.945 | 0.944 | 0.949 | 0.824 |
| 80 | 0.945 | 0.944 | 0.953 | 0.824 |
| 120 | 0.944 | 0.945 | 0.954 | 0.844 |

configuration ($k = 64$ audit-aligned, $B = 20$, batch 128), the audit runs at 126 s/epoch against a raw baseline of 14.9 s/epoch — an 8.5× overhead. In a separate 120-epoch validation run at $lr$=$10^{-6}$ with the same projection and buffer settings, the per-sample IPS–audit influence correlation reaches mean Pearson 0.941 at epoch 20 and remains stable at 0.944–0.945 through epoch 120 (Table 6), motivating the IPS-only variant as a faithful but dramatically lighter proxy.

Extrapolating the full-audit overhead to ImageNet was prohibitive: a single epoch would cost roughly $8.5 \times 2{,}489 \approx 21{,}000$ s ($\approx 5.9$ hours), and the momentum buffer alone — $B \cdot b \cdot n_{\text{params}} \cdot 4$ at full dimension — would substantially exceed single-GPU memory on a 11.7M-parameter model unless aggressive projection is used. The IPS-only path eliminates the momentum buffer entirely and brings the overhead to **2.7×** raw training: 6,772 s/epoch vs. 2,489 s/epoch.

We trained ResNet-18-GN with the standard He et al. recipe (SGD, $lr$=0.1, momentum 0.9, weight decay $10^{-4}$, RandomResizedCrop(224) + horizontal flip) for the first 30 epochs (corresponding to the first constant-LR stage of the canonical 90-epoch schedule, prior to the /10 decay at epochs 30 and 60). Total wall time was approximately 64 hours. Final validation accuracy was 41.3% top-1 / 67.7% top-5, somewhat below the BatchNorm baseline of ~52% top-1 at the same epoch count; we attribute the gap to the Group-Norm substitution (required for compatibility with the per-sample gradient path; see Appendix A.18.3), the batch size of 128 (vs. the conventional 256 with the linear-scaling-rule $lr$=0.2), and the absence of LR warmup. The model is meaningfully trained (top-5 accuracy of 67.7% indicates a clear non-trivial classifier) and the per-sample influence matrix is well-defined; we view this as a feasibility demonstration that PLGK can be applied at ImageNet scale on a single commodity GPU, not as a benchmark-competitive training result.

Checkpoints (model state, optimizer state, IPS accumulators, and projection matrix) were saved every 5 epochs to support resume and downstream analysis. The final IPS influence matrix $K \in \mathbb{R}^{1,281,167 \times 64}$ ($\approx 313$ MB) is the central artifact: each row gives the predicted contribution of one ImageNet training image to the loss change on each of the audit samples.

### A.18.2 Transformer Validation

To verify that PLGK is not specific to convolutional architectures, we ran the full audit on a Vision Transformer. We chose the Lee et al. 2021 CIFAR-shape baseline: ViT-Tiny/4 with patch size 4, embedding dimension 192, depth 9, 12 heads, MLP ratio 2 ($\sim 2.70$M parameters). The architecture is LayerNorm-only by construction — no BatchNorm substitution is needed — and works out of the box. Training used AdamW with weight decay 0 (see Appendix A.18.3 for why the standard wd=0.05 was disabled), cosine warmup schedule, RandomCrop+HFlip augmentation, 100 epochs, batch size 128, audit-aligned projection with $k = 64$ matching the audit set, supersample $= 2$.

The training learning rate had to be reduced substantially below the DeiT-standard $lr$=$5\times10^{-4}$ to keep the audit's trapezoidal approximation accurate. At the standard rate the end-of-training correlation between IPS-predicted and actual per-audit-sample loss change was only $\sim 0.28$; at our chosen $lr$=$1.5\times10^{-6}$ it landed at 0.68 at epoch 90 after starting at 0.998 at epoch 1 and decaying smoothly through training. Per-epoch wall

time was 178 s (full audit, $k = 64$, supersample $= 2$, $B = 35$) vs. a raw baseline of 15.8 s (overhead: $11.3\times$), consistent with the ResNet-18-GN audit overhead at similar parameter count. The audit successfully tracks training across all 90 completed epochs, with final test accuracy 46.8%. We believe this demonstrates that the PLGK framework is architecturally agnostic in the sense that no algorithmic modification was required.

### A.18.3 Discussion: Strengths, Weaknesses, and Extensions

**Architecture agnosticism.** PLGK reduces to per-sample gradient computation, so any architecture that supports stateless forward passes works without modification. The ViT result above ran on the same audit infrastructure as the ResNet, with only a model swap.

**BatchNorm incompatibility.** The principal architectural restriction is that any module with stateful per-batch statistics (most notably BatchNorm's running mean/variance) breaks the per-sample gradient path: `vmap` treats running statistics as differentiable but non-stateful, producing incorrect gradients. Practical workarounds are to substitute GroupNorm (the choice we made for ResNet, costing $\sim$3-5 percentage points of accuracy versus BN at moderate batch sizes per (Wu & He, 2018)) or LayerNorm. Modern architectures that are LayerNorm-only by design (ViT and successors) avoid the issue entirely.

**Learning rate sensitivity.** The audit's trapezoidal integration of per-step parameter motion is exact in the limit $lr \to 0$. In practice training rates yield correlations that degrade over the course of training, with the rate of degradation controlled most strongly by the learning rate. Supersampling (multiple audit-gradient evaluations per step) reduces trapezoidal error, but may not fully recover performance due to numerical error. The practical guidance is to drop $lr$ when running with audit; the model still trains, but reaches lower final-task accuracy. This is the principal tradeoff of the framework when prioritizing audit fidelity.

**Optimizer interactions.** The audit's momentum-buffer accounting currently does not support the simultaneous use of momentum *and* weight decay in the same optimizer. We disabled weight decay for the ViT runs above (using AdamW with wd=0); the underlying issue is that the weight-decay term $-\lambda\theta$ would enter the momentum buffer and require separate accounting. Extending the audit to handle this is straightforward but was outside the scope of the present work. The IPS-only path is unaffected since it reads the effective learning rate per step without modeling the momentum buffer explicitly.

**Extensions to NLP / fine-tuning settings.** We briefly address how the PLGK interacts with common LLM workflows. *Token-level vs. sequence-level loss gradients*: PLGK operates on $\phi = \partial L/\partial\theta$ for any differentiable $L$. The default in autoregressive language modeling sums per-token cross-entropy across a sequence, so the existing framework yields sequence-level per-sample influence with no modification. To audit at token granularity instead, one treats each token as a sample at the cost of a per-batch dimension increase proportional to sequence length; the framework otherwise applies unchanged. *Checkpoint sampling*: our production runs already operate in a checkpoint-aware regime, serializing the influence accumulator $K$, the projection basis, and the audit-target initialization $L_{\text{init}}$ every five epochs; the saved $K$ at each checkpoint is the IPS-predicted total influence accumulated through that point in training. *LoRA-only gradients*: running PLGK with $\theta$ restricted to LoRA adapter parameters is a direct substitution and produces influence with respect to the fine-tuning delta, not the base model. The small parameter count of LoRA ($r \cdot (d_{\text{in}} + d_{\text{out}})$ per adapted layer) typically removes the need for projection, making full-dimensional audit straightforward; the interpretive caveat is that the resulting influence attributes the fine-tuning behavior to fine-tuning data only, leaving the contribution of pre-training data invisible.

**Compute summary.** The three runs together consumed approximately 80 GPU-hours on a single A10G, with the bulk ($\sim$57 hours) in the ImageNet IPS-only run. The CIFAR audit experiments are inexpensive enough to run as ablations; ImageNet-scale IPS is feasible on a single commodity GPU within standard cloud cost envelopes.

