# OpenReview forum: "Path-Integrated Loss-Gradient Kernels: Auditing and Similarity for Trained Neural Networks"
_TMLR — Under review for TMLR_

### Review · Reviewer_aLcY · 2026-06-07

**Summary Of Contributions:**

This paper proposes the Path-Integrated Loss-Gradient Kernel (PLGK), a Mercer kernel derived from the training trajectory of gradient-descent-trained neural networks. The authors develop two tools based on PLGK: (1) Intrinsic Perceptual Similarity (IPS), a behavior-based similarity measure between inputs, and (2) a high-fidelity auditing framework that decomposes model predictions into contributions from individual training samples. While the work presents interesting theoretical insights and well-executed small-scale experiments, it suffers from critical limitations in scalability, novelty, and practical utility that prevent it from meeting TMLR's acceptance criteria in its current form.

**Audience:**

Yes

**Audience Explanation:**

Yes, but only a very narrow subset. While the theoretical connection between training trajectories and kernel methods is of interest to researchers in theoretical deep learning, the paper's limited scalability and incremental nature mean it will not appeal to the broader TMLR audience.

**Broader Impact Concerns:**

The paper does not include a Broader Impact Statement, which is required for TMLR submissions that describe work with potential for harm

**Claims And Evidence:**

No

**Claims Explanation:**

While the evidence presented is technically accurate for the small-scale experiments conducted, it is insufficient to support the generalizability and practical significance of the claims.
1. All experiments are conducted on tiny datasets (MNIST, SVHN) with small models (LeNet-5, simple CNN with ~2.5× LeNet-5 parameters). There is no evidence that the proposed methods scale to modern deep learning models (e.g., transformers, ResNets) or real-world tasks. The authors acknowledge this limitation in the conclusion but do not provide any evidence of scalability or propose viable solutions.
2. The profiling results in Appendix A.5 and A.16 show that auditing takes 17.8× longer than training on MNIST (4040s vs 226.5s) and 8.3× longer on SVHN (15059.4s vs 1805.1s). For even moderately sized models (e.g., ResNet-50 with 25M parameters), this would make auditing computationally infeasible. The authors mention subsampling as a potential optimization but do not provide a comprehensive analysis of the accuracy-cost tradeoff for practical use cases.
3. The comparison with TRAK (Table 4) only measures reconstruction accuracy, not the counterfactual prediction performance that TRAK is designed for. The authors correctly note that TRAK and auditing have different goals, but they do not demonstrate when their auditing framework provides unique value that justifies its much higher computational cost.
4. The adversarial experiments only test simple PGD attacks on small models. There is no evaluation against more sophisticated attacks or on larger models where adversarial robustness is a more pressing concern.

**Requested Changes:**

1. Demonstrate the methods on at least one moderately sized model (e.g., ResNet-152 on CIFAR-10 and the ImageNet dataset) and provide detailed profiling results.
2. Compare the auditing framework with TRAK and TracIn on both reconstruction accuracy and downstream tasks
3. Evaluate the adversarial analysis on larger models and more diverse attack types.
4. The author should analyze the convergence properties of PLGK as model width and training time increase.

---

> ### Author Response · Authors · 2026-06-26
> **Response to Reviewer aLcY**
>
> We thank the reviewer for the detailed assessment. The reviewer's central concerns are (1) scalability evidence to larger models and real-world tasks, (2) a comprehensive accuracy–cost analysis, (3) a TRAK/TracIn comparison on downstream tasks (not only reconstruction), (4) adversarial evaluation on larger models and more diverse attacks,  and (5) a required Broader Impact Statement. We have made substantial additions in response, and below we both describe those changes and, where a request exceeds the scope of the paper's claims, clarify how we have calibrated the claims to match the evidence. We note that TMLR's acceptance criteria turn on whether claims are correct and supported and whether some audience would find them valuable, rather than on state-of-the-art scale; we have aimed our revisions at supporting our claims at that bar.
>
> ## Scalability and the accuracy–cost tradeoff (Requests 1, 2)
>
> We agree the original submission was light on direct scaling evidence, and have added additional (larger) runs in the appendix. We have also sharpened the framing of the paper's main results as theory focused with the high-fidelity audit as diagnostic for smaller models, with several plausible methods presented for potential paths towards utility on larger-scale tasks.
>
> To address this, we made the following changes:
>
> 1. Ran several new, larger models in the appendix. In particular, a table (Appendix) covers: CIFAR ResNet-18-GN audit (8.5× overhead), CIFAR ViT-Tiny/4 audit (11.3× overhead), and a partial-ImageNet ResNet-18-GN run (2.7x overhead, IPS only), with the full profiling table (params, throughput, overhead, peak memory).
> 2. Added results supporting IPS as scalable path: Dropping the the exact-reconstruction requirement of the audit allows cheaper IPS to recover ~95% of dense audit's between-sample ranking on CIFAR, and scales to ImageNet at ~2.7× training overhead; We view this as a more viable solution for large-scale ranking/similarity, with dense auditing reserved for smaller-scale full-fidelity needs.
> 3. Added a section to the main text before the conclusion explicitly discussing the strengths and limitations of PLGK-based methods, and linking to additional details on cost/accuracy techniques and additional larger runs in the Appendix.
>
> On Request 1 specifically (e.g., ResNet-152 on ImageNet with a full dense audit): we note that a dense audit at that scale is precisely the O(T×M×N×P) infeasibility the paper characterizes, and is not a setting in which we claim the dense method is practical. We have scoped the claims to reflect this, and offer the IPS path as the scalable alternative; we believe this supports the paper's diagnostic-framework claims within TMLR's criteria, while a full large-model dense audit is out of scope for the contribution as stated.
>
> ## Unique value of auditing vs. TRAK/TracIn (Request 2)
>
> The reviewer correctly notes that Table 4 compares reconstruction accuracy, the auditing objective, rather than TRAK's counterfactual (LDS) objective, and asks where auditing's higher cost is justified.
>
> To address this, we made the following changes:
>
> 1. Added an explicit statement of auditing's unique value: unlike TRAK (which linearizes at the final parameters via the after-kernel) and TracIn, the audit can decompose a single realized training run along the parameter, training-time, and data axes simultaneously. This temporal/structural decomposition is what enables the per-layer, per-epoch, and per-mode adversarial analysis of Section 6.3, an analysis that counterfactual attribution methods cannot produce, since they discard the training trajectory. This is the capability that justifies the cost when faithful reconstruction of a specific run is the goal.
> 2. Clarified that auditing and counterfactual attribution answer different questions, so each method is expected to lead on its own home metric; the comparison is intended to establish reconstruction fidelity, not to claim superiority at counterfactual prediction.
>
> ## Adversarial evaluation: attacks and model scale (Request 3)
>
> To address this, we made the following changes:
>
> 1. Clarified that all non-illustrative adversarial results use four L∞ attacks (FGSM, PGD-L∞, BIM-L∞, DeepFool-L∞) rather than PGD alone, and confirmed the cancellation signature is consistent across them.
> 2. Added a caveat discussing mechanism's limited experimental scope: consistent across two architecture–dataset pairs and four L∞ gradient-based attacks, but only on small CNNs under an L∞ first-order threat model. We specifically (in both the adversarial conclusion and overall conclusion) present our findings as a well-supported candidate mechanism rather than a universal account of adversarial vulnerability.
>
> Broader-scale adversarial study is left as future work.
>
> ## Broader Impact Statement (required)
>
> We thank the reviewer for noting this omission, and have added a Broader Impact Statement.

---

### Review · Reviewer_b3qW · 2026-06-09

**Summary Of Contributions:**

### Summary
The paper introduces Path-Integrated Loss-Gradient Kernels (PLGK), a path-dependent kernel formed by accumulating inner products of loss gradients along the training trajectory. The central theoretical point is that replacing one-sided output-gradient weighting with two-sided loss-gradient weighting yields a symmetric PSD object, with path feature map given by concatenated loss-gradient features over training. The paper derives two tools from this view: Intrinsic Perceptual Similarity (IPS), a cosine-normalized similarity over PLGK path features, and an auditing method that decomposes changes in loss or predictions into per-training-example influences.

Empirically, the paper evaluates mainly on MNIST/LeNet-5 and an appendix SVHN/CNN setting. It reports high audit reconstruction fidelity, qualitative influence analyses of misclassified examples, IPS UMAP visualizations and supervised difficulty scores, pruning experiments, comparisons to TracIn/TRAK, and a substantial adversarial case study arguing that adversarial examples exhibit high-magnitude but canceling training influences that can be disrupted by a mode-aware perturbation.

### Strengths
**Theoretical Elegance**: The shift from output gradients to loss gradients to recover a true, symmetric Mercer kernel from gradient descent trajectories is an insightful, simple, and structurally sound theoretical contribution.

**High-Fidelity Auditing**: The introduction of the trapezoidal correction for numerical integration reduces the discretization error from $O(||\Delta\theta||^2)$ to $O(||\Delta\theta||^3)$ essentially for free. The empirical reconstruction fidelity (MRE $\approx 0.01$) vastly outperforms baselines like TracIn and TRAK for analyzing a specific training run.

### Weaknesses
**Severe Scalability Limitations**: The proposed auditing and IPS methods have a time and memory complexity of $O(T \times M \times N \times P)$ in the dense setting, as openly acknowledged by the authors. The experiments are restricted to tiny datasets and architectures (MNIST/LeNet-5, SVHN/Small CNN) trained for very few epochs (e.g., 50 epochs on MNIST achieving only 93% accuracy).

**Additional Comments:**

N/A

**Audience:**

Yes

**Audience Explanation:**

Yes. The paper would likely interest TMLR readers working on interpretability, data attribution, training dynamics, neural tangent/path kernels, and adversarial robustness.

**Broader Impact Concerns:**

No significant ethical concerns are apparent.

**Claims And Evidence:**

Yes

**Claims Explanation:**

Yes. The claims in the submission are convincingly supported by rigorous mathematical proofs for the PLGK kernel and near-exact reconstruction metrics for the auditing framework, though the authors appropriately acknowledge that the $O(T \times M \times N \times P)$ complexity strictly limits their current empirical evidence to small-scale models.

**Requested Changes:**

There appears to be a notation and normalization inconsistency in the audit completeness equations. Definition 9 writes completeness with an external $-\eta/M$ factor multiplying the sum of audit entries, while Definitions 10 and 11 define audit updates that already include the $-\eta/M$ term. Later, the Mean Reconstruction Error (MRE) compares $\sum_m \tilde{P}_{m,n}$ directly to $q(T) - q(0)$. This needs correction because it directly affects the interpretation, sign, and scale of the influence values.

---

> ### Author Response · Authors · 2026-06-26
> **Response to Reviewer b3qW**
>
> We thank the reviewer for the close reading and for recognizing the theoretical contribution: the output-gradient to loss-gradient shift recovering a symmetric Mercer kernel, and the high-fidelity auditing afforded by the trapezoidal correction. The reviewer's requested change concerns a notation/normalization inconsistency in the audit completeness equations; we address it below, and also note the scope-calibration changes made in response to the weakness raised in the summary.
>
> ## 1. Normalization inconsistency in the audit completeness equations
>
> The reviewer is correct, and we thank them for catching this. The audit entries defined in Definitions 10–11 already absorb the −η/M step factor (and the trapezoidal weights), so the external −η/M in the completeness statement (Definition 9, Eq. 17) double-counts it. The Mean Reconstruction Error (Definition 12) is already written consistently with the audit definitions (it compares Σₘ P̃*ₘₙ directly to q(T) − q(0)) and this is the convention under which all reported results were computed, so no experimental values are affected.
>
> To address this, we made the following changes:
>
> 1. Corrected Definition 9 / Eq. 17 to remove the external −η/M factor (and fix the sign), so completeness reads `L(ŷₙ(θ_T)) = L(ŷₙ(θ₀)) + Σₘ P̃ₘₙ`, consistent with Definitions 10–12. We also ported the same fix to the relevant section in the Appendix.
> 2. Verified directly in the audit code that the reconstructed quantity is the bare sum Σₘ P̃ₘₙ, confirming the reported MRE values are unaffected.
>
> ## 2. Scope and scalability (noted as a weakness)
>
> While not listed as a requested change, the reviewer notes the O(T × M × N × P) dense cost and the small-scale experiments as a weakness, which were also concerns other reviewers raised.
>
> To address this, we made the following changes:
>
> 1. Revised the conclusion to explicitly position high fidelity auditing as a diagnostic for small-to-moderate models.
> 2. Added a new scaling and extension to larger models discussion section (Section 7, full details in Appendix) presenting preliminary results on larger models, as well as discussing several techniques that could reduce cost. The most notable is a experiment demonstrating that dropping the exact-reconstruction requirement allows cheaper techniques (such as an IPS variant) to capture the influence patterns at lower cost.

---

### Review · Reviewer_R85K · 2026-06-16

**Summary Of Contributions:**

This paper proposes a path-integrated loss-gradient kernel (PLGK) for analyzing trained neural networks. The basic idea is to represent each example by the sequence of loss gradients it induces along the training path, and then compare examples through inner products between these path features. This changes earlier one-sided influence-style quantities into a symmetric positive semi-definite kernel.

The paper uses this object in two related ways. First, it defines IPS, a cosine-normalized similarity measure between examples based on their training-path loss gradients. Second, it develops an auditing method that decomposes changes in loss or prediction into contributions from individual training examples, with a trapezoidal correction to improve reconstruction accuracy.

I found the paper interesting and fairly coherent. The main idea is simple but useful: loss-gradient path features give a clean connection between kernel similarity and training-data attribution. The experiments show that the audit can reconstruct training dynamics with high fidelity on MNIST and SVHN, and the qualitative examples are helpful. I especially liked the adversarial analysis, where the authors argue that adversarial examples receive large gross influence from the training set but that these influences cancel in signed mode space.

The main weaknesses are scope and positioning. The method is expensive and the experiments are still mostly on small vision models. Also, the paper is close to prior work on path kernels and data attribution, so the novelty boundary needs to be stated carefully. I would also be interested in seeing whether this framework has a simple extension to language models or transformer-based models, even in a small toy setting.

**Additional Comments:**

I focused my review on the main construction, empirical evidence, and positioning against related work. I did not independently check every appendix derivation, but the main PSD claim seems sound once PLGK is written as an inner product over loss-gradient path features. My main concerns are about scope, scalability, and how broadly the empirical conclusions should be stated.

**Audience:**

Yes

**Audience Explanation:**

I think the paper would interest readers working on interpretability, data attribution, training dynamics, and kernel views of neural networks. Even if the method is currently expensive, the framing is useful: it treats the training run itself as something that can be queried and decomposed. The adversarial analysis is also interesting enough to be worth reporting, provided the scope of the evidence is made clear.

**Broader Impact Concerns:**

I do not see major broader impact concerns. The work is mainly about interpretability and auditing, which could help with debugging and understanding model behavior. The adversarial analysis has some dual-use potential, since better understanding adversarial mechanisms can inform both defenses and attacks, but I think a short note acknowledging this would be sufficient.

**Claims And Evidence:**

Yes

**Claims Explanation:**

The main PSD-kernel claim is convincing once PLGK is written as an inner product over concatenated loss-gradient path features. The auditing results are also reasonably supported in the small settings studied, especially through reconstruction error and correlation.

My reservations are mostly about generality. The evidence supports the method as a high-fidelity diagnostic tool for the studied models, but it does not yet show that the method is practical or equally informative at larger scale. The adversarial mode-cancellation story is interesting, but I would like either more robustness checks or more careful wording around how general the mechanism is.

**Requested Changes:**

1. *Clarify the novelty relative to prior path-kernel and data-attribution work.* The paper should more directly explain what is new compared with Domingos-style path kernels, TracIn, TRAK, and in-run Shapley methods. My current understanding is that the main novelty is the loss-gradient symmetrization that gives a PSD kernel, plus the use of the same object for IPS and high-fidelity auditing. This should be stated more clearly.
2. *Calibrate the claims about scope and scalability.* The current experiments are small-scale, and the dense audit cost is high. I think this is acceptable if the paper is presented as a diagnostic framework, but the authors should avoid implying that it is already a broadly practical auditing method for large neural networks. A clearer discussion of when the method is expected to be usable would help.
3. *Add limited robustness checks or soften the conclusion.* The mode-cancellation analysis is one of the strongest parts of the paper, but it would be more convincing with a small number of additional checks, such as extra seeds, another attack, or another small architecture. If this is not feasible, the paper should phrase the result more cautiously as evidence for a possible mechanism rather than as a general explanation.
4. *Add a small transformer/language-model extension or a concrete sketch of one if possible.* I do not expect a full LLM-scale audit, but a tiny transformer, character-level LM, synthetic sequence task, or even a detailed design sketch would make the paper more relevant to current model settings. For example, the authors could explain how token-level loss gradients, sequence-level aggregation, checkpoint sampling, or LoRA-only gradients would fit into PLGK/auditing.

---

> ### Author Response · Authors · 2026-06-26
> **Response to Reviewer R85K**
>
> We thank the reviewer for the careful and constructive review, and in particular for recognizing the core contributions: the loss-gradient path features give a clean connection between kernel similarity and training-data attribution, and for highlighting the adversarial mode-cancellation analysis. The reviewer raises four requested changes, centering on (1) novelty positioning relative to prior path-kernel and attribution work, (2) calibration of scope/scalability claims, (3) robustness of the mode-cancellation result, and (4) relevance to transformer/language-model settings. We address each below.
>
> ## 1. Clarifying novelty relative to prior path-kernel and data-attribution work
> The reviewer's own summary of the novelty (the loss-gradient symmetrization yielding a PSD kernel, plus reuse of the same object for IPS and high-fidelity auditing) matches our intended contribution exactly. The comparisons to Domingos-style path kernels, TracIn, TRAK, and in-run Shapley already exist in Section 2 and the Section 3.3 roadmap, but were distributed across the paper rather than stated as a single contribution claim. We added an explicit contribution statement to the intro, covering the two main claims.
>
> ## 2. Calibrating scope and scalability claims
> We agree the paper is best framed as a high-fidelity diagnostic framework rather than a broadly practical large-model auditing method, and have calibrated the claims accordingly.
>
> To address this, we made the following changes:
>
> 1. Revised the conclusion to explicitly position high fidelity auditing as a diagnostic for small-to-moderate models.
> 2. Added a new scaling and extension to larger models discussion section (Section 7, full details in Appendix) presenting preliminary results on CIFAR ResNet-18-GN and ViT-Tiny audit benchmarks, establishing the approximately linear cost-in-parameters trend. We also show that dropping the 'exact-reconstruction' requirement allows cheaper techniques (e.g. IPS) to still capture influence patterns, unlocking at least ImageNet scale.
>
> ## 3. Robustness of the mode-cancellation analysis
> To address this, we made the following changes:
>
> 1. Clarified (footnote at first mention) that all non-illustrative adversarial results use four L∞ attacks (FGSM, PGD-L∞, BIM-L∞, and DeepFool-L∞) and confirmed the cancellation signature holds.
> 2. Added a caveat paragraph near the end of Section 6.3 which discusses the mechanism's limitations: it is consistent across two architecture/dataset pairs and four L∞ gradient-based attacks, but only tested on small CNNs under using L∞ attacks. We re-emphasized that we view this adversarial framework as a well-supported mechanism rather than a universal account.
> 3. Added a conclusion sentence re-iterating the adversarial theories limited demonstration, and scoping extending it to e.g. transformers as future work
>
> ## 4. Transformer / language-model relevance
> To address this, we made the following changes:
>
> 1. Added a CIFAR ViT-Tiny/4 audit implementation (Appendix), demonstrating the framework applies to a real transformer architecture, confirming the method works out of the box on attention-based models.
> 2. Included in the appendix a small subsection discussing sequence-level aggregation (per-sequence is default, per-token is available at higher cost), checkpoint sampling, and LoRA-only gradients (perfectly valid, but will only capture influence of the fine-tuning process, ignoring any per-training dynamics).

---

### Author Response · Authors · 2026-06-27
**General Response to Reviewers**

We thank all three reviewers for their careful and constructive feedback. We were glad that the reviewers found the core construction sound and interesting: that the loss-gradient symmetrization recovers a true Mercer kernel from the training trajectory, with a high-fidelity audit enabled by using a trapezoidal correction, and the adversarial mode-cancellation analysis. The reviewers' concerns converged on three themes: scope/scalability, novelty positioning, and the generality of the adversarial mechanism. We summarize each here along with the changes made; reviewer-specific points are addressed in the individual responses.

## Framing: a diagnostic framework with a scalable companion tool

Several concerns stem from how broadly the method's practicality was claimed. We have reworked the paper (primarily introduction and conclusions) to state its contribution clearly: PLGK yields a **high-fidelity diagnostic** (dense auditing) for small-to-moderate models, with a sketch towards using IPS as the path forwards to still capturing influence patterns in higher cost scenarios. Our revisions are targeted to support these scoped claims, consistent with TMLR's emphasis on correctness and value to some audience rather than on large-model state-of-the-art results.

## Scope and scalability

To address this, we made the following changes:

1. Added several new scaling/profiling experiments in the Appendix: CIFAR ResNet-18-GN audit, CIFAR ViT-Tiny/4 audit, and a partial-ImageNet ResNet-18-GN run.
2. Demonstrated IPS as a more scalable technique: ~95% correlation with dense-audit per-sample ranking on CIFAR, while allowing scaling to ImageNet at ~2.7× training overhead.
3. Recalibrated the conclusion to frame dense auditing as a diagnostic tool for smaller models, with limitations explicitly mentioned.

## Novelty positioning

To address this, we made the following changes:

1. Added an explicit contribution statement (intro) explicitly framing what we believe are our two major contributions: loss-gradient symmetrization leading to Mercer kernel, and one object serving both IPS and high-fidelity auditing.
2. Stated the unique value of auditing relative to counterfactual (e.g. TRAK, appendix) methods: it decomposes a single realized run across parameter, time, and SVD-mode axes (the capability underlying the adversarial analysis section) which after-kernel and counterfactual methods cannot provide.

## Generality of the adversarial mechanism

To address this, we made the following changes:

1. Clarified that all non-illustrative adversarial results use four L∞ attacks (FGSM, PGD, BIM, DeepFool), with the cancellation signature consistent across them.
2. Added language clarifying the mechanism's evidence: consistent across two architecture–dataset pairs and four L∞ gradient-based attacks, established on small CNNs, and offered as a well-supported candidate mechanism rather than a universal account.

## Additional corrections and additions

1. Corrected the audit completeness normalization inconsistency (Definition 9 / Eq. 17 and Eq. 33); reported results are unaffected.
2. Added a Broader Impact Statement covering the dual-use nature of the adversarial analysis.